# Effector–host interactome map links type III secretion systems in healthy gut microbiomes to immune modulation

Pseudomonadota (formerly Proteobacteria) are prevalent in the commensal human gut microbiota, but also include many pathogens that rely on secretion systems to support pathogenicity by injecting proteins into host cells. Here we show that 80% of Pseudomonadota from healthy gut microbiomes also have intact type III secretion systems (T3SS). Candidate effectors predicted by machine learning display sequence and structural features that are distinct from those of pathogen effectors. Towards a systems-level functional understanding, we experimentally constructed a protein–protein meta-interactome map between human proteins and commensal effectors. Network analyses uncovered that effector-targeted neighbourhoods are enriched for genetic variation linked to microbiome-associated conditions, including autoimmune and metabolic diseases. Metagenomic analysis revealed effector enrichment in Crohn's disease but depletion in ulcerative colitis. Functionally, commensal effectors can translocate into human cells and modulate NF-κB signalling and cytokine secretion in vitro. Our findings indicate that T3SS contribute to microorganism–host cohabitation and that effector–host protein interactions may represent an underappreciated route by which commensal gut microbiota influences health.

Host-associated microbiota influences human health in complex, genotype-dependent ways. Especially the human gut microbiome, which is dominated by Firmicutes, Bacteroidetes and Pseudomonadota[1] (formerly Proteobacteria[2]), can alter the risk of diverse conditions, including metabolic disorders, and autoimmune and neurodegenerative diseases[3]. However, the underlying molecular mechanisms are incompletely understood. Most studies have focused on metabolites, extracellular microorganism-associated molecular patterns, or community-level microbiome properties[4]. However, the role of intracellular bacteria–host protein interactions is largely unexplored. The potential impact of such interkingdom protein interactions is illustrated by viral proteins in asymptomatic or non-acute infections, which can influence cellular signalling and cell physiology and thereby contribute to complex diseases in a likely host-genetics dependent manner[5–8].

In bacteria, the type III secretion system (T3SS) is a well-characterized apparatus for delivering proteins into eukaryotic cells. The T3SS is a highly conserved 'needle and syringe'-like machinery found in Pseusomonadota to inject bacterial proteins into host cells[9]. T3SS and their substrate effectors have been studied almost exclusively in human and plant pathogens such as *Yersinia*, *Pseudomonas* or *Salmonella*, for which protein translocation is a key pathogenic strategy. In the host, translocated effectors manipulate cellular processes including cytoskeletal dynamics or immune signalling to ensure bacterial survival and promote transmission[9]. Thus, the T3SS has traditionally been framed as a virulence determinant.

Insights from plant and insect systems challenge this pathogen-centric view of the T3SS. Many commensal and beneficial microorganisms for these hosts deploy T3SS or analogous machinery

✉e-mail: andreas.zanzoni@univ-amu.fr; pascal.falter-braun@helmholtz-munich.de

to translocate proteins that promote symbiosis or fine-tune host immunity[10–12]. Effector–host protein–protein interaction maps in plants further reveal that both pathogenic and mutualistic microorganisms converge on central host signalling nodes, suggesting conserved principles by which injected proteins can modulate eukaryotic biology across diverse symbioses[12–14].

Despite the emerging broader conceptual importance of protein injection and cross-kingdom protein interactions in diverse microorganism–host systems, it is unknown whether analogous mechanisms operate in the healthy human gut. Here we investigate the distribution, diversity and host interactions of T3SS and their effectors in commensal Pseudomonadota from human guts. By integrating comparative genomics, structural prediction, functional assays and host protein interaction networks, we uncover an underappreciated layer of direct, protein-mediated communication between commensal microorganisms and the human host, with implications for immune modulation, microbial competition and complex disease biology.

## Results

### T3SS are common in the human gut microbiome

We first analysed reference genomes of Pseudomonadota strains from healthy gut and stool samples isolated, for example, by the human microbiome project[15]. Using EffectiveDB[16], a widely used tool for secretion system identification, we detected complete T3SS in 44 of the 77 genomes (Supplementary Data 1). To expand the scope, we analysed genomes of 4,752 phylogenetically diverse strains of the human intestinal bacteria collection (HiBC)[17], Broad Institute–OpenBiome Microbiome Library (BIO–ML)[18] and Global Microbiome Conservancy (GMC)[19]. Of the 568 Pseudomonadota genomes, 449 (79%) have complete T3SS (Extended Data Fig. 1); similar proportions have T4SS (315) and T6SS (474), which may also inject effectors into host cells among other functions[20] (Extended Data Fig. 1 and Supplementary Data 2). Together, 527 of the 568 Pseudomonatoda genomes (92%) have at least one host-directed secretion system. Because culturing can bias taxon representations, we also screened 16,179 high- and intermediate-quality Pseudomonadota metagenome-assembled genomes (MAGs)[21–23], finding complete T3SS in 770 (5%) MAGs (Extended Data Fig. 1 and Supplementary Data 3). Notably, T3SS were only detected in Gammaproteobacteria, but not in Beta- or Epsilon-proteobacteria (except in *Helicobacter* strains), and were especially common among *Escherichia* (Fig. 1a and Supplementary Data 3). Among the T3SS-positive (T3SS+) species, 24 matched representatives in two cohorts of the Weizmann Institute of Science[24]: 59.4% of individuals in the Israeli and 47.1% in the Dutch cohort harboured potentially T3SS+ species in their gut microbiome at 0.80% and 0.48% relative abundance, respectively, with *Escherichia coli* being the most common. These observations indicate that T3SS+ strains are common members of the human gut microbiota and motivated our further investigation.

### Commensal effectors are unrelated to known pathogen effectors

Using three complementary machine-learning models[25–27], 3,002 effector candidates were confidently predicted in the T3SS+ reference genomes (hereafter: strain effectors) (Supplementary Data 4) and 182 in the 770 T3SS+ MAGs (meta-effectors) (Supplementary Data 4). Because T3SS effectors are classically associated with pathogenicity, we compared these candidate effectors to 1,195 T3SS effectors of known pathogens[28]. Only 17 out of 3,002 (0.5%) strain effectors and 6 out of 182 (3%) meta-effectors showed high sequence similarity to those of pathogens (≥90% across ≥90% length) (Supplementary Data 5). To find weak similarities, we performed iterative jackhmmer[29] searches against ~124 M non-redundant bacterial sequences from UniRef90. Yet, even with this sensitive approach, significant similarity to pathogen effectors was found only for 155 commensal strain effectors (~5%) and 42 meta-effectors (22.5%) (Fig. 1b and Extended Data Fig. 1).

As effectors can be structurally related despite sequence divergence, we clustered AlphaFold[30]-predicted tertiary structures using FoldSeek[31] for a structural comparison. Surprisingly, homogeneous clusters with effectors from only commensal strains or pathogens were highly overrepresented, whereas mixed clusters II and III, reflecting common structures of effectors from pathogens and commensal strains, were depleted (Fig. 1c and Supplementary Data 6; *P* << 0,0001, empirical *P* values). Meta-effectors clustered exclusively with strain effectors, albeit close to random expectation. All results were robust over varying FoldSeek parameters and when considering only vertebrate or human pathogens (Supplementary Data 6). Thus, candidate effectors in T3SS+ strains from healthy human guts markedly differ from pathogen effectors in both sequence and structure.

We analysed all candidate effectors from the strains for annotated domains. Besides 860 proteins without any identifiable domain, among the most common finds were the diguanylate cyclase, GGDEF domain (PF00990) (58 effectors), and EAL domain (PF00563) (50 effectors), none of which was found in pathogen effectors (Supplementary Data 5). Cyclic diguanylate is a known second messenger in bacterial signal transduction, and the EAL domain is thought to be a diguanylate phosphodiesterase, thus opposing the effect of the cyclase[32]. Furthermore, we observed a PAS-fold domain (PF08447) in 32 effectors, which can function as a ligand-binding sensor[32] and in some effectors co-occurs with a guanylate cyclase domain. As cyclic dinucleotides recently emerged as important immune regulators in all kingdoms of life[33], the observation that two domains acting on the same second messenger occur at high frequency among the commensal effector candidates makes a role for this signalling molecule in interkingdom communication plausible.

### Injection of commensal candidate effectors into human cells

A key question is whether commensal candidate effectors get injected into human cells by T3SS. To enable functional studies, we cloned open reading frames (ORFs) encoding effectors from 18 bacterial strains (Fig. 1d, Extended Data Fig. 1, Supplementary Table 9 and Supplementary Data 7 and 8). The generated human microbiome effector ORFeome v1 (HuMEOme_v1) contains 910 sequence-verified, full-length ORFs representing 746 strain effectors and 164 meta-effectors (Supplementary Data 7). Cloning failure mainly resulted from failed PCR amplification without indications of toxicity. Using *Salmonella enterica* subsp. *enterica* sv. Typhimurium (*S.* Typhimurium) as a model, we established a nano-luciferase-based injection assay[34] fusing an 11-amino acid Nano-Luc HiBiT tag to the C terminus of candidate effectors expressed in bacteria. HeLa cells stably expressed the complementary LgBiT fragment, so that effector injection reconstitutes functional nano-luciferase. Specificity was ensured by inclusion of the T3SS-defective *ΔsctV* mutant for all tests. Benchmarking with six pathogen effectors demonstrated effective translocation of four. Among 97 tested candidate effectors from 11 strains, 32 were specifically and significantly injected (Fig. 1e, Extended Data Fig. 1, Supplementary Table 9 and Supplementary Data 10). The slightly higher success rate for the positive controls probably reflects phylogenetic diversity and missing chaperones and cofactors. Thus, although some false effector identifications cannot be excluded, overall, our pipeline reliably identified bona fide T3SS substrate effectors from commensal strains in healthy human guts.

Next, we assessed the functionality of T3SS in the commensals. Of the 11 strains with at least one T3SS-injectable effector, 6 could not be tested due to antibiotic resistance or transformation failure. Whereas the two *E. coli* strains yielded no signals, *Citrobacter pasteurii* and *Phytobacter massiliensis* showed occasional signals suggesting sporadic activation of the T3SS. By contrast, *Edwardsiella tarda* reproducibly and significantly injected three out of four tested effectors into HeLa cells (Fig. 1f and Supplementary Data 10). Notably, only one, Eta_3, was also positive in the *Salmonella* system, supporting the notion

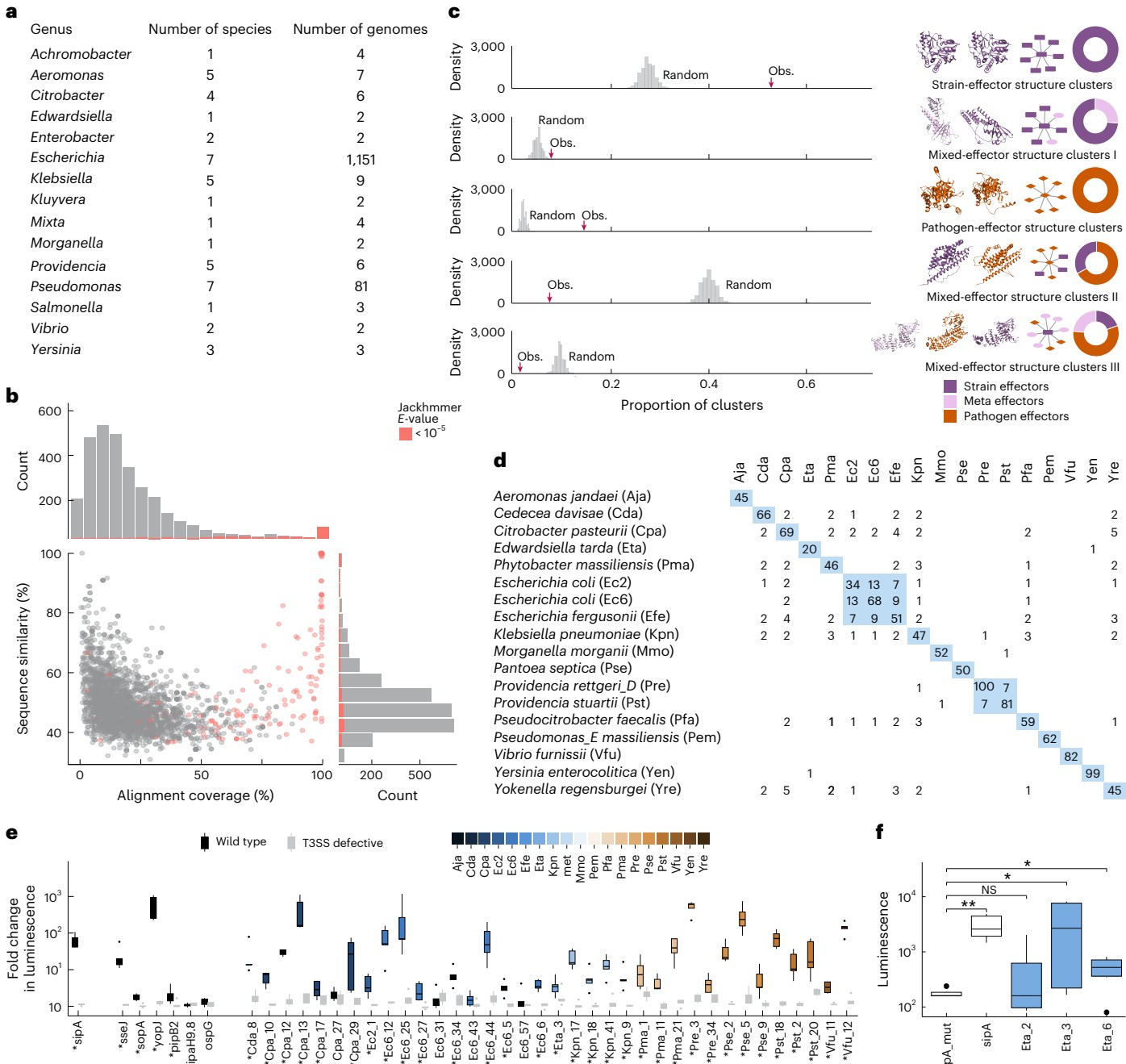

**Fig. 1 | T3SS in commensal gut bacteria. a**, Most abundant genera, species and genomes encoding complete T3SS among reference strains and MAGs from the human gut. **b**, Sequence similarity of 3,002 candidate commensal T3SS effectors with 1,195 effectors from pathogenic bacteria across alignment coverages (bottom left). Each dot represents a pairwise sequence comparison. Dot colour indicates effectors with significant and non-significant Jackhmmer results (inset legend) indicating homology to pathogen effectors. Marginal histograms display the aggregated distribution of alignment coverage (top) and aggregated sequence similarity (right), with colour indicating Jackhmmer outcome. **c**, Left, number of the structure clusters observed in FoldSeek analysis (red arrow) compared to random expectation for that group (homogeneous or mixed) in grey (exp. empirical $P < 0.0001$, $n = 10,000$, two-sided label permutation test). Middle, example structures for one cluster in the group; small networks are representative structure clusters for the group with an anchor structure in the centre and similar structures connected by links. Donut plots: proportion of proteins with origins indicated by colour in all clusters of (homogeneous or mixed) the structure-cluster group. **d**, Selection of 18 commensal Pseudomonadota strains for subsequent functional analyses.

Numbers indicate the count of shared effectors at >90% mutual sequence similarity across 90% sequence length. **e**, Injection of indicated effectors by wild-type and Δ*sctV* (T3SS-defective) *Salmonella* Typhimurium into HeLa cells detected by luminescence of reconstituted nano-luciferase (*y* axis). Control pathogen effectors (left): sseJ (A0A0F6B1Q8), sopA (Q8ZNR3) and pipB2 (A0A0F6B5H5) from *Salmonella* Typhimurium; yopJ (A0A0N9NCU6) from *Yersinia pseudotuberculosis*; and ipaH9.8 (Q8VSC3) and ospG (Q99PZ6) from *Shigella flexneri*. SipA is an assay control used as reference. Asterisks denote statistically significant differences between the wild-type and Δ*sctV*-negative strains (two-sided Wilcoxon test; five biological repeats with four technical repeats each). **f**, Injection of effectors from gut commensal *Edwardsiella tarda* into HeLa cells. SipA tested in wild-type and Δ*sctV Salmonella* Typhimurium were used as positive and negative controls, respectively (two-sided Wilcoxon test, *$P < 0.05$, **$P < 0.001$; NS, not significant; seven biological repeats with four technical repeats each). Raw data and precise *P* values for all panels are found in Supplementary Data 1, 3, 5, 6 and 10 as described in Supplementary Information. Boxplots (**e**,**f**) show the median (centre line) and the interquartile range (IQR, box), with whiskers extending to minimum and maximum values within 1.5× IQR.

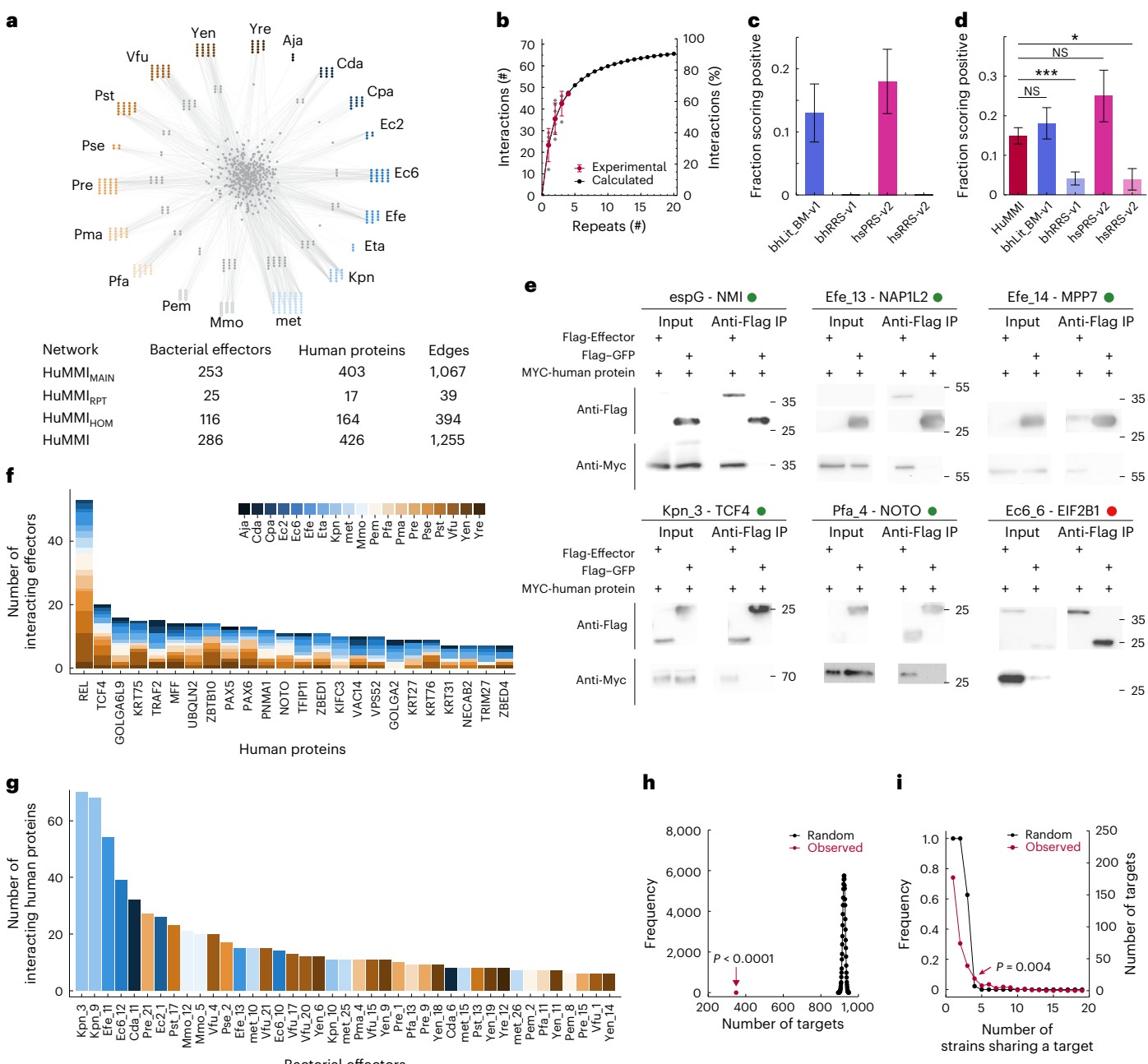

**Fig. 2 | Meta-interactome network map of bacterial effectors with human proteins. a**, Verified human microbiome meta-interactome (HuMMI) map; coloured nodes indicate effectors from strains according to colour legend in **f**. Grey nodes represent human proteins; outer layer human proteins are targeted only by the nearest strain, central human proteins by effectors from multiple strains. **b**, Sampling sensitivity: saturation curve calculated from HuMMI$_{RPT}$. Red dots represent the average of verified interactions found in any combination of indicated number of repeat screens; diamonds denote interaction counts per experiment over all sequential experiment combinations; error bars indicate standard deviation; black dots and line represent calculated saturation curve. **c**, Assay sensitivity: percentage of identified interactions from bhLit_BM-v1 ($n = 54$ pairs), bhRRS-v1 ($n = 72$ pairs), hsPRS-v2 ($n = 60$ pairs) and hsRRS-v2 ($n = 78$ pairs) in the Y2H system used for network mapping. Error bars represent the s.e. of proportion. **d**, Validation rate of a random sample of HuMMI interactions ($n = 294$ pair configurations) compared to four reference sets in the yN2H validation assay: bhLit_BM-v1 ($n = 94$ pair configurations), bhRRS-v1 ($n = 144$ pair configurations), hsPRS-v2 ($n = 44$ pair configurations) and hrRRS-v2

($n = 51$ pair configurations). Two-sided Fisher's exact test, *$P = 0.04$, ***$P = 0.0006$ (Supplementary Data 14). Error bars represent s.e. of proportion. **e**, Co-immunoprecipitation of MYC-tagged human proteins by Flag-tagged effectors or Flag–GFP as negative control. Input, cell lysates; green dots, successful co-immunoprecipitation; red dot, no co-immunoprecipitation; effector espG of *Escherichia coli* (Q7DB50) as positive control (one biological replicate). Molecular mass markers are given in kilodaltons. **f**, Most-targeted human proteins interacting with the indicated number of effectors from different strains. Colours represent strains according to indicated legend (full statistics in Supplementary Data 11). **g**, Most highly connected effectors interacting with the indicated number of human proteins (Supplementary Data 11). **h**, Observed number of effector-interacting human proteins compared to random expectation (two-sided permutation test, $P < 0.0001$; $n = 10,000$). **i**, Frequency distribution of human proteins targeted by effectors from the indicated number of different strains (red) compared to random expectation (two-sided permutation test, $P = 0.004$; $n = 10,000$).

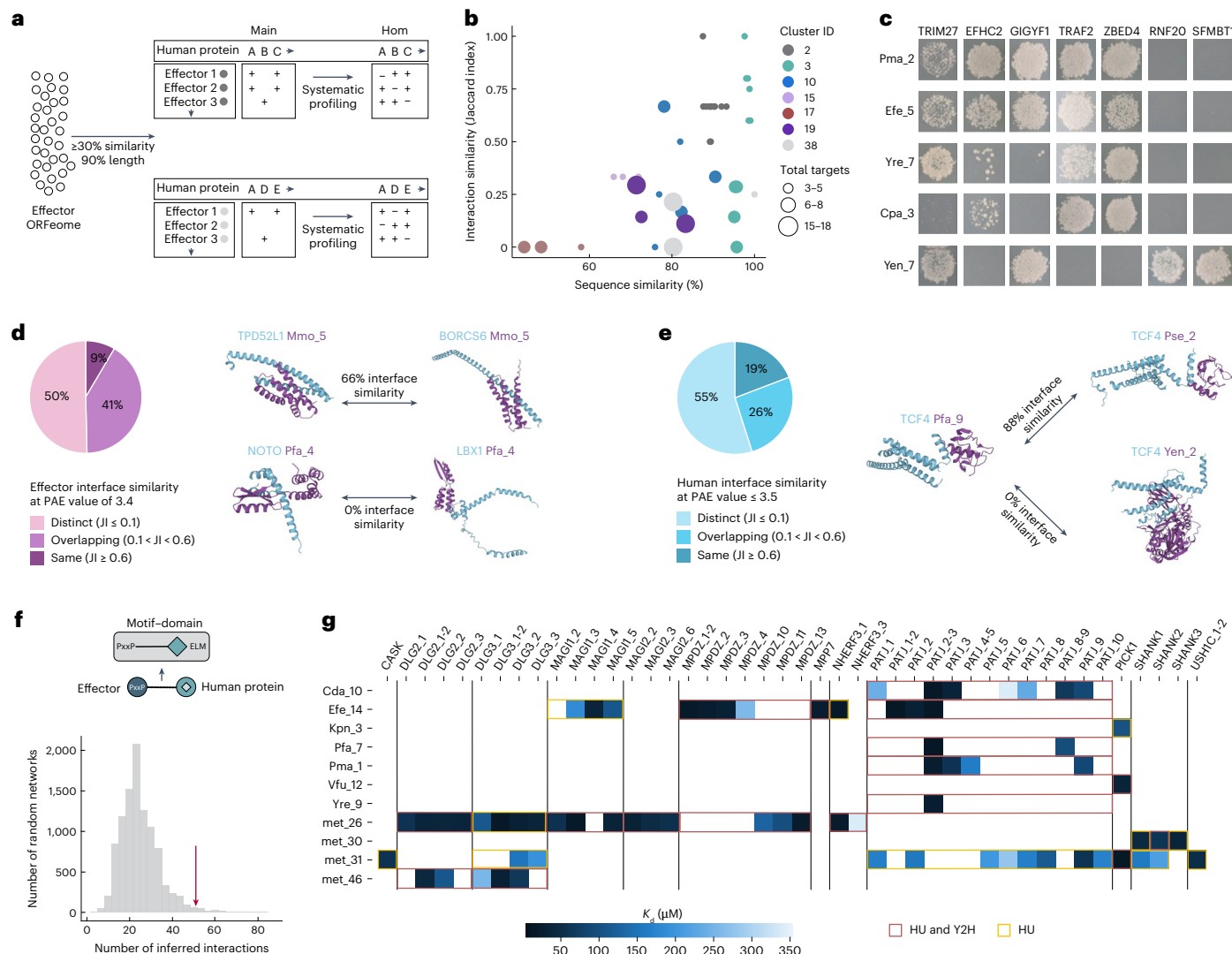

**Fig. 3 | Interaction specificity and interaction motifs. a**, Schematic representation of the systematic interaction profiling of homologous effectors. **b**, Scatterplot of mutual sequence similarity and Jaccard interaction similarity for all effector pairs in the indicated homology groups. The union of human proteins targeted by each effector pair is indicated by node size as denoted in the legend. Individual data in Supplementary Data 16. **c**, Yeast growth in one representative of four repeats of all effector–human interactions tested for homology cluster 3. **d**, Left, proportions of human-protein targets interacting with the same bacterial effector, grouped by interface similarity on the basis of the Jaccard index (JI) of target-contacting residues, categorized as: distinct (JI ≤ 0.1), overlapping (0.1 < JI < 0.6) and same (JI ≥ 0.6). Right, Mmo_5 interacts with example human proteins via the same interface (top, 66% overlap), whereas Pfa_4 uses distinct interfaces. **e**, Proportions of different effectors interacting

with the same human protein, grouped by interface similarity as in **d**. Example interface models (right) show effector binding to the same (Pfa_9, Pse_2) or distinct (Pfa_9, Yen_2) interfaces on human TCF4. **f**, Count of domain–motif interfaces identified in HuMMI_MAIN matching at least one stringency criterion (arrow) compared to random expectation (one-sided permutation test, $P = 0.0137$; $n = 10,000$). All data related to **c**–**f** are available in Supplementary Data 17. **g**, Results of holdup assay and comparison with Y2H results. Indicated PDZ domains of human proteins shown on $y$ axis were tested against 10-amino-acid C-terminal peptides of the effectors indicated on top. Calculated dissociation constant ($K_d$) values as indicated. Overlap between holdup (HU) and Y2H on protein level is indicated by coloured frames. Precise $P$ values and $n$ for each test are shown in Supplementary Data 19.

that missing cofactors may have caused false negatives in the first experiment. Overall, these data demonstrate that functional T3SS are present in strains from healthy human guts and can deliver identified effectors into human cells.

**A microbiome–host protein–protein meta-interactome map**
Next, we explored possible functions of the commensal effectors by systematically mapping their physical interactions with host proteins using our multi-assay mapping pipeline[35] (Extended Data Fig. 2). Screening all cloned effectors against the full human ORFeome9.1[36] identified 1,067 interactions constituting the human-microbiome meta-interactome (HuMMI) main dataset (HuMMI_MAIN) (Fig. 2a). Three

repeat screens with 290 effectors and 1,440 human proteins yielded 39 interactions (HuMMI_RPT) and indicated a sampling sensitivity of ~32% for the main screen (Fig. 2b), matching previous studies[37]. Lastly, we addressed how sequence similarity affects effector interaction profiles. We grouped effectors with ≥30% sequence identity (Supplementary Data 11) and experimentally tested them against the union of their interactors from the main screen. The resulting HuMMI_HOM dataset contains 394 interactions, of which 181 are non-redundant. Altogether, HuMMI contains 1,255 unique verified interactions between 286 effectors and 426 human proteins (Fig. 2a and Supplementary Data 11).

To experimentally assess data quality, we assembled a positive control set of 67 well-documented binary interactions of pathogen

effectors with human proteins (bacterial human literature binary multiple (bhLit_BM-v1)) and a negative control set of random effector–human protein pairs (bacterial host random reference set (bhRRS-v1)) (Supplementary Data 12). Benchmarking our yeast-two-hybrid (Y2H) assay with these, alongside established human reference sets (hsPRS-v2 and hsRRS-v2)[38], indicated an assay sensitivity of 13% and 17.5%, respectively, matching previous observations[35,37,38] (Fig. 2c and Supplementary Data 13). No negative control scored positive, demonstrating the reliability of our system. Next, we assessed the biophysical quality of HuMMI using the yeast nanoluciferase-two-hybrid assay (yN2H)[38] benchmarked against the four reference sets. Across thresholds, sets with bacterial proteins yielded fewer positive-scoring pairs than the human sets (Extended Data Fig. 2). As this included the negative controls and no effector toxicity was observed, prokaryotic proteins appear harder to test in this assay system, reinforcing the need for tailored reference sets. The 172 randomly selected HuMMI interactions were statistically indistinguishable from the positive control sets (Fig. 2d, Extended Data Fig. 2 and Supplementary Data 14). Thus, the biophysical quality of HuMMI is on par with well-documented literature interactions.

We aimed to demonstrate that interactions can occur within the human cell environment. We performed immunoprecipitation experiments in HEK293 cells (RRID: CVCL_0045, DSMZ) using Flag-tagged effectors and negative control Flag–GFP as baits and detecting the MYC-tagged human interaction partner by western blot. Of 32 pairs including 4 positive controls, 18 pairs and 3 controls yielded meaningful data, while 10 could not be evaluated due to unspecific binding of the human protein (3) or poor expression (7). Only one of the control pairs was positive, whereas 13 of the 18 candidate pairs yielded detectable bands specifically in the effector immunoprecipitation (Fig. 2e and Extended Data Fig. 2). Together, these results demonstrate that HuMMI contains biophysically reliable interactions that are robustly detectable in different assays and occur in human cellular environments. Importantly, functional effects may go in both directions and while in most cases effectors probably perturb the host cell, intracellular immune receptors may also recognize effectors to then initiate defence responses.

We started the functional exploration by analysing the topology of the microorganism–host interaction network (Fig. 2f,g). The degree distribution of HuMMI_MAIN shows that numerous human proteins interact with multiple effectors, often from different species (Fig. 2f and Supplementary Data 11). Random sampling demonstrates highly significant effector convergence on few host interactors (Fig. 2h), a phenomenon linked to the functional importance of the targeted host proteins as shown in plant–pathogen systems[13]. Moreover, interactions of human proteins with effectors from four bacterial strains are highly significant and unlikely to result from random processes (Fig. 2i and Supplementary Data 11). Thus, 60 human proteins are subject to effector convergence, highlighting their potential importance for microbiome–host interactions. To explore overlap with pathogen effectors, we extracted 265 high-quality binary interactions between 217 human proteins and 80 effectors from 17 pathogenic strains from IntAct[39] (Supplementary Data 15). We found a numerically low, albeit significant, number of 12 human proteins targeted by both groups ($P = 0.014$, Fisher's exact test, odds ratio = 2.26), of which 3 are subject to convergence by commensal effectors ($P = 0.067$, Fisher's exact test, odds ratio = 3.37, Supplementary Data 11). Although limited by sample size, experimental differences and the non-systematic nature of the pathogen data, these findings support both overlap and lifestyle-dependent specificity in commensal and pathogenic effector targeting[40].

### Structural features mediating effector–host interactions

Many inference approaches assume that sequence similarity implies functional and interaction similarity, and such similarity could also underlie convergence. However, in the homology clusters of the systematically tested HuMMI_HOM (Fig. 3a), we found that sequence and interaction similarity are only poorly correlated; instead, sequence similarity merely defines an upper limit for interaction similarity. For instance, cluster 3 contains 7 effectors sharing >90% sequence similarity, yet their interaction profile similarities range from identical to complementary (Fig. 3b,c and Supplementary Data 16). Conversely, clustering effectors unrelated in sequence and structure by their pairwise interaction similarity in HuMMI_MAIN identified substantial overlap outside homology clusters (Extended Data Fig. 3), suggesting that dissimilar effectors can have similar functions in the host. Thus, host effector function as measured by protein-interaction profiles is largely independent of overall sequence similarity.

To gain structural insights and potential functional leads, we modelled effector–host protein interactions using AlphaFold-Multimer, obtaining predictions for 123 pairs (10%). For proteins with multiple interactors, we classified interfaces as 'same' (≥60% shared contacting residues), 'different' (<10% overlap) or 'overlapping' (Fig. 3d,e and Extended Data Fig. 3). For instance, Mmo_5 binds to TPD52L1 and BORCS6 via the same interface, whereas Pfa_4 interacts with NOTO and LBX1 with different interfaces, possibly enabling simultaneous interactions with both (Fig. 3d and Supplementary Data 17).

Analogously, Pse_2 and Pfa_9 bind to the same interface of TCF4, whereas Yen_2 targets a different part of the protein (Fig. 3e). Identical interface binding was more frequent on human proteins than on effectors, suggesting the importance of targeting functions linked to specific domains. Mapping the binding interfaces to domain annotations strengthened this hypothesis, as even effectors binding via different interfaces may target the same domain (for example, the DNA-binding domain of LBX1). More commonly, however, effectors with different interfaces bind to distinguishable parts of the host protein. Efe_11 and Kpn_9 bind the same interface in the TRAF2 E3 ubiquitin ligase domain, whereas Pem_8 targets the C-terminal MATH domain of TRAF2, which mediates trimerization and receptor binding. Similarly, on REL, Pma_4 binds the DNA-binding and Yen_11 the dimerization domain.

Beyond large interfaces, many interactions are mediated by short linear motifs (SLiMs) in intrinsically disordered regions that bind to specific protein pocket-forming domains[41]. As AlphaFold often misses such interactions[42], we used the orthogonal mimicINT approach to identify SLiM–domain interactions, which matches interaction pairs to known SLiM–domain templates[43] (Fig. 3f). This identified putative interfaces for 54 HuMMI_MAIN interactions involving bacterial host-like SLiMs binding to human domains (Supplementary Data 18), of which 51 passed at least one (Fig. 3f, $P = 0.0137$, exp. $P$ value) and 22 passed two stringency criteria (Extended Data Fig. 3, $P = 0.0005$, exp. $P$ value). Some of the matched motifs encompass phosphorylation sites that interact with kinases or phosphorylation-dependent binding domains such as SH2 domains. Conversely, although several commensal effectors encode predicted enzymatic domains (Supplementary Data 5), using an analogous approach we found no case in which these engage cognate substrate motifs on host proteins, and only a single effector-domain–SLiM match consistent with known docking specificity: the calcineurin-like phosphoesterase domain (PF00149) of Efe_1 and the canonical LxVP docking motif in VAC14. The largest group of 23 interactions involved PDZ domains in human proteins binding PDZ-binding motifs (PBM) in the C terminus of the bacterial interaction partners. PDZ domain-containing proteins commonly mediate functions important for microorganism–host interactions including cell–cell adhesion, protein trafficking and immune signalling[44]. To experimentally validate these interfaces, individual and tandem PDZ domains from 13 human proteins and C-terminal peptides from 16 interacting bacterial effectors were tested via the quantitative in vitro interaction holdup assay[45]. Of 23 Y2H pairs, 16 (70%) showed at least one PDZ–peptide interaction, thus validating the mode of interaction (Fig. 3g and Supplementary Data 19). In three instances, two PDZ domains arranged in tandem were required for the interaction, suggesting that some Y2H pairs might have been missed by the

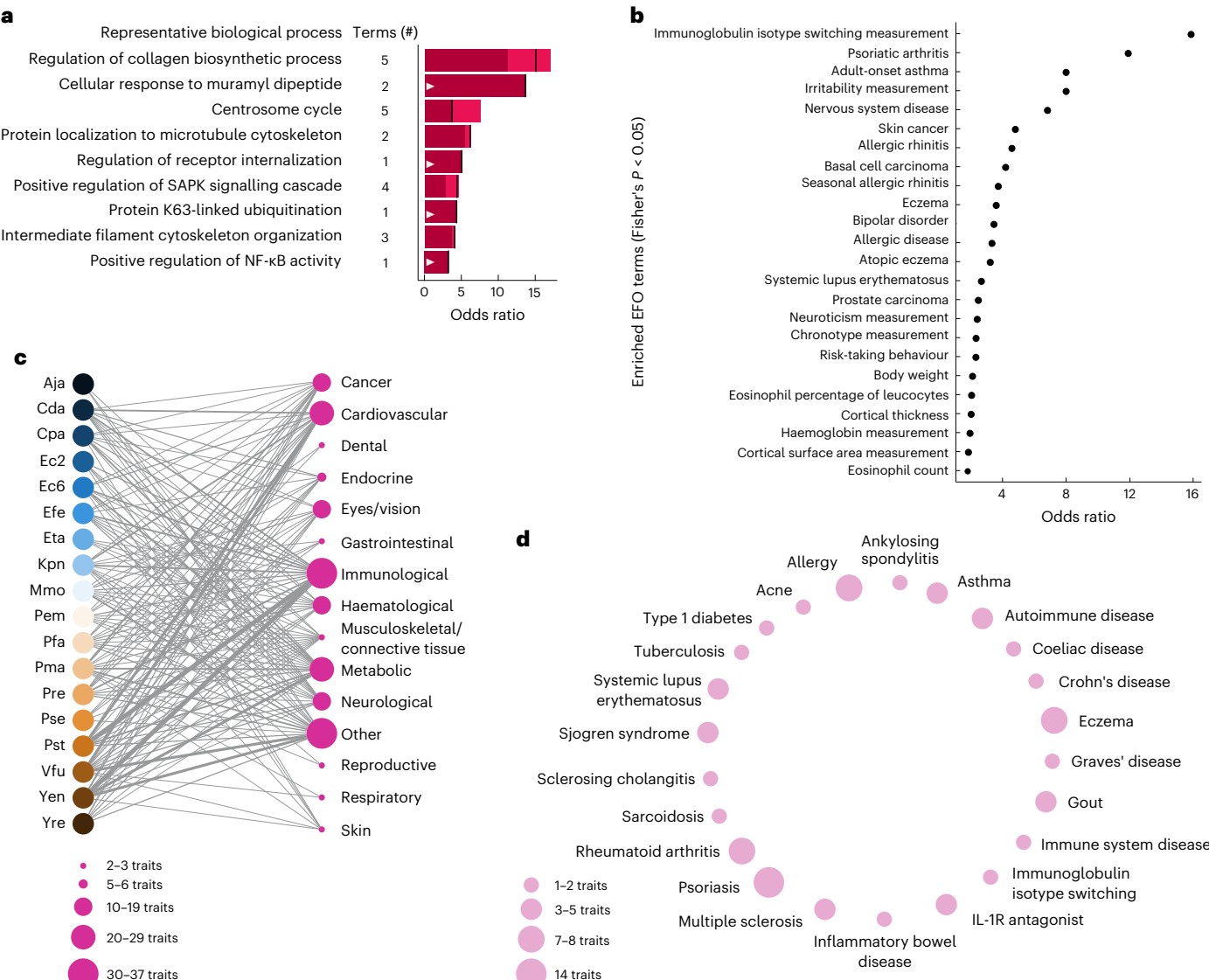

**Fig. 4 | Function and disease association of microbially targeted human proteins. a**, Odds ratios of representative functional annotations enriched among effector-targeted human proteins (FDR < 0.05, Fisher's exact test with Bonferroni FDR correction). Terms (#) show the number of represented terms. The lowest and highest odds ratios observed for the represented group are indicated by light shaded areas of bars. Black line indicates odds ratios for shown representative terms. White triangles indicate functions also enriched in pathogen targets. **b**, Genetic predisposition for traits and diseases enriched among human genes encoding effector-interacting proteins in HuRI ($\alpha$ = 0.05, Fisher's exact test; $n$ = 349). The odds ratio in **a** and **b** estimates the effect size of significant function/trait (two-sided Fisher's exact test FDR < 0.05) and is calculated as the odds of function-annotated/trait-associated human genes encoding effector targets to function-unannotated/trait-unassociated human genes encoding effector targets in the target set, divided by the same ratio in the HuRI set (see Methods). **c**, Disease groups for which genetic predisposition proteins are enriched in network neighbourhoods of effectors of the indicated strains. Trait node size corresponds to number of significantly targeted traits in that group as indicated in the legend. Thickness of strain–group edges reflects the number of underlying significant effector–trait links ($\alpha$ < 0.01 and odds ratio > 3, two-sided Fisher's exact test). **d**, Specific diseases underlying the 'immunological' group in **c**. Node size reflects the number of underlying effector–trait associations as indicated in the legend. Precise $P$ values and $n$ for all tests are provided in Supplementary Data 23.

holdup method due to untested combinations. As for the predicted globular interfaces, for human proteins with multiple PDZ domains, different effectors often target different domains, demonstrating specificity and functional specialization (Fig. 3g). Thus, while overall effector sequence similarity does not correlate with interaction profiles, structural modelling showed that some effectors target similar interfaces and domains, suggesting shared functions, whereas others bind distinct domains pointing to functional specialization.

## Effector-targeted functions and disease modules
We explored effector target functions using Gene Ontology (GO) enrichment (Fig. 4a, Extended Data Fig. 3 and Supplementary Data

20). Among the most enriched functions was 'response to muramyl dipeptide (MDP)', a bacterial cell wall-derived peptide. Intriguingly, the MDP receptor, NOD2, is a major susceptibility gene for Crohn's disease[46], a gut autoimmune disease with a strong aetiological microbiome contribution[46]. Central immune signalling pathways are also enriched, namely, the NF-κB and the stress-activated protein kinase and Jun-N-terminal kinase (SAPK/JNK) pathways. Remarkably, five significantly targeted convergence proteins belong to the NF-κB module (Extended Data Fig. 3), one of the evolutionarily oldest immune pathways in animals[47]. Using the Recon3D human genome-scale metabolic model[48], we further found significant enrichment for metabolic enzymes among the human interactors

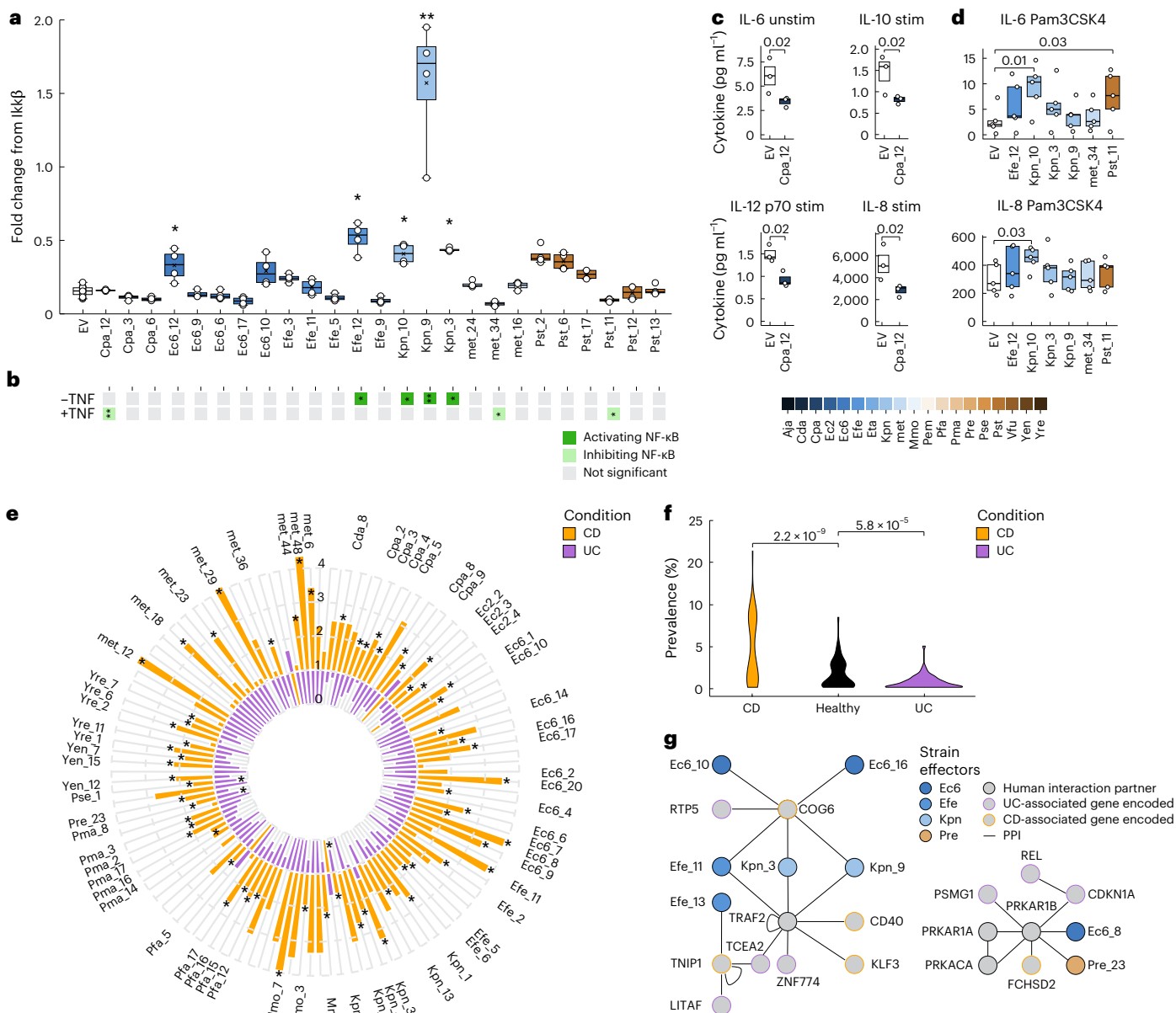

**Fig. 5 | Effector impact on human cell function and clinical prevalence in IBDs.**
**a**, Relative NF-κB transcription reporter activity in HEK293 cells expressing the indicated effectors at baseline conditions (no TNF) (Kruskal–Wallis test with Dunn's post hoc comparisons, *$P < 0.05$, **$P < 0.01$; $n = 4$ biological replicates). Boxes represent IQR, black line indicates the mean, whiskers indicate highest and lowest data point within 1.5× IQR. **b**, Summary of significant influence of effectors on normalized NF-κB transcriptional reporter activity at baseline conditions (−TNF) and after TNF stimulation (+TNF) (Kruskal–Wallis test with Dunn's post hoc comparisons, *$P < 0.05$, **$P < 0.01$; $n = 4$ biological replicates). **c**,**d**, Concentration of cytokines secreted by Caco-2 cells transfected with indicated effectors at basal conditions (unstim) or after stimulation with a proinflammatory cocktail (stim) (**c**) or with Pam3CSK4 (**d**). EV, empty vector mock control. Numbers above brackets indicate $P$ values calculated by Kruskal–Wallis test with Dunn's post hoc comparisons; $n = 3$ (**c**) and $n = 5$ (**d**) biological replicates. Boxes represent IQR, black line indicates the mean, circles indicate

individual data points. **e**, Radial barplot showing fold change in prevalence of 122 bacterial effectors in metagenomes of individuals with Crohn's disease (CD; $n = 504$ patient samples, orange) or ulcerative colitis (UC; $n = 302$ patient samples, purple) relative to healthy controls ($n = 334$ samples). Fold changes were calculated using pseudo-counts from healthy controls (Supplementary Data 26). Labels indicate effectors with significant prevalence in either Crohn's disease or ulcerative colitis (FDR < 0.01, Fisher's exact test with BH correction). Black asterisks mark statistical significance for individual bars. **f**, Prevalence of indicated effectors in metagenomes of individuals with Crohn's disease ($n = 504$) and ulcerative colitis ($n = 302$) compared to healthy controls (FDR < 0.01, two-sided Fisher's exact test, BH correction). **g**, HuMMI subnetworks showing human proteins (grey) associated with Crohn's disease (orange border) or ulcerative colitis (purple border) interacting with effectors (coloured nodes) from strains enriched and depleted in patient metagenomes. Effector colours indicated in legend. Edges represent protein–protein interactions in HuMMI.

($P = 0.0001$, Fisher's exact test); however, beyond glycerophospholipid metabolism, no metabolic subsystem stood out (Supplementary Data 20). Finally, we compared commensal-targeted functions to those of pathogens (Supplementary Data 20). Some pathways were common to targets of both groups, such as 'NF-κB signalling', whereas

others are specific to commensals including 'collagen biosynthesis' and 'response to muramyl dipeptide'. These findings reinforce the notion of lifestyle-dependent specificity and functional overlap in the molecular interactions of commensals and pathogens with the human host.

We wondered whether perturbations by commensal effectors could influence non-infectious human diseases, starting our analysis at the network level. Genetic variants, but also viruses, contribute to complex diseases by often subtly altering intracellular networks and disease-relevant functions. We first explored whether commensal effectors target proteins that are genetically relevant for diseases and other traits. We used 'causal genes' identified from genome-wide-association studies (GWAS) by the Open Targets initiative[49] to identify the encoded 'disease proteins', and unified traits by the Experimental Factor Ontology (EFO)[50] (Fig. 4b and Supplementary Data 21). The strong enrichment for 'immunoglobulin isotype switching' is intriguing as the evolutionarily older IgA antibodies have important roles in shaping the gut microbiome[51]. Effector targets are also associated with cancers and immune diseases, such as psoriasis, asthma, allergies and systemic lupus erythematosus, although none of these predominantly affect the gut. Given the abundance of immune-related measurement traits, it is possible that the effectors systemically perturb immune signalling and thereby contribute to lung and skin diseases. Alternatively, convergence proteins such as REL or TCF4 (Fig. 2f,h) may also be targeted by local microbiota in skin or lung tissues. Supporting this, 26% of HuMMI effectors are detectable in skin microbiome samples, suggesting that commensal effectors are shared across ecological niches (Supplementary Data 22).

In addition to disease proteins being direct targets, we previously discovered that relevant genetic variation often resides in their protein interaction neighbourhood[13,35]. To explore these, we performed short random walks in the binary human reference interactome (HuRI)[36] and defined 'neighbourhood' as proteins visited significantly more often in HuRI than in degree-preserved randomly rewired control networks. In these neighbourhoods, we assessed disease-protein enrichment using Open Targets, aggregated nominally significant associations at the strain level, and summarized them by disease group (Fig. 4c and Supplementary Data 23). Most disease groups we found are known to be affected by the gut microbiome[3]. Among immune diseases, inflammatory bowel disease (IBD) was enriched (nominal $P = 0.0008$, Fisher's exact test), particularly Crohn's disease (nominal $P = 8.5 \times 10^{-5}$, Fisher's exact test) but not ulcerative colitis (Fig. 4d and Supplementary Data 23). As for direct targets, neighbourhoods also harboured susceptibility for skin and lung diseases such as asthma and psoriasis. Considering the microbiota's relevance for metabolic disorders, effector targeting of neighbourhoods affecting high- and low-density lipoprotein (HDL and LDL, respectively) cholesterol levels (nominal $P = 0.006$ and $P = 0.008$, respectively, Fisher's exact test) and several diabetes-related traits is notable (Supplementary Data 23). Together, these findings suggest that commensal effectors modulate host immune signalling and the local metabolic and structural microenvironment. As the targeted proteins and neighbourhoods are genetically associated with several diseases, modulation of their functions by effectors may contribute to disease aetiology.

### Effector function in human cells and disease

We sought to experimentally verify that commensal effectors perturb some of the identified pathways and functions. We focused on NF-κB signalling, which is central to many diseases and emerged repeatedly in our study. Using a dual-luciferase assay[35] in HEK293 cells (RRID: CVCL_0045, DSMZ), 5 out of 26 commensal effectors significantly activated NF-κB activity in the absence of stimulation (Fig. 5a,b and Supplementary Data 24), while 3 effectors reduced NF-κB activity under strong TNF stimulation (Fig. 5b, Extended Data Fig. 4 and Supplementary Data 24). Next, we assessed whether the effectors modulate NF-κB signalling in unstimulated Caco-2 cells and after pro-inflammatory stimulation (Extended Data Fig. 4). Consistently, Cpa_12, an ABC domain-containing effector, reduced secretion of several cytokines with and without stimulation (Fig. 5c and Supplementary Data 25). Other effectors enhanced cytokine responses, particularly IL-6 and

IL-8, only after Pam3CSK4 stimulation, but not after TNF or flagellin stimulation (Fig. 5d, Extended Data Fig. 4 and Supplementary Data 25). Pam3CSK4 mimics TLR1/2 activation by triacylated lipopeptides abundant in Gram-positive Bacteroidetes, while flagellin mimics TLR5 activation by Gram-negative Pseudomonadota[52]. Thus, commensal effectors exert complex effects on intracellular immune signalling.

Given the genetic and functional links between commensal effectors and IBD, we wondered whether clinical data support a potential role of effectors in the disease. Hypothesizing that causal involvement in IBD aetiology may be reflected in altered effector prevalence, we analysed a metagenome study with over 800 individuals with IBD (504 Crohn's disease, 302 ulcerative colitis) and 334 healthy controls[53]. Focusing on effectors with physical interactions in HuMMI, 64 effectors were significantly more prevalent in individuals with Crohn's disease compared with healthy controls, whereas effectors were less common in individuals with ulcerative colitis (Fig. 5e,f and Supplementary Data 26). These opposing trends were unexpected as Pseudomonadota abundance reportedly increases in both IBDs[46,54]. Some hypotheses for mechanisms underlying this observation emerged from HuMMI: effectors from *K. pneumonia*, *E. coli* and *E. fergusonii*, all highly prevalent in Crohn's disease, interact with the Crohn's disease susceptibility protein COG6, which directly interacts with the ulcerative colitis susceptibility gene *RTP5* (Fig. 5g). Similarly, Efe_13 of *E. fergusonii* binds the Crohn's disease susceptibility protein TNIP1, which functions in NF-κB signalling and interacts with two genes associated with ulcerative colitis. Other enriched effectors show indirect links to Crohn's disease and ulcerative colitis proteins via shared interaction partners (Fig. 5g). While the mechanistic relevance of these interactions requires future studies, these direct and indirect connections to IBD disease proteins invite speculation that they may cause a homeostatic shift that increase the risk for Crohn's disease while decreasing the same for ulcerative colitis.

## Discussion

T3SS are traditionally viewed as hallmarks of pathogenicity, yet in plants and insects they mediate a wider range of functions including beneficial interactions[55]. Our findings extend this observation to the human gut, revealing that T3SS are unexpectedly common among commensal Pseudomonadota. In particular, *E. coli*, which resides close to the intestinal epithelium[56], frequently encodes complete systems. Although not detected in commensal beta- or delta-Pseudomonadota, divergent systems may have escaped current detection tools, thus underestimating their true distribution. Functional assays validated our predictions and revealed regulatory complexity: *C. pasteurii* and *P. massiliensis* showed inconsistent T3SS activation, whereas *E. tarda* reliably injected effectors into human cells. Using *S.* Typhimurium as a heterologous host that robustly initiates the T3SS, we confirmed translocation of 32 effectors from 11 species, indicating that many commensals harbour host-directed secretion capability. The regulatory diversity of T3SS activation is consistent with the idea that, in contrast to pathogens such as *S.* Typhimurium, commensals may require highly specific host or environmental cues to activate secretion. Whether human epithelial or immune cells, akin to plant hosts[57], can actively signal to commensals to induce T3SS, or whether secretion primarily reflects stress responses of potential pathobionts, remains an important area for future investigation.

Interpreting T3SS functionality in the human gut requires moving beyond species-level labels such as 'commensal' or 'pathogenic', which often obscure substantial within-species diversity. As observed in other host kingdoms[58], these categories are fluid: *E. coli* includes both highly pathogenic lineages (for example, EPEC or EHEC) and harmless or beneficial ones, such as the probiotic *E. coli* Nissle 1917[59]. In our analyses, strains isolated from apparently healthy individuals were considered commensals, whereas strains encoding known virulence effectors[28], including *P. aeruginosa* and *Salmonella* spp., were

designated pathogens. Importantly, between these poles lie opportunistic pathogens whose infectious potential emerges only in specific environmental or host-related conditions. A key question is therefore whether commensal T3SS primarily support opportunistic pathogenicity, or whether they have adaptive functions in the non-pathogenic lifestyle. Multiple lines of evidence from our study support the latter.

Comparative sequence, structure and host-target analyses revealed that commensal and pathogen effector repertoires are largely distinct, supporting a model in which commensal T3SS are adapted for cooperative rather than pathogenic interactions. Homotypic clustering of effector structures and depletion of mixed commensal–pathogen clusters suggest that commensal effectors follow separate selective trajectories. The domain analysis supports this, revealing many domains found only in commensal effectors possibly to support a non-pathogenic lifestyle. Notably, numerous effectors involved in cyclic diguanylate synthesis or degradation were identified, often paired with PAS sensor domains suggesting environmentally responsive functions. Intriguingly, several effectors from Gram-negative commensal Pseudomonadota potentiated Pam3CSK4-induced TLR1/2 signalling, suggesting that T3SS may modulate host responses to Gram-positive Bacteroidetes and thereby influence interphyla competition within the gut ecosystem.

Despite substantial divergence in effector structures, commensal and pathogen effectors exhibited both shared and unique host interactions within the meta-interactome. Although these comparisons are limited by the availability of hypothesis-driven interaction datasets for pathogen effectors, the observed patterns parallel findings in plant systems, where effectors from mutualists and pathogens converge on some common host targets while also interacting with proteins critical for distinct outcomes[40]. Across systems, convergence on a subset of host proteins emerges as a signature with biological importance[13,40]. These convergence proteins therefore emerge as key nodes in host–microorganism interactions and understanding their role in commensal versus pathogenic contexts is a promising entry point for understanding how pathogenicity emerges and how balanced immune responses are ensured.

The interaction–structure models provide leads for dissecting effector mechanisms. Targeted host protein domains can indicate which processes an effector may perturb and, when mediated by a corresponding motif, whether the effector may get post-translationally modified, as seen for *H. pylori* CagA[60]. Conversely, post-translational modification of host proteins is a common mechanism of pathogenicity[61]. A manual analysis matching the mimicINT workflow, however, revealed no clear cases in which these engage cognate host substrate motifs and only a single example consistent with known docking specificity. Whether this reflects differences between commensal and pathogen effectors, or the prevalence of functional mimicry without sequence similarity[61], or merely limitations of our approach, remain to be clarified. Post-translational modification of effectors by host enzymes may either enhance effector function or act in host defence, such as by targeting foreign proteins for degradation. The latter would, however, be expected to select against motif retention. Thus, while biochemical directionality from host domain to effector SLiM is plausible, the available evidence suggests that such modifications predominantly support the lifestyle of the injecting bacterium. When commensals act as pathobionts and contribute to non-communicable diseases, such interactions may become intervention targets.

Analysis of host pathways targeted by commensal effectors indeed revealed enrichment for proteins and genetic variation implicated in immune disorders, cancers and metabolic traits. Notably, commensal effectors target network neighbourhoods associated with Crohn's disease but not ulcerative colitis, and physically and functionally interact with key members of the TNF–NF-κB signalling axis. Consistent with these molecular data, T3SS effectors were enriched in the microbiomes of individuals with Crohn's disease while being depleted in ulcerative colitis. This pattern mirrors the differential clinical response to anti-TNF therapy, which is highly effective in Crohn's disease but not in ulcerative colitis. Understanding whether and how commensal effectors directly contribute to Crohn's disease risk or flares, and whether they may even confer protection in ulcerative colitis, are compelling questions for future mechanistic studies with potential therapeutic implications.

Together, our data position host-directed secretion as an underappreciated mode of communication between the human microbiota and its host. By integrating genomic, structural, functional and systems-level analyses, we provide an initial map of the commensal T3SS meta-interactome and establish a framework for exploring its roles across microbial niches, host genotypes and disease states. These findings broaden the conceptual boundaries of T3SS biology and highlight the need to examine secretion systems not only as virulence factors but also as potential modulators of mutualism, competition and host physiology within the human gut.

## Methods

### Identification of T3SS⁺ strains and candidate effectors in culture collections and MAGs

Reference genomes for Pseudomonadota strains isolated by the human microbiome project from human guts and available from DSMZ (via BacDive), ATCC (atcc.org) or BEI (beiresources.org) were identified and cross-referenced with GenBank (release 229), yielding 77 matches, and subjected to T3SS identification, along with 92,143 and 9,367 MAGs, respectively, from two different meta-studies[21,22] that were at least 50% complete and less than 5% contaminated. Prediction performance of EffectiveDB[16] was evaluated by fivefold cross-validation with five repeats using simulated MAGs of 0–100% completeness and 0–50% contamination (in 5% steps) by random sampling genes from the test set. A performance-improved re-implementation of the EffectiveDB classifier (https://github.com/univieCUBE/phenotrex, trained on EggNOG 4.0 annotations[62]) was used with a positive prediction threshold of >0.7.

For 770 T3SS⁺, MAGs protein coding sequences for 474,871 representative proteins were identified using prodigal (v.2.6.3)[63] and CD-HIT (v.4.8.1, parameters: '-c 1.0')[64]. A total of 61,115 proteins were encoded by 44 T3SS⁺ culture collection genomes. Three machine-learning tools (EffectiveT3 v.2.0.1, DeepT3 v.2.0[25] and pEffect[27]) were used to predict T3SS signal or effector homology. Predictions were integrated using a 0–2 scoring scheme: 2 for perfect score (pEffect >90, EffectiveT3 >0.9999, DeepT3: both classifiers positive prediction); 1 for positive prediction at default thresholds (pEffect >50, EffectiveT3 >0.95, DeepT3: one classifier); 0 for negatives. Sequences with a sum score above 4 were regarded as potential effectors. Sequences lacking start/ stop codons or containing transmembrane regions (TMHMM 2.0) were excluded. Proteins were clustered using 90% sequence identity (CD-HIT parameters: '-c 0.9 -s 0.9') to reduce redundancy. Effector clusters with diverse effector-prediction scores were removed (full data in Supplementary Data 1 and 2).

### Cohort analyses

T3SS were analogously predicted for 4,753 strains from the human gastrointestinal bacteria genome collection (HBC)[17], Broad Institute– Open Biome Microbiome Library (BIO–ML) and Global Microbiome Conservancy (GMC)[18,19]. To obtain phylogenetic relationships for T3SS⁺ strains, concatenated bac120 marker proteins from GTDB-Tk (v.2.1)[65] were used. T3SS⁺ genomes were matched to Weizmann Institute of Science representative genomes of the human gut[24] with FastANI v.1.0 using average nucleotide identity (ANI) values >95% (ref. 66). The relative abundance of the 10 matching representatives was identified across 3,096 Israeli and 1,528 Dutch individuals[24].

### Identification of effector similarities and homology groups

Effectors were aligned using the Needleman Wunsch algorithm and were considered 'homologous' in HuMMI_HOM using mutual sequence

identity of ≥30% over 90% of the common sequence length (Supplementary Data 11).

## Commensal vs pathogen effector similarity

Sequences of 1,195 known pathogenic T3 effectors were obtained from BastionHub[28] (29 August 2022), and sequence similarity between commensal and pathogenic effector sequences was assessed using BLAST (v.2.10)[67]. For each commensal effector, the pathogen effector with the highest sequence similarity was considered as the best match and used to calculate alignment coverage. Additional significant similarities were identified using iterative sequence searches against ~124 M non-redundant bacterial sequences from UniRef90 (January 2024) with Jackhmmer[29]. For each commensal effector, we ran five iterations using inclusion and comparison $E$-value thresholds of $<10^{-5}$ (Supplementary Data 5).

## Domain identification and analysis

Protein domain annotation for effectors was carried out using the standalone version of InterProScan (v.5.75-106.0), using Pfam v.37.4 as reference. Amino acid sequences in FASTA format were used as input across three datasets: effector proteins from commensal bacterial strains ($n = 3,002$) and MAGs ($n = 186$), human and vertebrate pathogen effectors obtained from BastionHub, and all reviewed human proteins from the UniProtKB/Swiss-Prot reference proteome. Pathogen effectors were classified on the basis of the host annotation (human, vertebrate) of the corresponding species or strain, as provided by PHI-base[68] and BV-BRC[69]. InterProScan used (translated) protein sequences (Supplementary Data 4) with default parameters. Domain hits with an $E$-value $< 10^{-5}$ were considered significant.

Domains identified as significant in commensal effectors were used as reference for comparative analysis and evaluated for their presence in pathogen effectors and human proteins, applying the same annotation criteria and significance threshold. All domain annotation results, including individual hits across datasets and the comparative summary, are provided in Supplementary Data 5.

## Structural effector similarity

Structures of pathogen and commensal effectors were compared using FoldSeek[70]. Effector structures were downloaded from the AlphaFold DB when available; otherwise, a model with >95% sequence identity and >90% sequence coverage was selected as representative. Clustering was performed by setting bidirectional query coverages (qcov) at 0.5, 0.7 and 0.9, and $E$-value thresholds at 0.001, 0.01 and 1 using FoldSeek Cluster's greedy set cover algorithm. To assess the statistical significance of the obtained cluster distributions, we performed label permutation tests ($n = 10,000$) while keeping the graph's topology intact. The clustering analysis was run for all commensals against three sets of pathogen effectors: all pathogens (895 structures from human, vertebrate and plant pathogens), human and vertebrate pathogens (536 structures) and human pathogen effectors only (488 structures) (Supplementary Data 6).

## Effector cloning

For PCR cloning, genomic DNA or bacterial stocks were obtained from culture collections: ATCC (via LGC Standards, Wesel, Germany, or ATCC, Manassas, VA, USA), DSMZ (Leibniz Institute DSMZ, Braunschweig, Germany) and BEI resources (Manassas, VA, USA) (Supplementary Data 9). Live strains were cultured according to supplier protocols and DNA was extracted using the NucleosSpin Plasmid mini kit. Effectors were cloned into pENTR223.1 by nested PCR to add Sfi sites and by restriction enzyme-based cloning using standard protocols, and verified by Sanger sequencing. Effectors identified from MAGs and effectors for the PRS were synthesized by Twist Bioscience. For experiments, effectors were moved into pDEST-DB (pPC97, Cen origin), the pDEST-N2H-N1 and -N2 and pMH-Flag-HA by Gateway LR reactions.

For bacterial injection assays, effector ORFs were cloned into a modified bacterial expression plasmid based on the pEYFP backbone (BD Biosciences, 6004-1). The EYFP sequence (positions 217–1,407) was removed and replaced with (1) SfiI and XbaI restriction sites for directional cloning of effector ORFs, (2) a 3× Flag epitope tag, (3) the HiBiT tag coding sequence VSGWRLFKKIS (Promega), and (4) the $E.$ $coli$ rrnB transcriptional terminator (pLac_FL_HiBiT). PCR-amplified effectors were ligated into pLac_FL_HiBiT at SfiI and XbaI restriction sites (primers in Supplementary Data 7). The positive control SipA was amplified from a pT10-based plasmid (pMIB6433)[34]. Cloning was verified by analytical PCR.

## Electroporation of plasmids into bacterial strains for injection experiments

Electrocompetent *S. enterica* sv. Typhimurium (wild-type SB300 and $\Delta sctV$ mutant SB1751)[34], *E. tarda* (ATCC 23685), *C. pasteurii* (DSM 28879) and *P. massiliensis* (DSM 26120) were generated in-house and electroporated with effector encoding plasmids using a Gene Pulser Xcell Electroporation System (Bio-Rad) at 2.5 kV and 200 Ω for ~5 ms. Transformed strains were cultured overnight in LB medium with ampicillin for subsequent use in injection assays.

## Injection assay

The injection assay was adapted from ref. 34. HeLa cells stably expressing LgBiT (HeLa-LgBiT) were grown using standard conditions (DMEM, 10% fetal bovine serum (FBS), 37 °C, 5% $CO_2$) for 24 h before infection. *S.* Typhimurium strains carrying pLac_FL_HiBiT effector constructs were cultured overnight in LB supplemented with 0.3 M NaCl and ampicillin. *Edwardsiella tarda*, *C. pasteurii* and *P. massiliensis* strains were cultured with 200 µM IPTG to induce effector expression. Overnight bacterial cultures were added to the HeLa-LgBiT cells at a multiplicity of infection of 50 and jointly incubated for 1 h (*Salmonella*) or 1.5 h. After media replacement, extracellular luminescence was quenched by addition of 1× DarkBiT peptide (Promega, CS3002A02) for 50 min. Luminescence was measured after addition of 25 µl fresh Nano-Glo reagent (Promega, N2012) using a SpectraMax ID3 microplate reader (1,000 ms). Each strain was tested with four technical replicates and five biological replicates performed on separate days. Luminescence values from technical replicates were averaged to obtain a single value for each biological replicate. Luminescence fold-change was calculated by dividing the average signal from the effector-expressing strain by that of the mock control separately for wild-type and ΔsctV strains. To assess effector translocation, fold-change values were statistically compared between wild-type and mutant strains for *Salmonella* and against the negative control (SipA in ΔsctV) for *E. tarda*, using Wilcoxon rank-sum test (Supplementary Data 10).

## Western blot analysis for injection assay and co-immunoprecipitations

Proteins were separated by 10% or 15% SDS–PAGE, transferred to PVDF membranes (Bio-Rad, 1620177) and blocked in blocking solution (5% non-fat dry milk in 1× PBS) for 1 h at room temperature or overnight at 4 °C. Blots were done with mouse anti-Flag M2 monoclonal antibody (Sigma-Aldrich, F1804, 1:5,000), rabbit anti-Myc (Abcam, ab9106, 1:1,000), followed by HRP-conjugated secondary antibody (Santa Cruz Biotechnology; anti-mouse: sc-516102; anti-rabbit: sc-2357, both 1:5,000) for 1 h each with three washes with blocking solution or PBST, respectively. Signal was detected with SuperSignal West Femto Substrate (Thermo Scientific, 34094) according to manufacturer instructions. Blots were imaged using the Intas ChemoStar imaging system.

## Meta-interactome mapping

A multi-assay interactome mapping pipeline was used[37] (Extended Data Fig. 2). In the initial screening by Y2H, candidate effectors fused to the Gal4 DNA-binding domain (DB-X) were screened against 17,472

human proteins fused to the Gal4 activation domain (AD-Y). Before screening, DB-X ORFs were tested for autoactivation by mating against AD-empty plasmids. Autoactivators were excluded. In the primary screen, DB-X strains in Y8930 (MATα) were mated on yeast extract peptone dextrose (YEPD) agar (1%) plates against minipools of ~188 AD-Y in Y8800 (MATa) representing the human ORFeome collection (v.9.1)[36]. After 24 h, yeasts were replica-plated onto selective media lacking leucine, tryptophan and histidine (SC-Leu-Trp-His), containing 1 mM 3-AT (3-amino-1,2,4-triazole) (3-AT plates) and replica-cleaned after 24 h. After 48 h, colonies were picked and then grown for 72 h in SC-Leu-Trp liquid medium for secondary phenotyping using the same selective +SC-Leu-His + 1 mM 3-AT + 1 mg l⁻¹ cycloheximide plates to identify spontaneous autoactivators. Clones growing on 3-AT plates but not on cycloheximide plates were processed for sequence identification using a modified Kilo-seq procedure[35]: ORFs were amplified and tagged by PCR using a universal 'term' reverse primer (5′-GGAGACTTGACCAAACCTCTGGC) and Gal4-AD and -DB specific forward primers with position barcodes (Supplementary Table 11) and a TruSeq P7 sequence (0.2 µl DreamTaq DNA polymerase (ThermoFisher, EP0702), 3 µl 2 µM term primer, 3 µl forward primer, 2 µl yeast lysis). For every 96-well plate, 5 µl from each well were pooled, purified with 24 µl magnetic beads (magtivio, MDKT00010075) and eluted in 25 µl TE buffer. The DNA concentration of each pool was quantified by the QuantiT PicoGreen dsDNA Assay kit (ThermoFisher, P7589) using a lambda DNA dilution series (50–0.390625 ng µl⁻¹), then diluted to 1–2 ng µl⁻¹ and tagmented with 0.25 µl TDE enzyme (Illumina Tagment DNA TDE1 Enzyme and Buffer kit, 20034197). A second PCR added plate-specific Nextera i5/i7 indices (Supplementary Table 11) (8 µl tagmented DNA, 0.2 µl DreamTaq (ThermoFisher, EP0702), 1 µl 10 µM i5/i7 primers), followed by bead cleanup (80 µl beads per 100 µl PCR, eluted in 30 µl). Libraries were sequenced on a MiSeq v.2 kit (Illumina, MS-102-2002) and demultiplexed with bcl2fastq2 (v.2.20.0.422) by Illumina.

Finally, haploid yeasts of the DB-X and AD-Y candidate interaction pairs were mated individually and tested four times on selective plates using empty AD and DB plasmids as negative controls. Growth scoring was performed using a custom dilated convolutional neural network[35]. Pairs scoring positive in at least three out of four repeats qualified as bona fide Y2H interactors. The AD-Y and DB-X constructs were again identified by Illumina sequencing. All interaction data are provided in Supplementary Data 11.

### Assembling reference sets for quality control
To identify reliably documented interactions between bacterial effectors and human proteins for our control set, we queried the IMEx consortium protein interaction databases[71] for pairs supported by multiple evidence and at least one experiment detecting direct interactions. We manually recurated the corresponding publications and identified 67 well-documented direct interactions between 29 T3 effectors and 64 human proteins, described in 38 distinct publications constituting bhLit_BM-v1. To assemble bhRRS-v1, we randomly paired T3 effectors from bhLit_BM-v1 with human proteins in HuRI (Supplementary Data 12). Effector ORFs were cloned into Entry and experimental plasmids as described above. Human hsPRS/RRS-v2 ORFs were taken from hORFeome9.1 (ref. 36) and verified by end-read Sanger sequencing.

### Interactome validation by yN2H
yN2H was used to independently validate the quality of the HuMMI dataset[38]. A total of 200 interaction pairs were randomly picked from HuMMI; all ORFs (Supplementary Data 14) were transferred by Gateway LR reactions into pDEST-N2H-N1 and pDEST-N2H-N2, and transformed into haploid Saccharomyces cerevisiae Y8800 (MATa) and Y8930 (MATα) strains. Protein pairs from all datasets were randomly distributed across matching 96-well plates. Luminescence from reconstituted NanoLuc for each sample was measured on a SpectraMax ID3 (Molecular Devices) with a 2-s integration time. The normalized

luminescence ratio (NLR) was calculated by dividing the raw luminescence of each pair (N1-X N2-Y) by the maximum luminescence value of one of the two background measurements. All obtained NLR values were log₂ transformed and the positive fraction for each dataset was determined at log₂ NLR thresholds between −2 and 2, in 0.01 increments. Statistical results were robust across a wide range of stringency thresholds. Supplementary Data 14 reports the results at log₂NLR = 0. Reported P values were calculated by Fisher's exact test.

### Co-immunoprecipitation of selected effector–host interactions
We evaluated whether N-terminally Flag-HA-tagged effector, or negative control Flag-GFP, co-immunoprecipitated the human proteins carrying an N-terminal MYC tag. Transfections for test and control pairs were always processed in parallel. HEK293 cells (RRID: CVCL_0045, DSMZ) were seeded in 10-cm dishes at a density yielding 60–70% confluency on the day of transfection. Plasmid DNA and X-tremeGENE transfection reagent (Roche) were mixed at a ratio of 1:2 (µg DNA:µl reagent) in serum-free DMEM. Per dish, 10 µg plasmid DNA (consisting of 3 µg effector- or GFP-encoding plasmid, 3 µg plasmid encoding the human protein and 4 µg empty vector) was diluted in 500 µl serum-free medium, followed by the addition of 20 µl X-tremeGENE reagent. The mixture was inverted twice, incubated for 15 min at room temperature and then added dropwise to the culture dish containing cells in complete growth medium. Cells were incubated under standard culture conditions (37 °C, 5% CO₂) for 24 h before downstream analysis.

For cell lysate preparation, all steps were performed on ice. Culture medium was aspirated, and cells were washed three times with ice-cold 1× PBS by rinsing and aspirating sequentially. Cells were lysed directly on the plate by adding 1 ml NP-40 lysis buffer per plate (50 mM Tris-HCl, pH 8.0, 150 mM NaCl, 1% (v/v) NP-40 and 2.5 mM EDTA, with Roche complete protease inhibitor). Cells were detached using a rubber policeman and transferred to a 1-ml centrifuge tube. Lysates were incubated on ice for 30 min and cleared by centrifugation at 30,000g for 15 min at 4 °C. The supernatant was collected and the protein concentration was measured using the Bradford assay (Bio-Rad); the lysate was immediately used.

For immunoprecipitation experiments, 1 mg of cleared lysates of each sample was diluted into a final volume of 750 µl, and then 50 µl of an NP-40 buffer equilibrated with 20% anti-Flag M2 affinity gel (Sigma-Aldrich, A2220) slurry was added. Samples were rotated at 4 °C for 1 h. For washing, the tube was centrifuged at maximum speed for 30 s, the supernatant aspirated and 1 ml NP-40 wash buffer added, followed by a brief inversion. After three washes, the beads were resuspended in 50 µl NP-40 buffer, 50 µl Laemmli loading buffer was added, and the beads were heated at 98 °C for 10 min and briefly centrifuged before analysis. For analysis, 10 µl of cleared lysates and 15 µl of all immunoprecipitates were loaded on SDS–PAGE and processed through western blots as described above.

### Interactome framework parameter calculation
The completeness of an interactome map is an important parameter that enables assessment of overlap and how complete a given biology is covered by the map. The framework incorporates assay sensitivity (that is, the proportion of interactions the assay can detect), sampling sensitivity (that is, saturation of the screen) and search space, describing all pairwise protein combinations. For the meta-interactome studied here, the search space cannot reasonably be estimated due to the uncertainty of T3SS-containing microorganisms in all human guts and the resulting inability to define that dimension of the problem.

Assay sensitivity ($S_a$) was assessed using the effector bh_LitBM-v1 (54 pairs) and bhRRS-v1 (72 pairs) as well as the human hsPRS/RRS-v2 (60 and 78 pairs, respectively) for benchmarking. All reference sets were tested four times using the Y2H screening pipeline (Supplementary Data 13). To assess sampling sensitivity ($S_s$), a repeat screen was

conducted. A total of 288 bacterial effectors were screened 4 times against 5 pools comprising 1,475 human proteins. A saturation curve was calculated as described[37]. In brief, all combinations of the number of interactions of the four repeats were assembled and the reciprocal values calculated. From these, a linear regression was determined to obtain the slope and the intercept. Reciprocal parameters were calculated and the Michaelis Menten equation was used with modified variables: analogous to increasing substrate concentrations in enzyme reactions, repeat screens progressively drive the screen to saturation[91]. Hence a saturation curve was predicted using $Ni(R) = Ni_{max} \times R/K_m + R$, with Ni representing the interaction count after $R$ repeats, $Ni_{max}$ the saturation limit and $K_m$ the Michaelis constant. Overall sensitivity emerges from both sampling and assay limitations and was calculated as $S_o = S_a \times S_s$.

### Intra- and interspecies effector convergence

To estimate the significance of effector convergence, we performed a permutation test by randomly sampling with replacement 979 target nodes from HuRI[36] ($n = 8,274$). In each iteration, we counted the number of unique targets, and the distribution from 10,000 random permutations was used to compute the $z$-score for the observed 349 targets. A $P$ value was obtained from the $z$-scores using the 'pnorm()' R command and multiplied by 2 for a two-tailed test. To avoid overestimation and increase stringency, we restricted the analysis to Y2H positive proteins in HuMMI$_{MAIN}$ and HuRI. To assess interspecies convergence, we used a conditional permutation test that preserves the strain contribution. Each iteration generated 18 samples corresponding to the observed number of targets for each strain (Supplementary Data 11). For every protein, the frequency of selection across all strains was recorded as its convergence value. On the basis of 10,000 iterations, we derived the convergence value distribution, calculated $z$-scores and obtained the $P$ value using the pnorm() R function. Significance was observed from four strains onward ($P < 0.004$), and proteins targeted by at least four strains were considered to show interspecies convergence.

### Sequence similarity and interaction profile

To investigate the relationship between the effector sequence and the interaction profile similarity, we calculated the pairwise Jaccard indices for all effector pairs within each homology cluster. The index was defined as the ratio of shared to total human targets. Pairs with fewer than three targets were excluded.

### AlphaFold-based interaction modelling

To analyse the interfaces of effector–host interaction pairs, all identified pairs were subjected to structural prediction using AlphaFold v.2.3.1 with the following options: –model_preset=multimer, –db_preset=full_dbs, –max_template_date=2023-12-19, –num_multimer_predictions_per_model=1, –enable_cpu_relax and –use_precomputed_msas. Predictions were not generated for pairs whose combined length exceeded 2,500 residues. The predicted aligned error (PAE) matrix was extracted from the AlphaFold pickle output using alphapickle v.1.4.1 (https://github.com/mattarnoldbio/alphapickle, https://doi.org/10.5281/zenodo.5752375). To assess confidence, we use the confident contacts count (CCC), which is the number of residue–residue contacts[72] with PAE < 4 Å. Each putative interface residue was assigned a PAE value. When a residue was in contact with multiple residues on the partner protein, the minimum PAE value among those contacts was used. Structure predictions were considered confident when the CCC was ≥5.

### Interface similarity analysis using PAE thresholding

Protein sequences (Supplementary Data 17) were converted from single-letter aa notation to three-letter residue annotation, and residue identifiers were assigned to match their positions in the AlphaFold PAE matrix. Only human proteins targeted by at least two bacterial effectors were retained. Residue contacts were extracted and matched to PAE coordinates, and pooled PAE values defined the 25th, 50th, 75th and 95th percentile thresholds. Contacts with PAE values equal or below the threshold were retained, and the corresponding human and bacterial residues and total retained contacts were recorded. This procedure was repeated for the 25th, 50th, 75th and 95th percentiles, and the resulting subsets were merged into the main dataset.

### Interface similarities

Interface similarity between bacterial effectors targeting the same human protein was assessed using the Jaccard index across all PAE thresholds. For each targeted human protein, all interacting bacterial effectors were identified, and all possible effector–effector combinations were generated. At each threshold, the Jaccard index was calculated as the number of overlapping human interface residues divided by the total number of unique residues in both interfaces. Indices were classified as distinct (Jaccard index ≤ 0.1), overlapping (0.1 < Jaccard index < 0.6) or same (Jaccard index ≥ 0.6). Analogous calculations were performed to analyse interfaces of human proteins targeted by the same bacterial effector.

### Interface domain annotations

Domains were assigned to the interacting human proteins using Inter-ProScan v.5.75 with InterPro release 106.0, run through the EBI web server. Domain coordinates, descriptions and confidence scores were retrieved. The number of interface residues within each domain boundary (n_interface_residues_in_domain) was then counted, along with the total residues in the predicted interface (n_residues_in_interface), the percentage of interface residues in the domain (IF%), the number of residues in the domain (Domain_length) and the proportion of the domain length relative to the full protein length (Domain%).

### SLiM–domain interface predictions

We used as mimicINT[43] input, a representative set of effectors identified in isolated strains (2,300 sequences clustered at 90% identity) and all effectors identified in MAGs (186). mimicINT detects domains in effector sequences using the signatures from the InterPro v.81.0 database[73], retaining matches with an $E$-value < $10^{-5}$. For motif detection, mimicINT uses definitions available in the ELM database[74]. The IUPred 1.0 algorithm[75] was employed to detect motifs in disordered regions with both short and long models (motif disorder propensity = 0.2 (ref. [76]), minimum size = 5). The interface inference step used the 3did database[77] for domain–domain templates and the ELM database (2022 release) for motif–domain templates. Two scoring strategies were applied. First, domain binding specificity within the same family was accounted for by computing a profile HMM-based domain score[41] (stringency threshold = 0.3). Second, given the degenerate nature of motifs[41], mimicINT uses Monte Carlo simulations to estimate the probability of a SLiM occurring by chance, by shuffling disordered regions of the input sequences to generate $N$ randomized proteins. Effectors were first grouped by strain, with MAG-derived effectors assigned to the closest strain. Disordered regions were shuffled 100,000 times using two backgrounds: same-strain effectors (within-strain shuffling) and full effector set (interstrain shuffling). Motif occurrences in each effector were compared to those in the shuffled sequences, retaining only those with an empirical $P < 0.1$ in both backgrounds. To assess whether the number of inferred interface-resolved interactions exceeded random expectation, the analysis was controlled using 10,000 degree-controlled random networks generated from the human interaction search space (Supplementary Data 18).

For the reverse analysis of bacterial domains interacting with SLiMs in the human proteins, the annotated bacterial domains were matched to domains in the ELM templates. For interactors of the so-identified effectors (Efe_1, Pfa_18, Pre_16, Pst_8, Vfu_32), we identified disordered regions as above and screened these for motifs matching the templates in the ELM database, yielding the reported example.

## Holdup assay

Holdup is a biochemical assay used to validate the interface predictions involving PDZ domains. A total of 54 human PDZ domains and 11 tandem constructs were recombinantly expressed as His$_6$-MBP-PDZ constructs in *E. coli* BL21(DE3) pLysS and purified by Ni$^{2+}$-affinity columns using 800 µl of beads (Chelating Sepharose Fast Flow immobilized metal affinity chromatography, Cytiva) per target. After elution, purified proteins were desalted using PD10 columns (GE healthcare, 17085101) into 3.5 ml 50 mM Tris (pH 8.0), 300 mM NaCl and 10 mM imidazole buffer. Protein concentrations were determined using $A_{280\,nm}$ on a PHERAstar FSX plate reader (BMG LABTECH), and purity assessed by SDS–PAGE and capillary electrophoresis; 4 µM stocks were stored at −20 °C. Biotinylated peptides (10-mer) corresponding to the C-terminal sequences of effectors were synthesized by GenicBio Limited; the N-terminal biotin was attached via a 6-aminohexanoic acid linker, and all peptides were >95% pure (HPLC and MS). Peptides were solubilized in dH$_2$O, 1.4% ammonia or 5% acetic acid, aliquoted at 10 mM and stored at −20 °C.

For the assay, 2.5 µl of streptavidin resin (Cytiva, 17511301) was incubated in a 384-well filter plate (Millipore, MZHVN0W10) for 15 min with 20 µl of a 42 µM peptide solution. The resin was washed with 10 resin volumes (resvol) of holdup buffer (50 mM Tris-HCl, 300 mM NaCl, pH 8.0, 10 mM imidazole, 5 mM dithiothreitol), incubated for 15 min with 5 resvol 1 mM biotin and washed three times with 10 resvol of holdup buffer. Individual PDZ domains were added to wells, incubated for 15 min, and unbound PDZ recovered by centrifugation into 384-well black assay plates for fluorescence readout. Concentrations were quantified by intrinsic Trp fluorescence, and fluorescein/mCherry was used for peak normalization. Binding affinities and equilibrium dissociation constants were calculated as previously described[45], using the mean PBM concentration. Raw values and statistical analysis are provided in Supplementary Data 19.

## Function enrichment analysis

Functional enrichment of effector targets was assessed using the 'gost()' function in the 'gprofiler2' R package (v.0.2.1)[78] with HuRI as the background (custom_bg), excluding electronic annotations (exclude_iea = TRUE), with Benjamini–Hochberg correction (correction_method = 'fdr'). Functional categories were drawn from Gene Ontology biological process terms (GO:BP), Kyoto Encyclopedia of Genes and Genomes (KEGG) pathways, and the Reactome pathways database (sources = c('GO:BP', 'KEGG', ' REAC')). Odds ratios and fold enrichments were calculated to estimate effect sizes, where the odds ratios was the ratio of odds in the target set to those in the HuRI background, and fold enrichment compared observed to expected annotated targets. Expected values were based on random sampling from the HuRI background (GO:BP = 6,988; KEGG = 3,250; Reactome = 4,592) (Supplementary Data 20). Similar analyses were performed for functional enrichment analysis of human proteins targeted by pathogens (Supplementary Data 20).

## Metabolic subsystem analysis

We assessed enrichment of targeted enzymes across metabolic subsystems using the human genome-scale model Recon3D[48]. Recon3D is a curated static model of human metabolism that lacks post-translational and allosteric regulation. Ligases and kinases were excluded to focus on metabolic enzymes. For each of the 95 Recon3D subsystems, enrichment was tested using the 'phyper()' R function, with inputs corresponding to annotated and unannotated targeted enzymes and BH false discovery rate (FDR) correction. OR and fold enrichment were calculated as described for functional analyses (Supplementary Data 20).

## Disease enrichment analysis

Associations of effector targets and convergence proteins with human disease genetics were tested using a two-sided Fisher's exact test. Disease-causal genes were obtained from the Open Targets genetic portal (access date 23 August 2022), which integrates variant-to-gene distance, quantitative trait loci co-localization, chromatin interactions and variant pathogenicity[79]. The portal's machine-learning model assigns each locus-to-gene (L2G) score to genes in loci identified in GWAS to identify the most probable causal gene. Genes with L2G ≥ 0.5 were considered causal as recommended[80]. Ensembl identifiers were converted to gene symbols using the biomaRt R package (v.2.60.1, Bioconductor 3.19), and Fisher's exact test was implemented in R (fisher.test), stats v.4.2.2 using default parameters on 2 × 2 contingency tables comparing causal gene presence in query and background sets. HuRI protein encoding genes were used as the background, and targets or convergence proteins as the query sets. FDR correction and OR and fold enrichments were calculated as done for functional enrichment (Supplementary Data 21).

## Random walk-based determination of commensal effector network neighbourhoods

We implemented a random walk with restart (RWR) algorithm, RWR-MH[81], to explore the network neighbourhood of 338 human proteins targeted by 243 commensal effectors in HuRI[36] (HuMMI-$_{MAIN}$). Human targets were used as seeds, with the restart probability of 0.7 generating a ranked list of proteins. Statistical significance was assessed by random walks in degree-preserved randomized networks. We generated 1,000 random networks from HuRI and computed RWR scores for each protein, retaining as network neighbour only those with empirical $P < 0.01$.

For each set of significant neighbourhood proteins, we tested for enrichment of Open Targets causal genes (L2G ≥ 0.5) linked to traits supported by at least three causal genes. Enrichment in each strain neighbourhood was assessed using two-sided Fisher's exact test with BH correction. No associations were significant (FDR < 0.05). We therefore focused on 400 associations with nominal $P < 0.01$ and odds ratio > 3. Disease categorizations were refined to reflect aetiology; Sjogren syndrome, eczema and psoriasis were grouped as immunological rather than eye or skin traits, and osteoarthritis as musculoskeletal/connective tissue rather than metabolic traits. For Fig. 4d, related asthma and psoriasis terms were merged (Supplementary Data 23).

## NF-κB activation assay

HEK293 cells (RRID: CVCL_0045, DSMZ) were maintained in DMEM, 10% FBS, 100 U ml$^{-1}$ penicillin–streptomycin at 37 °C and 5% CO$_2$. IKKβ (pRK5-Flag) and A20 (pEF4-Flag) served as positive and negative controls, respectively. Cells (1 × 10$^6$ per 60 mm dish) were transfected with 10 ng NF-κB reporter plasmid (6× NF-κB firefly luciferase pGL2), 50 ng pTK reporter (*Renilla* luciferase) and 2 µg bacterial ORF in pMH-Flag-HA using the calcium phosphate method. After 6 h, medium was replaced. To assess NF-κB inhibition, cells were treated for 4 h with 20 ng ml$^{-1}$ TNF (Sigma-Aldrich, SRP3177) at 24 h post transfection. Lysates were analysed using the dual-luciferase reporter kit (Promega, E1980) with a luminometer (Berthold Centro LB960 microplate reader, software: MikroWin 2010). NF-κB induction was determined as firefly/*Renilla* luminescence. $P$ values were calculated using Kruskal–Wallis test with Dunn's post hoc comparisons followed by FDR correction. Raw values and statistical analysis are provided in Supplementary Data 24.

Protein expression was analysed by western blot as described above with following modifications: blocking solution contained 0.1% Tween-20. Membranes were incubated overnight at 4 °C with primary antibodies in 2.5% BSA in PBST, washed and probed with anti-mouse secondary antibody in PBST for 1 h at room temperature (1:10,000; Jackson ImmunoResearch Labs, RRID:AB_2340770). Primary antibodies used were: anti-β-actin (1:10,000; Santa Cruz Biotechnology, RRID:AB_626632), anti-Flag M2 (1:500; Sigma-Aldrich, RRID:AB_259529) and anti-HA (1:1,000; Sigma-Aldrich, RRID:AB_514505). Signals were detected using LumiGlo reagent (CST, 7003S) and chemiluminescence film (Sigma-Aldrich, GE28-9068-36).

## Cytokine assays

Caco-2 cells (RRID: CVCL_0025) were maintained in DMEM glutamax medium (Gibco) with 10% FBS and 1% Pen/Strep at 37 °C and 5% $CO_2$. Experiments in Fig. 5c were performed by transfecting Caco-2 cells using 40,000 MW linear polyethylenimine (PEI MAX) (Polysciences) at a ratio of 1:5 pDNA:PEI. Cells were exposed to the transfection mixture for 16 h, washed, recovered for 6 h and then sorted (BD FAC-SAria III cell sorter, BD Biosciences). After 24 h recovery, cells were activated for 48 h using a stimulation mix containing 200 ng ml$^{-1}$ phorbol-12-myristate-13-acetate (P8139, Sigma-Aldrich), 100 ng ml$^{-1}$ lipopolysaccharide (L6529, Sigma-Aldrich) and 100 ng ml$^{-1}$ TNF (130-094-014, Miltenyi Biotec). During activation, proliferation was monitored in the Incucyte S3 Live Cell Analysis system (Essen BioScience). Cytokine levels were determined using the human inflammation panel 1 LEGENDplex kit (Biolegend). We performed three biological repeats, each with three or four technical repeats. Statistical significance was tested on the average of the technical replicates using Kruskal–Wallis test with Dunn's post hoc comparisons. Experiments in Fig. 5d and Extended Data Fig. 4d were performed by transfecting cells using the 4D-Nucleofector system (Lonza). Collected cells were resuspended in SF nucleofector solution, added with 0.6 µg plasmid, and pulsed (code DG-113) and plated in DMEM + 5% FBS. Cells were allowed to recover overnight and then rested in culture medium for 24 h. Cells were stimulated with 10 µg ml$^{-1}$ Pam3CSK4 (tlrl-pms, Invivogen), 1 µg ml$^{-1}$ flagellin (tlrl-stfla, Invivogen) or 100 ng ml$^{-1}$ TNF (130-094-014, Miltenyi Biotec) for 24 h. We performed five biological repeat experiments with three technical repeats each. For each experiment, pooled supernatants were analysed using the Human Anti-virus Response Panel V02 (BioLegend). The data were analysed using Kruskal–Wallis test with Dunn's post hoc comparisons. Raw and statistical summary data are available in Supplementary Data 25.

## Protein ecology on IBD metagenomes

Metagenomic assemblies from the Inflammatory Bowel Disease Multiomics DataBases (IBDMBD)[53] and from the skin metagenome[82] were downloaded, and protein repertoires predicted using Prodigal (option: -p meta)[83]. Effectors were compared to the metagenomic protein repertoires using DIAMOND 0.9.24 (options: >90% query length, >80% identity). For analyses in Fig. 5, samples were grouped into individuals with ulcerative colitis ($n = 304$), Crohn's disease ($n = 508$) and the controls without IBD ($n = 334$). Binary presence and absence vectors for each effector across the sample were generated and the prevalence of each effector in patients compared to the controls was assessed using Fisher's exact test, implemented within the SciPy 1.9.3 Python 3.10.12 module, and FDR corrected using BH correction. Differences in prevalence distributions between healthy and either patient cohort were estimated using the Wilcoxon rank-sum test, implemented in the 'wilcox.test()' R function. We used fold change as the measure of effect size in Fig. 5e, calculated as prevalence in the test group divided by prevalence in the healthy group. To avoid division by zero, we applied a small pseudo-count to the healthy cohort data for individuals with 0% prevalence. The pseudo-count was equivalent to half a case in the healthy cohort ($n = 334$ individuals), ensuring minimal influence on results while enabling calculation of fold change. Statistical details are provided in Supplementary Data 26.

## Statistics and reproducibility

Data were subjected to statistical analysis and plotted in Microsoft Excel 2010 or Python or R scripts. For comparison of normally distributed values, we used one-way analysis of variance (ANOVA). For comparison of values not passing the normality tests, we used either Kruskal–Wallis test with Dunn's correction for multiple-group comparisons or Wilcoxon rank-sum test for two-group comparisons as indicated. Enrichments were calculated using Fisher's exact test with Bonferroni FDR correction. All statistical evaluations were done as two-sided tests. Generally, a

corrected $P < 0.05$ was considered significant. GO, KEGG and Reactome functional enrichments were calculated using the gprofiler2 R package with the indicated background sets. For the disease target enrichments and neighbourhood associations, no associations were significant after multiple hypothesis correction, which is why nominally significant associations calculated by Fisher's exact tests were used for Fig. 4c,d. All raw values, $n$ and statistical details are presented in Supplementary Data as indicated in figure legends and in Methods.

### Reporting summary

Further information on research design is available in the Nature Portfolio Reporting Summary linked to this article.

## Data availability

All sequence, interaction and functional data generated in this study are available as supplementary information. The effectors identified and cloned for interactome mapping are presented in Supplementary Data 7. All protein–protein interaction data can be found in Supplementary Data 11. The data for functional validation assays can be found in Supplementary Data 24–26. All protein interaction data have been deposited to the IMEx consortium (http://www.imexconsortium.org) through IntAct and assigned the identifier IM-29849. New effector sequences have been submitted to GenBank: BankIt2727690: OR372873–OR373035 and OR509516–OR509528. Source data are provided with this paper.

## Code availability

Data and scripts related to the prediction of T3SS-positive reference strains and metagenomes are available on Zenodo at https://doi.org/10.5281/zenodo.17825584 (ref. 84). All data and scripts generated to perform the structural comparison between commensal and pathogen effectors are available on Zenodo at https://doi.org/10.5281/zenodo.11951539 (ref. 85). The full set of inferred SLiM–domain and domain–domain interactions and the randomly generated networks are available on Zenodo at https://doi.org/10.5281/zenodo.11400863 (ref. 86). The mimicINT[43] code can be found on GitHub at https://github.com/TAGC-NetworkBiology/mimicINT/releases/tag/v1 (ref. 87). The 1,000 randomized control networks for the random walk analysis are available on Zenodo at https://doi.org/10.5281/zenodo.12743976 (ref. 88). The AlphaFold predictions of effector–host interaction pairs along with all confidently predicted structures are available on Zenodo at https://doi.org/10.5281/zenodo.16816224 (ref. 89). The datasets and analysis scripts for convergence analysis, functional and disease enrichment analysis, and AlphaFold human–effector interface similarity analysis are available on Zenodo at https://doi.org/10.5281/zenodo.16883544 (ref. 90).

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

## Acknowledgements

This work was supported by HDHL-INTIMIC 'Interrelation of the Intestinal Microbiome, Diet and Health' (BMBF 01EA1803 to P.F.-B., ANR ANR-17-HDIM-0001 to C.B. and FFG 11819559 to T.R.), the European Union's Horizon 2020 Research and Innovation Programme (Project ID 101003633, RiPCoN; P.F.-B., C.B.); the Free State of Bavaria's AI for Therapy (AI4T) Initiative through the Institute of AI for Drug Discovery (AID) (P.F.-B.), the French government under the France 2030 investment plan, as part of the initiative d'Excellence d'Aix-Marseille Université – A*MIDEX (AMX-21-PEP-043, to A.Z.), and the FRS-FNRS (J.-C.T. and S.B.M.). The computational results presented were achieved in part using the Vienna Scientific Cluster (VSC). Centre de Calcul Intensif d'Aix-Marseille is acknowledged for granting access to its high-performance computing resources. S.A.C. received funding from the 'Espoirs de la recherche' programme managed by the French Fondation pour la Recherche Médicale (FRM, FDT202106013072). The project leading to this publication also received funding from France 2030, the French Government programme managed by the French National Research Agency (ANR-16-CONV-0001) and from the Excellence Initiative of Aix-Marseille University – AMIDEX. J.F.-M. was funded by Consejo Nacional de Humanidades Ciencias y Tecnologias

(CONAHCYT) Becas al Extranjero Convenios GOBIERNO FRANCES 2021 – 1 grant 795494, and received support from the 'Espoirs de la Recherche' programme managed by the French Fondation pour la Recherche Medicale (FRM, FDT202404018637). The AFMB contribution was supported by the French Infrastructure for Integrated Structural Biology (FRISBI) ANR-10-INSB-05-01. D. Krappmann was supported by Deutsche Forschungsgemeinschaft (ID 210592381 – SFB 1054 A04). T.C. was supported by the Deutsche Forschungsgemeinschaft (project 403224013 – SFB 1382 Q02). S.R. was supported by an ERS Long-Term Research Fellowship (LTRF2024-01131).

## Author contributions

P.F.-B. conceived the project. P.H., T.R., T.C.A.H., S.A., C.B., A.Z. and P.F.-B. performed T3SS and effector identification. V.Y., H.H., B.W., S.T.R., M.R., M.A., A.S. and P.F.-B. performed ORF cloning. J.F.-M., L.B. and A.Z. conducted structural pathogen–commensal comparison. H.H., B.W., S.T.R., Y.M.T., L.P., B.D. and P.F.-B. conducted injection assays. V.Y., S.T.R., B.W., P.S., A.S. and P.F.-B. performedY2H analyses. V.Y., M.A., S.A., M. Boujeant, A.Z., C.F. and P.F.-B. curated bhLit-BM data. B.D., D.S., V.Y., J.F.-M., L.L., L.B., J.P., C.-W.L., M.H., C.B., A.Z. and P.F.-B. performed data analyses. B.W., S.T.R. and V.Y. conducted yN2H validation assays. S.A.C., L.B., J.P. and A.Z. performed interface-SLiMs analyses. L.L., B.D., M.A.C., M.V. and P.F.-B. conducted interface-AF analyses. J.F.-M., S.B.M., J.-C.T. and R.V. performed the holdup and peptide assays. T.C.A.H. and T.C. analysed effector ecology. V.Y., N.S.v.H., D.K.J.P., S.R., F.O., M.T., J.B., D. Kotlarz, D. Krappmann, M. Boes and P.F.-B. performed cell-based assays. V.Y., B.D., H.H., J.F.-M. and P.F.-B. generated visualization and developed the figures. P.F.-B., C.F., A.Z., C.B., T.R. and D. Krappmann acquired funding. P.F.-B., B.D., A.Z., V.Y., B.W., T.H. and C.F. wrote the manuscript.

## Funding

## Competing interests

The authors declare no competing interests.

## Additional information

**Extended data** is available for this paper at https://doi.org/10.1038/s41564-025-02241-y.

**Correspondence and requests for materials** should be addressed to Andreas Zanzoni or Pascal Falter-Braun.

**Veronika Young** [1,23], **Bushra Dohai**[1,23], **Hridi Halder**[1], **Jaime Fernandez-Macgregor**[2,3], **Niels S. van Heusden** [4], **Thomas C. A. Hitch** [5], **Benjamin Weller**[1], **Patrick Hyden**[6], **Deeya Saha**[2,22], **Daan K. J. Pieren** [4], **Sonja Rittchen**[4], **Luke Lambourne** [7,8,9], **Sibusiso B. Maseko**[10], **Chung-Wen Lin**[1], **Ye Min Tun**[1], **Jonas Bibus**[11], **Luisa Pletschacher**[1], **Mégane Boujeant**[2], **Sébastien A. Choteau**[2], **Lou Bergogne** [2], **Jérémie Perrin**[2], **Franziska Ober**[12], **Patrick Schwehn**[1], **Simin T. Rothballer**[1], **Melina Altmann**[1], **Stefan Altmann**[1], **Alexandra Strobel**[1], **Michael Rothballer**[1], **Marie Tofaute**[12], **Daniel Kotlarz**[11,13,14], **Matthias Heinig**[15,16], **Thomas Clavel** [5], **Michael A. Calderwood** [7,8,9], **Marc Vidal**[7,8,9], **Jean-Claude Twizere** [10,17,18], **Renaud Vincentelli** [3], **Daniel Krappmann** [12], **Marianne Boes** [19], **Claudia Falter**[1], **Thomas Rattei** [6], **Christine Brun**[2,20], **Andreas Zanzoni** [2] ✉ & **Pascal Falter-Braun** [1,21] ✉

[1]Institute of Network Biology (INET), Molecular Targets and Therapeutics Center (MTTC), Helmholtz Munich, German Research Center for Environmental Health, Neuherberg, Germany. [2]Aix-Marseille University, INSERM, TAGC, UMR_S1090, Turing Center for Living Systems, Marseille, France. [3]Aix-Marseille University, CNRS, AFMB, UMR CNRS 7257, Turing Center for Living Systems, Marseille, France. [4]Center for Translational Immunology, University Medical Center Utrecht, Utrecht University, Utrecht, the Netherlands. [5]Functional Microbiome Research Group, Institute of Medical Microbiology, University Hospital of RWTH Aachen, Aachen, Germany. [6]Department of Microbiology and Ecosystem Science, Research Network: Chemistry meets Microbiology, University of Vienna, Vienna, Austria. [7]Center for Cancer Systems Biology (CCSB), Dana-Farber Cancer Institute, Boston, MA, USA. [8]Department of Genetics, Blavatnik Institute, Harvard Medical School, Boston, MA, USA. [9]Department of Cancer Biology, Dana-Farber Cancer Institute, Boston, MA, USA. [10]Laboratory of Viral Interactomes, GIGA Institute, University of Liège, Liège, Belgium. [11]Dr von Hauner Children's Hospital, Department of Pediatrics, University Hospital, Ludwig-Maximilians-Universität Munich, Munich, Germany. [12]Research Unit Signaling and Translation, Molecular Targets and Therapeutics Center (MTTC), Helmholtz Munich, German Research Center for Environmental Health, Neuherberg, Germany. [13]Institute of Translational Genomics, Helmholtz Zentrum München, German Research Center for Environmental Health, Neuherberg, Germany. [14]German Center for Child and Adolescent Health (DZKJ), partner site Munich, Munich, Germany. [15]Institute of Computational Biology (ICB), Computational Health Center, Helmholtz Munich, German Research Center for Environmental Health, Neuherberg, Germany. [16]Department of Computer Science, TUM School of Computation, Information and Technology, Technical University of Munich, Garching, Germany. [17]TERRA Teaching and Research Centre, University of Liège, Gembloux, Belgium. [18]Laboratory of Algal Synthetic and Systems Biology, Division of Science, New York University Abu Dhabi, Abu Dhabi, United Arab Emirates.

[19]Center for Translational Immunology, Pediatrics Department, University Medical Center Utrecht, Utrecht University, Utrecht, the Netherlands. [20]CNRS, Marseille, France. [21]Microbe-Host Interactions, Faculty of Biology, Ludwig-Maximilians-Universität (LMU) München, Planegg-Martinsried, Germany. [22]Present address: Department of Biotechnology, Faculty of Life and Allied Health Sciences, Ramaiah University of Applied Sciences, Bangalore, India. [23]These authors contributed equally: Veronika Young, Bushra Dohai. ✉e-mail: andreas.zanzoni@univ-amu.fr; pascal.falter-braun@helmholtz-munich.de

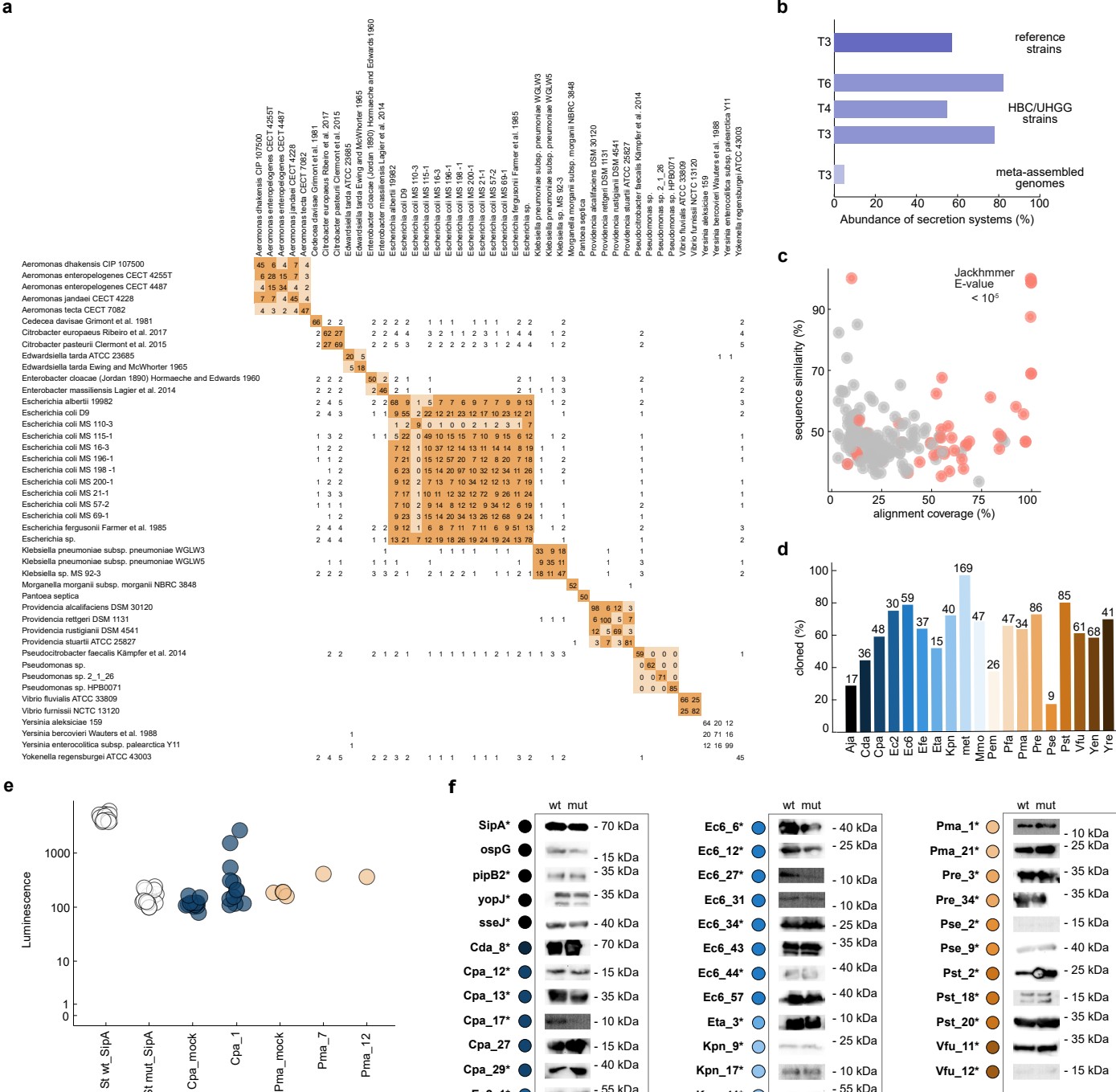

**Extended Data Fig. 1 | T3SS in strains of the commensal gut microbiome.**
**a**, Effector complements comparison of the 44 T3SS+ Pseudomonadota reference strains. Numbers indicate the count of shared effectors at > 90% mutual sequence similarity across 90% common sequence length among the indicated strains (Supplementary Data 1). **b**, Abundance of secretion systems in Pseudomonadota genomes among the 77 reference strains of human intestinal and stool samples, in a collection of 4,475 strains isolated from normal human guts (HBC/BIO-ML/GMC strains) and in meta-assembled genomes (MAG) of normal human guts. **c**, Similarity of identified 182 candidate effectors from the 770 T3SS+ MAGs with 1,195 effectors from pathogenic microbes across the range of alignment coverages. Full data for all panels in Supplementary Data 4. **d**, Cloning success: success rates of effector open reading frame (ORF) cloning for the indicated reference strains, and the number of obtained and sequence verified ORFs (on top of bars) (Supplementary Data 7). **e**, Luminescence from injection assays with *Salmonella* Typhimurium wt and Δ*sctV* strains expressing SipA, and *Citrobacter pasteurii* and *Phytobacter massiliensis* expressing the indicated effectors. Each data point represents a single technical replicate. **f**. Western blots showing expression of FLAG-tagged effectors in wt and Δ*sctV S*. Typhimurium in the indicated strains.

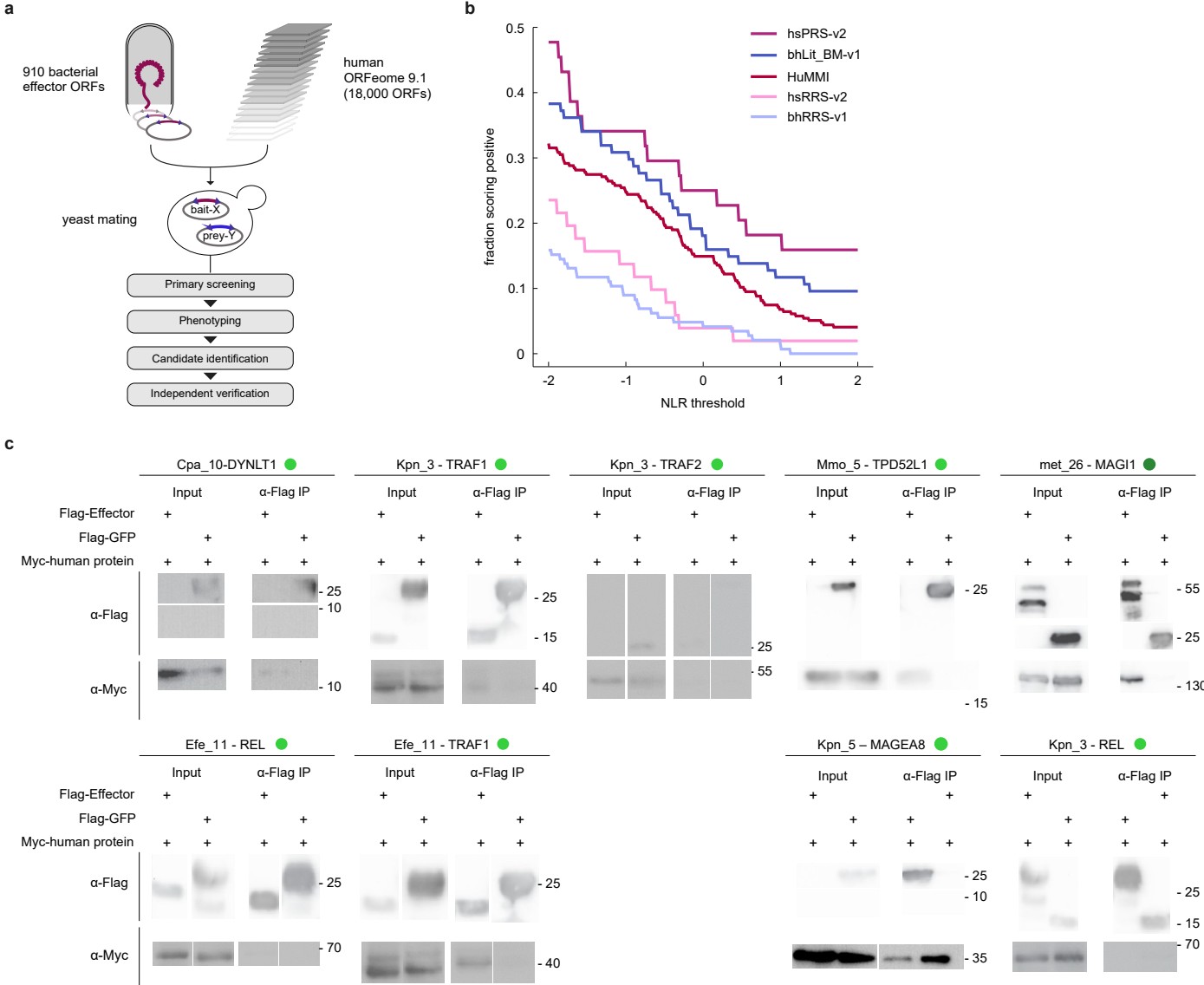

**Extended Data Fig. 2 | Discovery and validation of HuMMI. a**, Schematic of the multi-assay screening pipeline based on initial screening of bacterial ORFs against the human ORFeome 9.1. The primary screening involved screening against human protein pools, followed by retesting of positives, identification of candidate pairs by sequencing and final, independent four-fold verification. **b**, Detection rates of protein pairs in different sets across varying thresholds in yN2H. Fractions scoring positive of the HuMMI dataset and benchmarking datasets (hsPRS-v2, bhLit_BM-v1, hsRRS-v2, bhRRS-v1) depending on the threshold of the normalized luminescence ratio (NLR). Full data in Supplementary Data 14. **c**, Co-immunoprecipitation of indicated Myc-tagged human proteins by indicated FLAG-tagged effectors or FLAG-GFP as negative control. Input: Cell lysates. Molecular weight markers are given in kDa. Dark green dots: successful co-immunoprecipitation. Light green dots: successful co-immunoprecipitation, but weak or no effector detection in lysate. Blot lanes were partially rearranged.

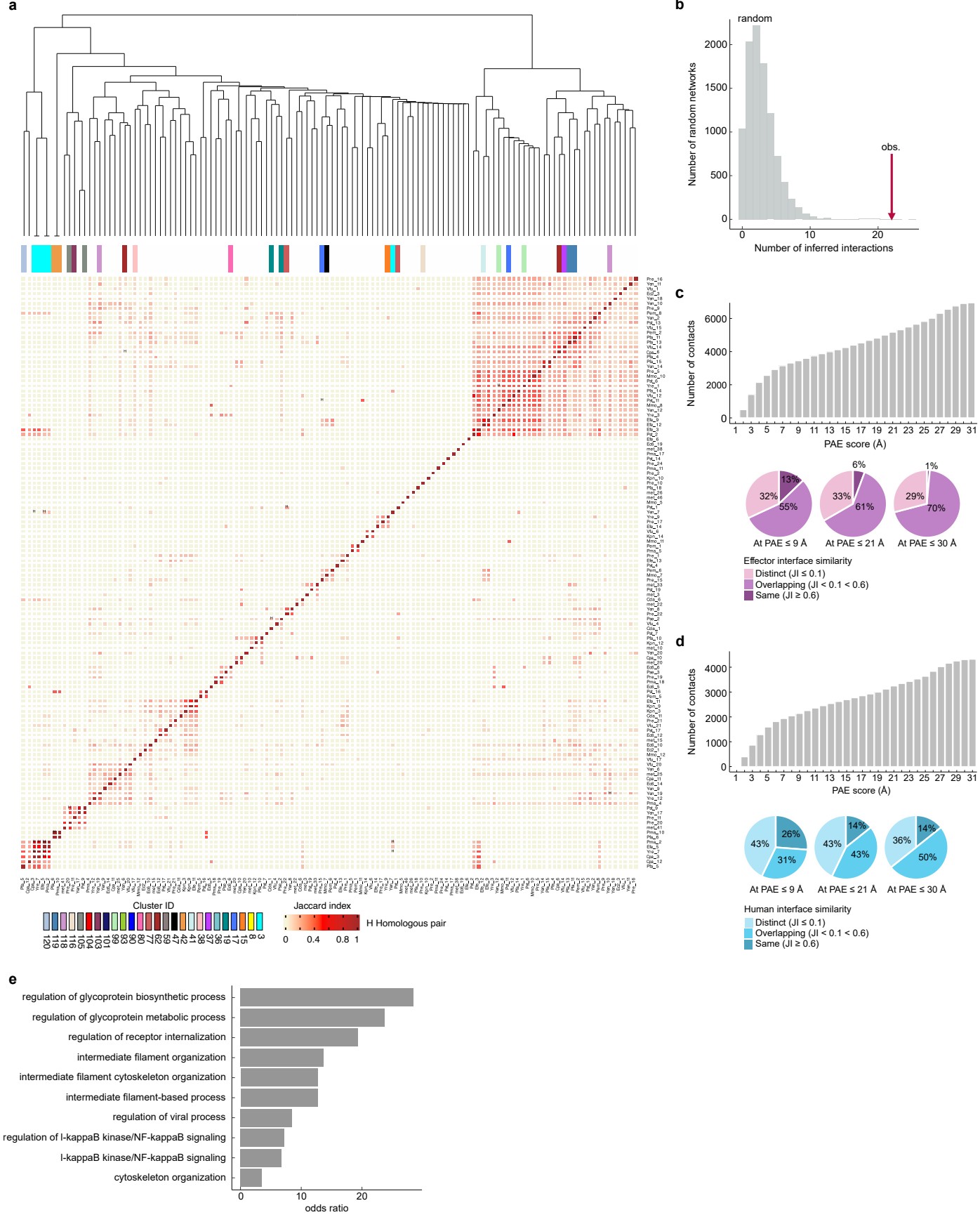

**Extended Data Fig. 3 | See next page for caption.**

**Extended Data Fig. 3 | Patterns of bacterial effector-human protein interactions. a**, Jaccard-interaction similarity of all interacting effector-pairs with at least three shared human interactors. Color-intensity correlates with Jaccard-index. Effector pairs marked with "H" share the same homology cluster. Clusters are color-coded according to the legend. **b**, Actual count of motif-domain pairs matching at least two stringency criteria identified in HuMMI$_{MAIN}$ (arrow) compared to n = 10,000 randomized control networks (empirical $P$ = 0.0003). **c**, (Top) Distribution of residue–residue contacts across predicted aligned error (PAE) scores for interfaces between bacterial effectors and their human targets. (Bottom) Proportions of human–human protein pairs targeted by the same bacterial effector, grouped by interface similarity at different PAE thresholds (Jaccard Index (JI) categories: *Distinct* ≤ 0.1, *Overlapping* 0.1–0.6, *Same* ≥ 0.6). Pie charts show similarity distributions for contacts with PAE ≤ 9Å

(50th percentile), PAE ≤ 21Å (75th percentile), and PAE ≤ 30 Å (9$^{5th}$ percentile). **d**, Top: Distribution of residue–residue contacts across predicted aligned error (PAE) scores for interfaces between human proteins and their bacterial effectors. Bottom: Proportions of bacterial effector–effector pairs targeting the same human protein, grouped by interface similarity at different PAE thresholds (JI categories: *Distinct* ≤ 0.1, *Overlapping* 0.1 – 0.6, *Same* ≥ 0.6). Pie charts show similarity distributions for contacts with PAE ≤ 9 Å (50th percentile), PAE ≤ 21Å (75th percentile), and PAE ≤ 30 Å (95th percentile). **e**, GO enrichment for convergence proteins. OR of functional annotations enriched among effector-targeted human proteins that are subject of convergence (FDR < 0.05, Fisher's exact test with Bonferroni FDR correction). Full data and precise FDR and OR values in Supplementary Data 20.

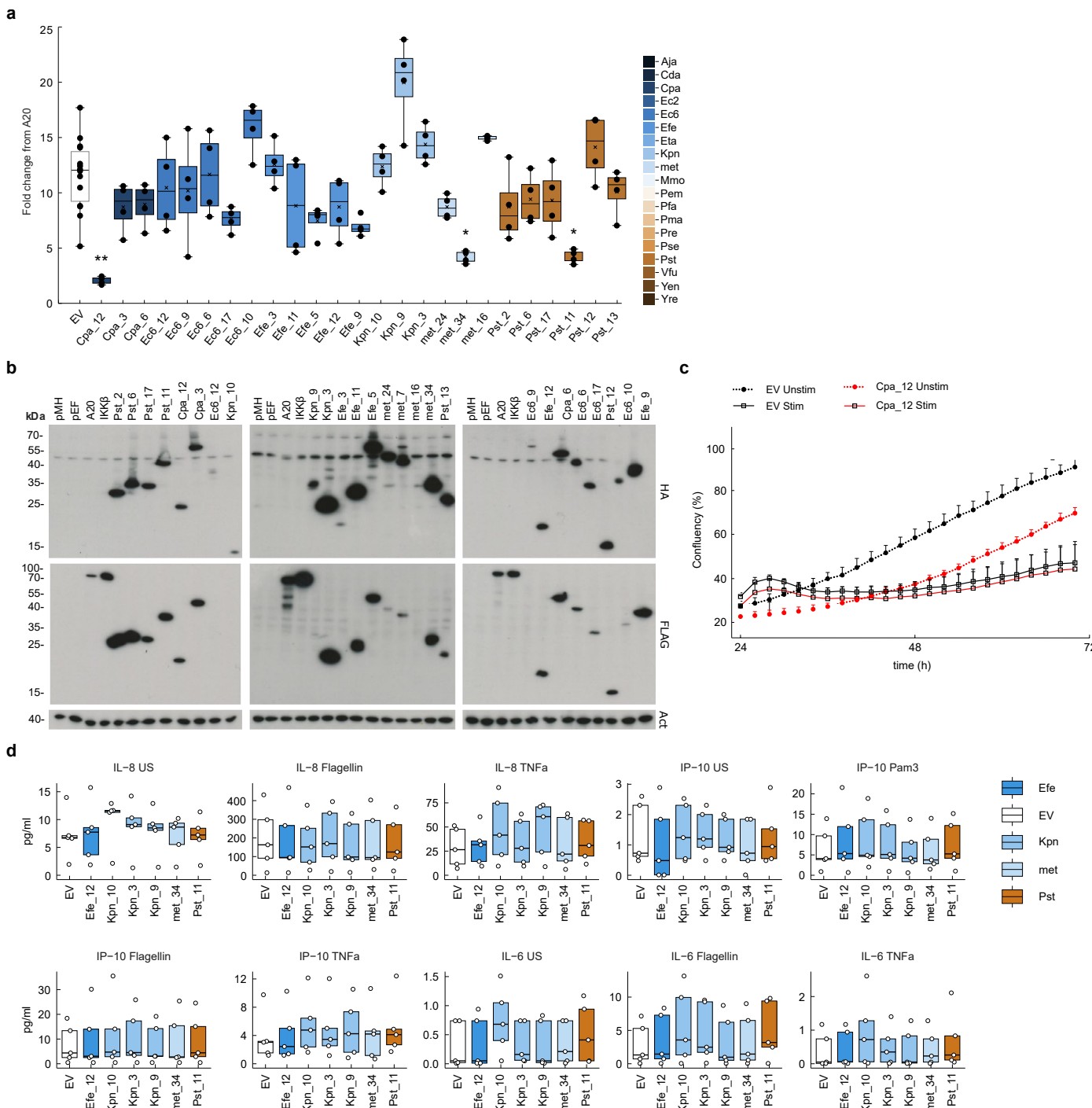

**Extended Data Fig. 4 | Effector impact on human cell function. a**. Relative NF-κB transcriptional reporter activity of HEK293 cells expressing the indicated effectors under TNF-stimulated conditions (Kruskal-Wallis test with Dunn's post-hoc comparisons, *P < 0.05, **P = 0.01, n = 4 biological replicates). Boxes represent IQR, with the bold black line representing the mean; whiskers indicate highest and lowest data point within 1.5 IQR. **b**, Representative anti-Hemagglutinin (HA) and anti-Flag (FLAG) Western blots showing expression of transfected effector proteins relative to actin control (ACT), which was run on a different blot. Empty pMH-Flag-HA (pMH), empty pEF4 (pEF). **c**. Representative proliferation curves of Caco-2 cells transfected with empty vector (EV) or

Cpa_12 in basal conditions (unstim) or following pro-inflammatory stimulation (stim) over 72 h after sorting. Error bars indicate one standard deviation above and below the mean. **d**, Concentration of cytokines secreted by Caco-2 cells transfected with the indicated effectors in basal conditions (US) or following stimulation with the indicated elicitors. EV indicates empty vector mock control. Indicated P values calculated by Kruskal-Wallis test with Dunn's post-hoc comparisons (n = 5). Boxes represent IQR, with the bold black line representing the mean; whiskers indicate highest and lowest data point. Raw measurements, n, and precise P values for all panels in Supplementary Data 24 and 25.

# Reporting Summary

## Statistics

For all statistical analyses, confirm that the following items are present in the figure legend, table legend, main text, or Methods section.

| n/a | Confirmed | |
|---|---|---|
| ☐ | ☒ | The exact sample size (*n*) for each experimental group/condition, given as a discrete number and unit of measurement |
| ☐ | ☒ | A statement on whether measurements were taken from distinct samples or whether the same sample was measured repeatedly |
| ☐ | ☒ | The statistical test(s) used AND whether they are one- or two-sided<br>*Only common tests should be described solely by name; describe more complex techniques in the Methods section.* |
| ☐ | ☒ | A description of all covariates tested |
| ☐ | ☒ | A description of any assumptions or corrections, such as tests of normality and adjustment for multiple comparisons |
| ☐ | ☒ | A full description of the statistical parameters including central tendency (e.g. means) or other basic estimates (e.g. regression coefficient) AND variation (e.g. standard deviation) or associated estimates of uncertainty (e.g. confidence intervals) |
| ☐ | ☒ | For null hypothesis testing, the test statistic (e.g. $F$, $t$, $r$) with confidence intervals, effect sizes, degrees of freedom and $P$ value noted<br>*Give P values as exact values whenever suitable.* |
| ☒ | ☐ | For Bayesian analysis, information on the choice of priors and Markov chain Monte Carlo settings |
| ☒ | ☐ | For hierarchical and complex designs, identification of the appropriate level for tests and full reporting of outcomes |
| ☐ | ☒ | Estimates of effect sizes (e.g. Cohen's *d*, Pearson's *r*), indicating how they were calculated |

*Our web collection on statistics for biologists contains articles on many of the points above.*

## Software and code

Policy information about availability of computer code

| Data collection | DSMZ ( via BacDive)<br>ATCC (atcc.org)<br>BEI (beiresources.org)<br>GenBank (release 229)<br>Human gastrointestinal genome collection (HBC)<br>Unified gastrointestinal genome collection (UHGG)<br>BastionHub database (August 29th , 2022)<br>ORFeome collection v9.1<br>IMEx consortium protein interaction databases<br>PSICQUIC webservice (May 10th, 2021)<br>InterPro database<br>ELM database<br>3did database<br>Recon3D<br>Open Targets Genetics<br>Inflammatory Bowel Disease Multi'omics DataBases (IBDMD)<br>Broad Institute-OpenBiome Microbiome Library (BIO-ML)<br>Global Microbiome Conservancy (GMC)<br>UniRef90 (January 2024)<br>AlphaFold DB |
|---|---|

| | 26 Supplementary data files generated in this study are inccluded and described in the Supplementary information file |
|---|---|
| Data analysis | EffectiveDB classifier<br>EggNOG 4 annotations<br>CheckM<br>Prodigal (version 2.6.3)<br>CD-HIT (version 4.8.1)<br>EffectiveT3 v.2.0.1<br>DeepT3 2.0<br>pEffect<br>TMHMM version 2.0<br>Needleman Wunsch algorithm<br>Emboss package<br>BLAST (stand-alone, version 2.10)<br>GTDB-Tk (v2.1)<br>iTOL v4<br>FastANI v1.0<br>R (version 4.2.1)<br>mimicINT<br>InterProScan v5.75<br>SLiMProb tool<br>SLiMSuite software package v1.11.0<br>gprofiler2 (version 0.2.1 R package)<br>RWR-MH<br>DIAMOND 0.9.24<br>scipy 1.9.3<br>python 3.10.12<br>cytoscape (version 3.9.0)<br>Vienna Scientific Cluster (VSC)<br>Jackhmmer tool<br>Foldseek<br>AlphaFold<br>alphapickle script v1.4.1<br>InterPro release 106.0<br>IUPred 1.0<br>IUPred3<br>biomRt R package (version 2.60.1; Bioconductor 3.19)<br>Microsoft Excel<br>https://doi.org/10.5281/zenodo.11951539<br>https://doi.org/10.5281/zenodo.11400863<br>https://doi.org/10.5281/zenodo.12743976<br>https://doi.org/10.5281/zenodo.16816224<br>https://doi.org/10.5281/zenodo.16883544 |

For manuscripts utilizing custom algorithms or software that are central to the research but not yet described in published literature, software must be made available to editors and reviewers. We strongly encourage code deposition in a community repository (e.g. GitHub). See the Nature Portfolio guidelines for submitting code & software for further information.

# Data

Policy information about availability of data

All manuscripts must include a data availability statement. This statement should provide the following information, where applicable:
- Accession codes, unique identifiers, or web links for publicly available datasets
- A description of any restrictions on data availability
- For clinical datasets or third party data, please ensure that the statement adheres to our policy

All sequence, interaction, and functional data generated in this study are available as supplementary information. The effectors identified and cloned for interactome mapping are presented in Supplementary Data 7. All protein-protein interaction data acquired in this study can be found in Supplementary Data 11. The data for functional validation assays can be found in Supplementary Data 24,25,26. All protein interaction data have been deposited to the IMEx consrtium(http://www.imexconsortium.org) through IntAct and assigned the identifier IM-29849. New effector sequences have been submitted to GenBank: BankIt2727690: OR372873 - OR373035 and OR509516 - OR509528.

# Research involving human participants, their data, or biological material

Policy information about studies with human participants or human data. See also policy information about sex, gender (identity/presentation), and sexual orientation and race, ethnicity and racism.

| Reporting on sex and gender | *Use the terms sex (biological attribute) and gender (shaped by social and cultural circumstances) carefully in order to avoid confusing both terms. Indicate if findings apply to only one sex or gender; describe whether sex and gender were considered in study design; whether sex and/or gender was determined based on self-reporting or assigned and methods used.*<br>*Provide in the source data disaggregated sex and gender data, where this information has been collected, and if consent has been obtained for sharing of individual-level data; provide overall numbers in this Reporting Summary. Please state if this* |
|---|---|

*information has not been collected.*
*Report sex- and gender-based analyses where performed, justify reasons for lack of sex- and gender-based analysis.*

| Reporting on race, ethnicity, or other socially relevant groupings | *Please specify the socially constructed or socially relevant categorization variable(s) used in your manuscript and explain why they were used. Please note that such variables should not be used as proxies for other socially constructed/relevant variables (for example, race or ethnicity should not be used as a proxy for socioeconomic status).*<br>*Provide clear definitions of the relevant terms used, how they were provided (by the participants/respondents, the researchers, or third parties), and the method(s) used to classify people into the different categories (e.g. self-report, census or administrative data, social media data, etc.)*<br>*Please provide details about how you controlled for confounding variables in your analyses.* |
|---|---|
| Population characteristics | *Describe the covariate-relevant population characteristics of the human research participants (e.g. age, genotypic information, past and current diagnosis and treatment categories). If you filled out the behavioural & social sciences study design questions and have nothing to add here, write "See above."* |
| Recruitment | *Describe how participants were recruited. Outline any potential self-selection bias or other biases that may be present and how these are likely to impact results.* |
| Ethics oversight | *Identify the organization(s) that approved the study protocol.* |

Note that full information on the approval of the study protocol must also be provided in the manuscript.

# Field-specific reporting

Please select the one below that is the best fit for your research. If you are not sure, read the appropriate sections before making your selection.

☒ Life sciences ☐ Behavioural & social sciences ☐ Ecological, evolutionary & environmental sciences

For a reference copy of the document with all sections, see nature.com/documents/nr-reporting-summary-flat.pdf

# Life sciences study design

All studies must disclose on these points even when the disclosure is negative.

| Sample size | No statistical methods were used to predetermine sample size. |
|---|---|
| Data exclusions | No data were excluded. |
| Replication | All attempts at replication were successful. |
| Randomization | Randomization for network permutations was done while maintaining network topology (maintaining the number of interactions but shuffling the edges) |
| Blinding | CNN supported manual scoring (confirmation) of Y2H pairwise tests was blinded. Statistical analysis of NF-kappaB activation assays was blinded. |

# Behavioural & social sciences study design

All studies must disclose on these points even when the disclosure is negative.

| Study description | *Briefly describe the study type including whether data are quantitative, qualitative, or mixed-methods (e.g. qualitative cross-sectional, quantitative experimental, mixed-methods case study).* |
|---|---|
| Research sample | *State the research sample (e.g. Harvard university undergraduates, villagers in rural India) and provide relevant demographic information (e.g. age, sex) and indicate whether the sample is representative. Provide a rationale for the study sample chosen. For studies involving existing datasets, please describe the dataset and source.* |
| Sampling strategy | *Describe the sampling procedure (e.g. random, snowball, stratified, convenience). Describe the statistical methods that were used to predetermine sample size OR if no sample-size calculation was performed, describe how sample sizes were chosen and provide a rationale for why these sample sizes are sufficient. For qualitative data, please indicate whether data saturation was considered, and what criteria were used to decide that no further sampling was needed.* |
| Data collection | *Provide details about the data collection procedure, including the instruments or devices used to record the data (e.g. pen and paper, computer, eye tracker, video or audio equipment) whether anyone was present besides the participant(s) and the researcher, and whether the researcher was blind to experimental condition and/or the study hypothesis during data collection.* |
| Timing | *Indicate the start and stop dates of data collection. If there is a gap between collection periods, state the dates for each sample cohort.* |
| Data exclusions | *If no data were excluded from the analyses, state so OR if data were excluded, provide the exact number of exclusions and the* |

| Data exclusions | *rationale behind them, indicating whether exclusion criteria were pre-established.* |
|---|---|
| Non-participation | *State how many participants dropped out/declined participation and the reason(s) given OR provide response rate OR state that no participants dropped out/declined participation.* |
| Randomization | *If participants were not allocated into experimental groups, state so OR describe how participants were allocated to groups, and if allocation was not random, describe how covariates were controlled.* |

# Ecological, evolutionary & environmental sciences study design

All studies must disclose on these points even when the disclosure is negative.

| Study description | *Briefly describe the study. For quantitative data include treatment factors and interactions, design structure (e.g. factorial, nested, hierarchical), nature and number of experimental units and replicates.* |
|---|---|
| Research sample | *Describe the research sample (e.g. a group of tagged Passer domesticus, all Stenocereus thurberi within Organ Pipe Cactus National Monument), and provide a rationale for the sample choice. When relevant, describe the organism taxa, source, sex, age range and any manipulations. State what population the sample is meant to represent when applicable. For studies involving existing datasets, describe the data and its source.* |
| Sampling strategy | *Note the sampling procedure. Describe the statistical methods that were used to predetermine sample size OR if no sample-size calculation was performed, describe how sample sizes were chosen and provide a rationale for why these sample sizes are sufficient.* |
| Data collection | *Describe the data collection procedure, including who recorded the data and how.* |
| Timing and spatial scale | *Indicate the start and stop dates of data collection, noting the frequency and periodicity of sampling and providing a rationale for these choices. If there is a gap between collection periods, state the dates for each sample cohort. Specify the spatial scale from which the data are taken* |
| Data exclusions | *If no data were excluded from the analyses, state so OR if data were excluded, describe the exclusions and the rationale behind them, indicating whether exclusion criteria were pre-established.* |
| Reproducibility | *Describe the measures taken to verify the reproducibility of experimental findings. For each experiment, note whether any attempts to repeat the experiment failed OR state that all attempts to repeat the experiment were successful.* |
| Randomization | *Describe how samples/organisms/participants were allocated into groups. If allocation was not random, describe how covariates were controlled. If this is not relevant to your study, explain why.* |
| Blinding | *Describe the extent of blinding used during data acquisition and analysis. If blinding was not possible, describe why OR explain why blinding was not relevant to your study.* |

Did the study involve field work?  ☐ Yes   ☐ No

## Field work, collection and transport

| Field conditions | *Describe the study conditions for field work, providing relevant parameters (e.g. temperature, rainfall).* |
|---|---|
| Location | *State the location of the sampling or experiment, providing relevant parameters (e.g. latitude and longitude, elevation, water depth).* |
| Access & import/export | *Describe the efforts you have made to access habitats and to collect and import/export your samples in a responsible manner and in compliance with local, national and international laws, noting any permits that were obtained (give the name of the issuing authority, the date of issue, and any identifying information).* |
| Disturbance | *Describe any disturbance caused by the study and how it was minimized.* |

# Reporting for specific materials, systems and methods

We require information from authors about some types of materials, experimental systems and methods used in many studies. Here, indicate whether each material, system or method listed is relevant to your study. If you are not sure if a list item applies to your research, read the appropriate section before selecting a response.

## Materials & experimental systems

| n/a | Involved in the study |
|---|---|
| ☐ | ☒ Antibodies |
| ☐ | ☒ Eukaryotic cell lines |
| ☒ | ☐ Palaeontology and archaeology |
| ☒ | ☐ Animals and other organisms |
| ☒ | ☐ Clinical data |
| ☒ | ☐ Dual use research of concern |
| ☒ | ☐ Plants |

## Methods

| n/a | Involved in the study |
|---|---|
| ☒ | ☐ ChIP-seq |
| ☒ | ☐ Flow cytometry |
| ☒ | ☐ MRI-based neuroimaging |

# Antibodies

| | |
|---|---|
| Antibodies used | anti-Actin beta (SCBT cat. no. sc-47778, RRID:AB_626632)<br>anti-FLAG M2 (Sigma Aldrich cat. no. F3165, RRID:AB_259529)<br>anti-HA (Sigma-Aldrich cat. no. 11583816001, RRID:AB_514505)<br>LEGENDplex™ kit (Biolegend)<br>anti-mouse secondary antibody (Jackson ImmunoResearch Labs cat. no. 715-035-150, RRID:AB_2340770)<br>anti-mouse IgG secondary antibody (Santa Cruz Biotechnology, m-IgGκ BP-HRP; Cat. no. sc-516102) |
| Validation | - anti-Actin beta was not validated.<br>- anti-FLAG M2's sensitivity and specificity was validated by the manufacturer: Detects 2 ng of FLAG-BAP fusion protein by dot blot using chemiluminescent detection. Detects a single band of protein on a western blot from an E. coli crude cell lysate.<br>- anti-HA was validated by the manufacturer by western blot: Each lot of anti-HA antibody is tested for functionality and purity relative to a reference standard to confirm the quality of each new reagent preparation.<br>- LEGENplex kit: according to the manufacturer each lot of the Legendplex is compared to an internal gold-standard.<br>- anti-mouse secondary antibody from Jackson ImmunoResearch was not validated<br>- anti-mouse and anti-rabbit secondary antibodies from Santa Cruz Biotechnology were validated by the manufacturer. |

# Eukaryotic cell lines

Policy information about cell lines and Sex and Gender in Research

| | |
|---|---|
| Cell line source(s) | HEK 293 (RRID: CVCL_0045, DSMZ)<br>Caco2 cells (RRID:CVCL_0025)<br>HeLa-LgBit (gift from Prof. Samuel Wagner's lab, Universitätsklinikum Tübingen) |
| Authentication | None of the cell lines used were re-authenticated |
| Mycoplasma contamination | Caco2 cell lines were tested negative for mycoplasma contamination. HEK 293 and HeLa-LgBit were not tested for mycoplasma contamination. |
| Commonly misidentified lines<br>(See ICLAC register) | N/A |

# Palaeontology and Archaeology

| | |
|---|---|
| Specimen provenance | *Provide provenance information for specimens and describe permits that were obtained for the work (including the name of the issuing authority, the date of issue, and any identifying information). Permits should encompass collection and, where applicable, export.* |
| Specimen deposition | *Indicate where the specimens have been deposited to permit free access by other researchers.* |
| Dating methods | *If new dates are provided, describe how they were obtained (e.g. collection, storage, sample pretreatment and measurement), where they were obtained (i.e. lab name), the calibration program and the protocol for quality assurance OR state that no new dates are provided.* |

☐ Tick this box to confirm that the raw and calibrated dates are available in the paper or in Supplementary Information.

| | |
|---|---|
| Ethics oversight | *Identify the organization(s) that approved or provided guidance on the study protocol, OR state that no ethical approval or guidance was required and explain why not.* |

Note that full information on the approval of the study protocol must also be provided in the manuscript.

# Animals and other research organisms

Policy information about studies involving animals; ARRIVE guidelines recommended for reporting animal research, and Sex and Gender in Research

| | |
|---|---|
| Laboratory animals | *For laboratory animals, report species, strain and age OR state that the study did not involve laboratory animals.* |
| Wild animals | *Provide details on animals observed in or captured in the field; report species and age where possible. Describe how animals were caught and transported and what happened to captive animals after the study (if killed, explain why and describe method; if released, say where and when) OR state that the study did not involve wild animals.* |
| Reporting on sex | *Indicate if findings apply to only one sex; describe whether sex was considered in study design, methods used for assigning sex. Provide data disaggregated for sex where this information has been collected in the source data as appropriate; provide overall numbers in this Reporting Summary. Please state if this information has not been collected. Report sex-based analyses where performed, justify reasons for lack of sex-based analysis.* |
| Field-collected samples | *For laboratory work with field-collected samples, describe all relevant parameters such as housing, maintenance, temperature, photoperiod and end-of-experiment protocol OR state that the study did not involve samples collected from the field.* |
| Ethics oversight | *Identify the organization(s) that approved or provided guidance on the study protocol, OR state that no ethical approval or guidance was required and explain why not.* |

Note that full information on the approval of the study protocol must also be provided in the manuscript.

# Clinical data

Policy information about clinical studies
All manuscripts should comply with the ICMJE guidelines for publication of clinical research and a completed CONSORT checklist must be included with all submissions.

| | |
|---|---|
| Clinical trial registration | *Provide the trial registration number from ClinicalTrials.gov or an equivalent agency.* |
| Study protocol | *Note where the full trial protocol can be accessed OR if not available, explain why.* |
| Data collection | *Describe the settings and locales of data collection, noting the time periods of recruitment and data collection.* |
| Outcomes | *Describe how you pre-defined primary and secondary outcome measures and how you assessed these measures.* |

# Dual use research of concern

Policy information about dual use research of concern

## Hazards

Could the accidental, deliberate or reckless misuse of agents or technologies generated in the work, or the application of information presented in the manuscript, pose a threat to:

No | Yes
☐ ☐ Public health
☐ ☐ National security
☐ ☐ Crops and/or livestock
☐ ☐ Ecosystems
☐ ☐ Any other significant area

## Experiments of concern

Does the work involve any of these experiments of concern:

| No | Yes | |
|----|-----|---|
| ☐ | ☐ | Demonstrate how to render a vaccine ineffective |
| ☐ | ☐ | Confer resistance to therapeutically useful antibiotics or antiviral agents |
| ☐ | ☐ | Enhance the virulence of a pathogen or render a nonpathogen virulent |
| ☐ | ☐ | Increase transmissibility of a pathogen |
| ☐ | ☐ | Alter the host range of a pathogen |
| ☐ | ☐ | Enable evasion of diagnostic/detection modalities |
| ☐ | ☐ | Enable the weaponization of a biological agent or toxin |
| ☐ | ☐ | Any other potentially harmful combination of experiments and agents |

# Plants

| | |
|---|---|
| Seed stocks | *Report on the source of all seed stocks or other plant material used. If applicable, state the seed stock centre and catalogue number. If plant specimens were collected from the field, describe the collection location, date and sampling procedures.* |
| Novel plant genotypes | *Describe the methods by which all novel plant genotypes were produced. This includes those generated by transgenic approaches, gene editing, chemical/radiation-based mutagenesis and hybridization. For transgenic lines, describe the transformation method, the number of independent lines analyzed and the generation upon which experiments were performed. For gene-edited lines, describe the editor used, the endogenous sequence targeted for editing, the targeting guide RNA sequence (if applicable) and how the editor was applied.* |
| Authentication | *Describe any authentication procedures for each seed stock used or novel genotype generated. Describe any experiments used to assess the effect of a mutation and, where applicable, how potential secondary effects (e.g. second site T-DNA insertions, mosiacism, off-target gene editing) were examined.* |

# ChIP-seq

## Data deposition

☐ Confirm that both raw and final processed data have been deposited in a public database such as GEO.

☐ Confirm that you have deposited or provided access to graph files (e.g. BED files) for the called peaks.

| | |
|---|---|
| Data access links<br>*May remain private before publication.* | *For "Initial submission" or "Revised version" documents, provide reviewer access links.  For your "Final submission" document, provide a link to the deposited data.* |
| Files in database submission | *Provide a list of all files available in the database submission.* |
| Genome browser session<br>(e.g. UCSC) | *Provide a link to an anonymized genome browser session for "Initial submission" and "Revised version" documents only, to enable peer review.  Write "no longer applicable" for "Final submission" documents.* |

## Methodology

| | |
|---|---|
| Replicates | *Describe the experimental replicates, specifying number, type and replicate agreement.* |
| Sequencing depth | *Describe the sequencing depth for each experiment, providing the total number of reads, uniquely mapped reads, length of reads and whether they were paired- or single-end.* |
| Antibodies | *Describe the antibodies used for the ChIP-seq experiments; as applicable, provide supplier name, catalog number, clone name, and lot number.* |
| Peak calling parameters | *Specify the command line program and parameters used for read mapping and peak calling, including the ChIP, control and index files used.* |
| Data quality | *Describe the methods used to ensure data quality in full detail, including how many peaks are at FDR 5% and above 5-fold enrichment.* |
| Software | *Describe the software used to collect and analyze the ChIP-seq data. For custom code that has been deposited into a community repository, provide accession details.* |

# Flow Cytometry

## Plots

Confirm that:

☐ The axis labels state the marker and fluorochrome used (e.g. CD4-FITC).

☐ The axis scales are clearly visible. Include numbers along axes only for bottom left plot of group (a 'group' is an analysis of identical markers).

☐ All plots are contour plots with outliers or pseudocolor plots.

☒ A numerical value for number of cells or percentage (with statistics) is provided.

## Methodology

| | |
|---|---|
| Sample preparation | Transfected Caco2 cells in basal condition or after pro-inflammatory stimulation with BD GolgiStop treatment were detached using ice-cold PBS or trypsin/EDTA. After washing steps, the mean fluorescent intensity of the GFP+ cell population was measured. |
| Instrument | FACSFortessa |
| Software | FlowJo V10.8.1 |
| Cell population abundance | Cell counts are in Supplementary Table S6 |
| Gating strategy | We gated for GFP positive cells to assure successful transfection before following the protocols of LEGENDplex™ kit. |

☐ Tick this box to confirm that a figure exemplifying the gating strategy is provided in the Supplementary Information.

# Magnetic resonance imaging

## Experimental design

| | |
|---|---|
| Design type | *Indicate task or resting state; event-related or block design.* |
| Design specifications | *Specify the number of blocks, trials or experimental units per session and/or subject, and specify the length of each trial or block (if trials are blocked) and interval between trials.* |
| Behavioral performance measures | *State number and/or type of variables recorded (e.g. correct button press, response time) and what statistics were used to establish that the subjects were performing the task as expected (e.g. mean, range, and/or standard deviation across subjects).* |

## Acquisition

| | |
|---|---|
| Imaging type(s) | *Specify: functional, structural, diffusion, perfusion.* |
| Field strength | *Specify in Tesla* |
| Sequence & imaging parameters | *Specify the pulse sequence type (gradient echo, spin echo, etc.), imaging type (EPI, spiral, etc.), field of view, matrix size, slice thickness, orientation and TE/TR/flip angle.* |
| Area of acquisition | *State whether a whole brain scan was used OR define the area of acquisition, describing how the region was determined.* |

Diffusion MRI     ☐ Used     ☐ Not used

## Preprocessing

| | |
|---|---|
| Preprocessing software | *Provide detail on software version and revision number and on specific parameters (model/functions, brain extraction, segmentation, smoothing kernel size, etc.).* |
| Normalization | *If data were normalized/standardized, describe the approach(es): specify linear or non-linear and define image types used for transformation OR indicate that data were not normalized and explain rationale for lack of normalization.* |
| Normalization template | *Describe the template used for normalization/transformation, specifying subject space or group standardized space (e.g. original Talairach, MNI305, ICBM152) OR indicate that the data were not normalized.* |
| Noise and artifact removal | *Describe your procedure(s) for artifact and structured noise removal, specifying motion parameters, tissue signals and physiological signals (heart rate, respiration).* |

Volume censoring | *Define your software and/or method and criteria for volume censoring, and state the extent of such censoring.*

## Statistical modeling & inference

Model type and settings | *Specify type (mass univariate, multivariate, RSA, predictive, etc.) and describe essential details of the model at the first and second levels (e.g. fixed, random or mixed effects; drift or auto-correlation).*

Effect(s) tested | *Define precise effect in terms of the task or stimulus conditions instead of psychological concepts and indicate whether ANOVA or factorial designs were used.*

Specify type of analysis: ☐ Whole brain ☐ ROI-based ☐ Both

Statistic type for inference | *Specify voxel-wise or cluster-wise and report all relevant parameters for cluster-wise methods.*

(See Eklund et al. 2016)

Correction | *Describe the type of correction and how it is obtained for multiple comparisons (e.g. FWE, FDR, permutation or Monte Carlo).*

## Models & analysis

| n/a | Involved in the study |
|-----|----------------------|
| ☐ | ☐ Functional and/or effective connectivity |
| ☐ | ☐ Graph analysis |
| ☐ | ☐ Multivariate modeling or predictive analysis |

Functional and/or effective connectivity | *Report the measures of dependence used and the model details (e.g. Pearson correlation, partial correlation, mutual information).*

Graph analysis | *Report the dependent variable and connectivity measure, specifying weighted graph or binarized graph, subject- or group-level, and the global and/or node summaries used (e.g. clustering coefficient, efficiency, etc.).*

Multivariate modeling and predictive analysis | *Specify independent variables, features extraction and dimension reduction, model, training and evaluation metrics.*

