## [Peer Review File · Nature Microbiology]

Effector–host interactome map links type III secretion systems in healthy gut microbiomes to immune modulation

Corresponding Author: Professor Pascal Falter-Braun

Version 0:

Reviewer comments:

Reviewer #1

(Remarks to the Author)

This manuscript reports a comprehensive story of the identification of type III effectors from the human microbiome and their functional association with human disease.

The authors detected T3SS and type III effectors in the genomes and metagenomes of human commensal bacteria belonging to Pseudomonadota (in particular Gammaproteobacteria). These commensal type III effectors are sequence-unrelated to pathogen effectors. A large-scale Y2H screen identified human proteins interacting with commensal effectors. Many of those effectors target proteins related to disease and immunity, to a certain degree convergently and with specificity. In addition, they validated the implication of some of commensal effectors in the modulation of immune responses. Interestingly, they also found that the prevalence of effectors is different in metagenomes of disease and healthy patients. Based on the presented data, they conclude that type III effectors from the human microbiome modulate immunity, thereby affecting human disease.

This manuscript is a novel and interesting work and is very well presented and written, with a huge amount of data and analyses. As the function of commensal type III effectors is unknown in both human and plant systems, it conceptually advances our understanding of the function of bacterial type III effectors in human microbiome, and in general microbiome. Nevertheless, I have two key suggestions and other comments that would further improve this manuscript.

(Major points)

1. While the authors showed that commensal effectors are sequence-unrelated to known pathogen effectors, it remains unclear whether their functions are similar or different compared to pathogen effectors as many of microbial effectors have been shown to be sequence-unrelated anyway. Thus, it is important to compare targets of commensal and pathogen type III effectors. This will be widely influential as the difference in properties of commensal and pathogenic bacteria receives huge attention in host-microbe interactions including plant-bacterial interactions. The authors mention that functions of commensal effectors are outside of the pathogen lifestyle (line 134). Yet, for CD, it is suggested that commensal effectors promote disease development (line 425). Thus, an important question is that functions of effectors from pathogenic and commensal bacteria are fundamentally different or similar as they did in Osborne et al 2023. With either result, this will be strongly influential to the community. The authors could perform Y2H with pathogen effectors. Alternatively, they may analyze already existing data with deep discussion.
2. The authors showed that the effector function is largely independent of sequence similarity (line 213). It is known that effectors recognized by allelic receptors can be totally sequence-unrelated (i.e. doi.org/10.1073/pnas.2307604120) but fold a similar structure. Thus, it would be interesting to utilize AlphaFold to predict the structure of some sequence-unrelated effectors with predicted similar functions.

(Other points)

- In lines 72-74 and 397, the authors did not clearly distinguish between fungal and bacterial type III effectors. As the function of commensal T3SS and type III effectors in plants is unknown, describing this more accurately will make readers even more appreciate this work.
- The authors detected T3SS in 44 of 77 reference strain genomes (60%), in 478 of 2272 human gut gram-negative strains (21%), and in 770 of 16179 Pseudomonadota (5%). It is important to discuss this difference so that readers can get an idea what proportion of Pseudomonadota has T3SS in the human gut microbiome. I think that this would become a good basis of future comparisons in other systems.

Reviewer #2

(Remarks to the Author)

The authors present a very interesting premise, asking if commensals express and translocate T3SS effectors in a similar manner to what is seen in plants, where T3SS effectors not only are involved in pathogenesis, but can also aid in establishing mutualistic relationships. The concept is novel, and the *in silico* analysis performed combined with the *in vitro* assays (Y2H and

protein domain interactions via hold up and competition assays) provide a large wealth of new information that could open new routes of study, and would be of interest to many researchers in the fields of gut health, microbiome and immunity. However I do have a major qualm with the study in regards to biological relevance which I will detail below.

The authors identify a series of commensal strains encoding a complete T3SS from genomic data from several sources using the EffectiveDB classifier, and then proceed to use machine learning tools to identify putative T3SS effectors. From then onwards, all of these putative effectors are called effectors and are assumed to: i) be expressed by the strains and ii) be translocated by the strains. This jump I am unable to agree with, without at least some examples where these hypothetical effectors are actually being expressed and translocated into host cells. The rest of the study defines the potential relevance of these effectors via the study of their interactions with host proteins (all in vitro) and the effects of their overexpression in two different cell types. While this provides interesting insights, we still do not know if any of the commensal strains actually express and assemble a functional T3SS, and whether the effectors are being expressed and translocated, thus reducing the relevance of these networks. I do understand that commensals can be genetically intractable, but some of the effectors are encoded by E.coli strains which, if impossible to genetically manipulate, would allow these effectors and T3SS to at least be expressed in laboratory E.coli. The absence of any biological data supporting the expression and translocation of these effectors makes the rest of the study lack a solid foundation, as there is no data supporting that these commensal strains indeed "communicate" with host cells via T3SS effectors.

In regards to the statistics, to the best of my knowledge, the statistical tests carried out are appropriate and defined in the figure legends, but I do have an issue with how some data sets are considered; even if the statistical test in itself is correct, I believe its not being applied on the adequate values. This refers to Figure 5c and d, and extended figure 5f, which all refer to either Icam1 expression or cytokine release from Caco-2 cells. The graphs mix both technical/analytical and biological repeats (denoted by the symbols and clearly shown in the extended data tables 6D, 6H, 6I) and the authors carry statistical tests using both indistinguishably, which is incorrect if the aim is to extract biologically-relevant and robust conclusions. To take into account biological variability and obtain reproducible data, I believe one should take the average of technical replicates before carrying out the statistical tests.

Additional comments:

- To avoid just relying on basic in vitro data, an immunoprecipitation experiment could be performed to see if the overexpressed effector indeed pulls-down the endogenously-expressed prey.
- Because of the absence of any examples of effectors being expressed and translocated, I find some of the inferences made a bit far-fetched. In particular the relationships between these hypothetical effectors, their putative targets and their association to disease are very clearly mentioned in the results section (lines 262-277 for example), which seems a bit too hypothetical and based on vague connections to be in this section (maybe better in discussion) or at least should be toned down.
- In lines 125-127 the authors emphasise that commensals can have T3SS effectors similar to those in pathogens despite being in different orders, pointing towards HGT, but the examples given actually are two species that can be pathogenic (*E. albertii* and *Y. enterocolitica*), which does not help in illustrating this point. To confuse matters further, the authors then say that effectors in commensals are distinct from those in pathogens, which sounds a bit contradictory (line 132-134). The figures seem to largely support the latter conclusion, and so I would appreciate if the text was more clear.
- How many GFP+ cells are counted to obtain the MFI in the Icam expression experiments? A representative dot plot would be nice to show the populations.

Reviewer #3

(Remarks to the Author)

The paper is an exploration of T3SS effector-human protein interactions, specifically focussing on commensal Proteobacteria. With 1,263 interactions discovered between 289 effectors and 430 human proteins, the study is an opportunity to uncover some exciting new biology, however it falls short in this due to its lack of sensitive sequence/structure searching for protein function prediction, and lack of in depth molecular exploration of any new effector-human protein pair representatives. There is a problem with uninformative figures in the main text, and over-interpretation of data. My detailed comments follow below.

Major:

1. I am unsure about the initial premise that there is a clear distinction between pathogenic vs non-pathogenic bacteria. Some pathogens can be opportunistic, or condition-specific. Since the distinction is blurry, it seems a reasonable expectation that commensals can also have T3SSs for supporting growth in certain circumstances.
2. The authors acknowledge that the algorithm they use may miss systems in certain bacteria (lines 97-98). Biases of the algorithm may also be why they found many hits in the well studied *Escherichia* (line 99). So given that the results are unlikely to be exhaustive, I'm not sure Fig 1 is useful as a main figure. There is nothing wrong with such an approach for finding candidates for subsequent detailed experimental validation and exploration to find new biology, but since this computational part of the pipeline may be underestimating T3SS presence so much, the actual numbers of hits and in which taxa may not be meaningful.
3. Figure 2A is not interesting data worth reporting in a main figure. It can be hard to clone toxins, this does not tell us anything about what they do. However it should be noted that the ones that fail to clone are probably the most important effectors as they are they are the most toxic.
4. The section "Commensal effectors are unrelated to known pathogen effectors" is problematic since only identity calculated by simple pairwise sequence alignment is used. It would be appropriate here to use tools like JackHMMer to sensitivity cluster effectors using iterative HHM searching, HHpred to find hits in the effectors to PFam domains, Uniprot and PDB entries, and

AlpaFold/RosettaFold/ESMFold coupled with Foldseek to find similarities that are only apparent at the structure level. Horizontally transferred toxins/effectors can often be small and divergent, meaning more sensitive methods are required for finding homology. The sentence "In fact, clustering effectors by their pairwise interaction similarity identified substantial overlap outside the homology clusters (Extended Data Fig. 3), indicating that dissimilar effectors may have similar functions in the host" (line 209) suggests that some effectors classified as dissimilar might actually be similar if more appropriate sensitive methods are used.

5. In showing which "effectors" and human proteins interact, Fig 2G is an interesting panel but it is not clear what kinds of proteins we're looking at. The names of the proteins (especially the human ones) should be given here (rotate by 90 degrees for ease of reading as some names will be long). In general this is a problem throughout the paper and SI that full names of proteins are not used, making the results not very human readable. It is usual to include Uniprot protein names and IDs in the SI for human proteins picked up in screens.

6. Why is sensitivity so low? Perhaps true effector-target interactions are often transient? Isn't it also possible that many of the interactions picked up are proteins that interact for other reasons? The authors seem to assume that interactions of a predicted effector with a human protein mean that the human protein is the target of the effect. While this might be true in the majority of cases, is not clear to me that this is always so. Is it not possible that some interactions represent an immune response to inactivate effector-like proteins? And maybe some protein domains (perhaps the PDZ domain) are just 'sticky' in that they can readily bind other proteins with low specificity?

7. The authors should take advantage of state of the art structural prediction methods to try to model some of their interactions. AlphaFold now makes it easy to predict complexes and therefore interaction interfaces.

8. Fig 3c is not possible to extract any meaning from. How are the proteins organised on the vertical axes?

9. After identifying PDZ domains as potential interactors, the paper goes off on a tangent to investigate viral PDZ domains. It is not convincing that the 'competition' they observe is biologically meaningful. PDZ domains seem to be common domains that can bind to many different kinds of proteins.

10. What are the effectors that are more prevalent in CD than healthy guts? What kinds of proteins are they? Are these some that couldn't be cloned suggesting toxicity? Which are good targets for experimental exploration? The results are presented in a way that is removed from biological discovery and makes it hard for a reader to evaluate. The effector abbreviations used in the paper and SI are not helpful without a key (in each SI table) that allows the reader to easily find the gene/protein name and protein sequence of the effector.

Minor:

It is not properly explained how interfaces are predicted. The principle should be briefly outlined in the results.

line 65: "Especially" seems out of place here and can be deleted.

line 200: "The function of unknown proteins can often be inferred from better-studied orthologues, but convergence could also result from high sequence similarity among effectors." What is meant by convergence here? Convergent evolution? It needs better explaining.

Reviewer #4

(Remarks to the Author)

Key results

The manuscript highlights the role of Type III Secretion Systems (T3SS) in the human gut microbiome focusing on the Pseudomonadota phylum. Analysing large datasets from human gut samples demonstrates that T3SS effectors show minimal sequence similarity to known pathogenic effectors, suggesting distinct functions. The study reveals the microbiome-host interactions highlighting significant convergence of effectors on specific host proteins. The manuscript also explores the structural features, such as PDZ domains, and proposes competition between bacterial and viral proteins for host binding partners. The manuscript further investigates the contribution of effectors to immunomodulatory functions and disease pathogenesis. The authors identified enrichment in immune signalling, particularly in the NF- κ B pathway, as well as metabolism, linking to conditions like cancer and immunological diseases. Functional assays demonstrate that certain effectors can both activate and inhibit the NF- κ B pathway, influencing cell adhesion and cytokine secretion. Clinical analysis of patients with inflammatory bowel disease (IBD) suggests altered prevalence of effectors, potentially distinguishing Crohn's disease and ulcerative colitis proposing that commensal effectors' role in disease pathogenesis by modulating host cellular functions.

Validity

The presented workflow is robust and reproducible, some databases and tools have their version number, but I would suggest describing it for all (eg. ELM, IUPred, InterProScan). All source code is provided as zip file, but to track changes and facilitate collaborations, I would suggest storing them in a version control system (e.g. Git).

Significance

The study provides a new insight into the immunomodulatory effect of commensal T3SS effectors. I would suggest to emphasise better the difference between pathogenic and commensal T3SS effectors. The authors compared them in terms of sequence similarity but it would be great to have a look at the differences from a network perspective (eg. convergence) and comparing the immunomodulatory effect of commensal effectors to pathogens according to the literature. The 'Discussion' section is heavily

focused on the disease context and does not conclude the rest of the analysis carried out in the study (eg. the microbe-host interactome, convergence analysis or interface prediction).

Data and methodology

The manuscript describes a robust multi-layered analysis including experimental and in silico approaches in a well-organised, structured way. However, in the 'Meta-interactome mapping' method section, further clarification or contextualisation could improve the understanding, especially for readers less familiar with certain methodologies.

The utilisation of the Recon3D human metabolic model for metabolic subsystem analysis is appropriate. However, the assumptions and limitations of this model should be acknowledged for a more comprehensive interpretation of the metabolic findings.

Finally, limitations-related to this comprehensive analysis should be discussed in the 'Discussion' section.

Analytical approach

In general, the authors have employed a variety of statistical and computational methods to prove a robust and comprehensive workflow. However, the protein-protein interaction interface analysis highlights weaknesses in the computational pipeline. The domain-motif interaction based approach gives a directionality for the microbe-host interactions and the current study describes how domains on human proteins bind to motifs on bacterial effectors. But the authors aimed to demonstrate the impact of effectors on host proteins, hence the analysis does not answer the proposed goals. Besides, the workflow uses an earlier version of the IUPred tool that should be updated and the applied cut-off (0.2) is inappropriate to predict disordered regions on protein sequences.

As a computational biologist, assessing the experimental protocols and methods is outside the scope of my expertise.

Clarity and context

The manuscript provides a detailed description of methods and results referencing specific studies, databases, and tools. While most of the abbreviations are explained, others could be clarified to enhance understanding (eg. HU, HMM, BD, DMEM, HEK 293 - mention this is a cell line). One potential improvement could be the inclusion of a brief summary or overview at the beginning of the methods section to provide readers with a high-level understanding of the overall approach before delving into the details. Additionally, the integration of visual aids, such as flowcharts or diagrams, could enhance the clarity of the complex computational and experimental processes described in the methods.

References

The manuscript cites 110 articles covering the field of host-microbe interaction research, however there is a need for formal improvements: the citation for BastionHub is duplicated (22 and 79); in line 1152, there is a missing tab; doi IDs should be formatted properly, either highlighting them as a link or using only the ID.

Suggested improvements

Major improvements

The authors explored the interface between microbe-host proteins connecting bacterial motifs to human domains. As the domain-motif interactions are directed (the domain is binding to the target SLiM), the current analysis describes how human proteins are able to bind to bacterial effectors. To explore the effect of commensal effectors on human proteins, I suggest redoing the interface analysis considering the bacterial domains connected to human SLiMs.

Besides, I suggest using one of the newer versions of IUPred (2A or 3) as they provide a more precise prediction of disordered regions through defining two scores.

The authors defined the cut-off for 0.2 in the manuscript while the IUPRED tool describes the following: 'Residues with a predicted score above 0.5 are considered disordered, while residues with lower scores considered to be ordered.' Hence, the currently used 0.2 cut-off is not appropriate to predict disordered regions, the score should be at least 0.5 for both IUPred and ANCHOR values.

Considering and describing the limitations of the study in the 'Discussion' section.

Minor improvements

The 'Main' section is currently very short and it provides a brief background about the topic

The 'Result' header is missing

'From 16,179 Pseudomonadota MAG bins with high or intermediate genome quality' - define what is high/intermediate quality (line 92-93)

Missing '(' in line 438

Provide more details on the previously developed Y2H screening pipeline

Insert a paragraph about the workflow summary/create a workflow figure

Extend the 'Discussion' section concluding all the significant results not only focusing on disease-related importance of the commensal effectors.

The authors mention 1638 effectors derived from BastionHub database (in line 117), later 1195 gathered effectors are mentioned (in line 474) - Please clarify this.

Description of 'odd ratio' is missing from the text

The authors used their mimicINT tool for interface prediction that has not been published yet. Why is this the best option for interface prediction? Is not there any similar tool that has been already published?

Provide version number for all tools and databases (eg. ELM, InterProScan)

The authors compared commensal and pathogen effectors in terms of sequence similarity but it would be great to have a look at the differences from a network perspective (eg. convergence) and comparing the immunomodulatory effect of commensal effectors to pathogens according to the literature.

The assumptions and limitations of the Recon3D metabolic model should be acknowledged for a more comprehensive interpretation of the metabolic findings.

While some of the abbreviations are explained, others could be clarified to enhance understanding (eg. HU, HMM, BD, DMEM,

HEK 293 - mention this is a cell line).
BastionHub is cited twice (ref 22 and 79).
In line 1152, there is a missing tab by the citation.
doi IDs should be formatted properly, either highlighting them as a link or using only the ID.

Comments on figures

There is no need for Fig. 1c as the two numbers have been mentioned in the text
Describe the colour legend in Fig. 1d (blue-grey scale)
Reference for Fig. 2a is missing in the text
Fig. 2g is consisting of two panels that is confusing
What is 'HU' on Fig. 3e? Also, the yellow border is less visible
Fig. 3f is small, the 'FP' should be described in the figure caption
I would suggest changing the layout for Fig. 4d to a list/table instead of the current circle arrangement

Comments on Supplementary tables

As the tables consist of several worksheets related to different parts of the analysis, I would suggest to refer to the exact worksheet instead of the table number
Extended Data Table 2A-B describes the predicted effectors from collected reference strains and MAGs. Where is the HBC/UHGG-related data?

Points that should be clarified

Why did the authors select other phyla in the first part of the analysis from HBC and UHGG?
How did the authors select the commensal Pseudomonadota strains?
How did the authors select the pathogens from the BastionHub database?
What does 'mostly confidently predicted' mean in line 114 ("3,002 effector candidates from the 44 reference strains that were most confidently predicted by all tools")?
The authors set up a cut-off for 50% completeness and less than 5% contamination but how reliable are the computational tools of T3SS prediction when MAGs are incomplete or contaminated?
How representative are the collected 44 reference genomes and 770 MAGs of the overall gut microbiome?
How does the observed convergence in commensal effectors compare to the previously noted convergence in pathogenic effectors?

Tamas Korcsmaros

Decision Letter:

8th December 2023

Dear Professor Falter-Braun,

Thank you for your patience while your manuscript "A gut meta-interactome map reveals modulation of human immunity by microbiome effectors" was under peer-review at Nature Microbiology. It has now been seen by 4 referees, whose expertise and comments you will find at the end of this email. Although they find your work of some potential interest, they have raised a number of concerns that will need to be addressed before we can consider publication of the work in Nature Microbiology.

In particular, we ask you to address the technical concerns raised by Referees 3 and 4. Additionally, as suggested by all referees, please provide more in depth comparison between commensal and pathogen effectors. We also ask you to provide experimental validation for at least one strain, to show that these effectors are expressed and translocated via a functional T3SS. This could be an in vitro validation.

Should further experimental data allow you to address these criticisms, we would be happy to look at a revised manuscript.

Please include a data availability statement as a separate section after Methods but before references, under the heading "Data Availability". This section should inform readers about the availability of the data used to support the conclusions of your study. This information includes accession codes to public repositories (data banks for protein, DNA or RNA sequences, microarray, proteomics data etc...), references to source data published alongside the paper, unique identifiers such as URLs to data repository entries, or data set DOIs, and any other statement about data availability. At a minimum, you should include the following statement: "The data that support the findings of this study are available from the corresponding author upon request", mentioning any restrictions on availability. If DOIs are provided, we also strongly encourage including these in the Reference list (authors, title, publisher (repository name), identifier, year). For more guidance on how to write this section please see:

- * Include a "Response to referees" document detailing, point-by-point, how you addressed each referee comment. If no action was taken to address a point, you must provide a compelling argument. This response will be sent back to the referees along with the revised manuscript.
- * If you have not done so already we suggest that you begin to revise your manuscript so that it conforms to our Article format instructions at <http://www.nature.com/nmicrobiol/info/final-submission>. Refer also to any guidelines provided in this letter.
- * Include a revised version of any required reporting checklist. It will be available to referees (and, potentially, statisticians) to aid in their evaluation if the manuscript goes back for peer review. A revised checklist is essential for re-review of the paper.

When submitting the revised version of your manuscript, please pay close attention to our [href="https://www.nature.com/nature-portfolio/editorial-policies/image-integrity">Digital Image Integrity Guidelines.](https://www.nature.com/nature-portfolio/editorial-policies/image-integrity) and to the following points below:

Link Redacted

Note: This url links to your confidential homepage and associated information about manuscripts you may have submitted or be reviewing for us. If you wish to forward this e-mail to co-authors, please delete this link to your homepage first.

Nature Microbiology is committed to improving transparency in authorship. As part of our efforts in this direction, we are now requesting that all authors identified as 'corresponding author' on published papers create and link their Open Researcher and Contributor Identifier (ORCID) with their account on the Manuscript Tracking System (MTS), prior to acceptance. This applies to primary research papers only. ORCID helps the scientific community achieve unambiguous attribution of all scholarly contributions. You can create and link your ORCID from the home page of the MTS by clicking on 'Modify my Springer Nature account'. For more information please visit www.springernature.com/orcid.

If you wish to submit a suitably revised manuscript we would hope to receive it within 6 months. If you cannot send it within this time, please let us know. We will be happy to consider your revision, even if a similar study has been accepted for publication at Nature Microbiology or published elsewhere (up to a maximum of 6 months).

Yours sincerely,

Reviewer Expertise:

- Referee #1: effector-host interactions (plant field)
- Referee #2: T3SS, host-effector interactions
- Referee #3: bioinformatics, prediction of protein function, toxins
- Referee #4: systems biology, network modelling, microbe-cell interactions

Reviewer Comments:

Reviewer #1 (Remarks to the Author):

This manuscript reports a comprehensive story of the identification of type III effectors from the human microbiome and their functional association with human disease.

The authors detected T3SS and type III effectors in the genomes and metagenomes of human commensal bacteria belonging to Pseudomonadota (in particular Gammaproteobacteria). These commensal type III effectors are sequence-unrelated to pathogen effectors. A large-scale Y2H screen identified human proteins interacting with commensal effectors. Many of those effectors target proteins related to disease and immunity, to a certain degree convergently and with specificity. In addition, they validated the implication of some of commensal effectors in the modulation of immune responses. Interestingly, they also found that the

prevalence of effectors is different in metagenomes of disease and healthy patients. Based on the presented data, they conclude that type III effectors from the human microbiome modulate immunity, thereby affecting human disease.

This manuscript is a novel and interesting work and is very well presented and written, with a huge amount of data and analyses. As the function of commensal type III effectors is unknown in both human and plant systems, it conceptually advances our understanding of the function of bacterial type III effectors in human microbiome, and in general microbiome. Nevertheless, I have two key suggestions and other comments that would further improve this manuscript.

(Major points)

1. While the authors showed that commensal effectors are sequence-unrelated to known pathogen effectors, it remains unclear whether their functions are similar or different compared to pathogen effectors as many of microbial effectors have been shown to be sequence-unrelated anyway. Thus, it is important to compare targets of commensal and pathogen type III effectors. This will be widely influential as the difference in properties of commensal and pathogenic bacteria receives huge attention in host-microbe interactions including plant-bacterial interactions. The authors mention that functions of commensal effectors are outside of the pathogen lifestyle (line 134). Yet, for CD, it is suggested that commensal effectors promote disease development (line 425). Thus, an important question is that functions of effectors from pathogenic and commensal bacteria are fundamentally different or similar as they did in Osborne et al 2023. With either result, this will be strongly influential to the community. The authors could perform Y2H with pathogen effectors. Alternatively, they may analyze already existing data with deep discussion.

2. The authors showed that the effector function is largely independent of sequence similarity (line 213). It is known that effectors recognized by allelic receptors can be totally sequence-unrelated (i.e. doi.org/10.1073/pnas.2307604120) but fold a similar structure. Thus, it would be interesting to utilize AlphaFold to predict the structure of some sequence-unrelated effectors with predicted similar functions.

(Other points)

- In lines 72-74 and 397, the authors did not clearly distinguish between fungal and bacterial type III effectors. As the function of commensal T3SS and type III effectors in plants is unknown, describing this more accurately will make readers even more appreciate this work.

- The authors detected T3SS in 44 of 77 reference strain genomes (60%), in 478 of 2272 human gut gram-negative strains (21%), and in 770 of 16179 Pseudomonadota (5%). It is important to discuss this difference so that readers can get an idea what proportion of Pseudomonadota has T3SS in the human gut microbiome. I think that this would become a good basis of future comparisons in other systems.

Reviewer #2 (Remarks to the Author):

The authors present a very interesting premise, asking if commensals express and translocate T3SS effectors in a similar manner to what is seen in plants, where T3SS effectors not only are involved in pathogenesis, but can also aid in establishing mutualistic relationships. The concept is novel, and the *in silico* analysis performed combined with the *in vitro* assays (Y2H and protein domain interactions via hold up and competition assays) provide a large wealth of new information that could open new routes of study, and would be of interest to many researchers in the fields of gut health, microbiome and immunity. However I do have a major qualm with the study in regards to biological relevance which I will detail below.

The authors identify a series of commensal strains encoding a complete T3SS from genomic data from several sources using the EffectiveDB classifier, and then proceed to use machine learning tools to identify putative T3SS effectors. From then onwards, all of these putative effectors are called effectors and are assumed to: i) be expressed by the strains and ii) be translocated by the strains. This jump I am unable to agree with, without at least some examples where these hypothetical effectors are actually being expressed and translocated into host cells. The rest of the study defines the potential relevance of these effectors via the study of their interactions with host proteins (all *in vitro*) and the effects of their overexpression in two different cell types. While this provides interesting insights, we still do not know if any of the commensal strains actually express and assemble a functional T3SS, and whether the effectors are being expressed and translocated, thus reducing the relevance of these networks. I do understand that commensals can be genetically intractable, but some of the effectors are encoded by *E.coli* strains which, if impossible to genetically manipulate, would allow these effectors and T3SS to at least be expressed in laboratory *E.coli*. The absence of any biological data supporting the expression and translocation of these effectors makes the rest of the study lack a solid foundation, as there is no data supporting that these commensal strains indeed "communicate" with host cells via T3SS effectors.

In regards to the statistics, to the best of my knowledge, the statistical tests carried out are appropriate and defined in the figure legends, but I do have an issue with how some data sets are considered; even if the statistical test in itself is correct, I believe its not being applied on the adequate values. This refers to Figure 5c and d, and extended figure 5f, which all refer to either *Icam1* expression or cytokine release from Caco-2 cells. The graphs mix both technical/analytical and biological repeats (denoted by the symbols and clearly shown in the extended data tables 6D, 6H, 6I) and the authors carry statistical tests using both indistinguishably, which is incorrect if the aim is to extract biologically-relevant and robust conclusions. To take into account biological variability and obtain reproducible data, I believe one should take the average of technical replicates before carrying out the statistical tests.

Additional comments:

- To avoid just relying on basic *in vitro* data, an immunoprecipitation experiment could be performed to see if the overexpressed effector indeed pulls-down the endogenously-expressed prey.
- Because of the absence of any examples of effectors being expressed and translocated, I find some of the inferences made a bit far-fetched. In particular the relationships between these hypothetical effectors, their putative targets and their association to disease are very clearly mentioned in the results section (lines 262-277 for example), which seems a bit too hypothetical and

based on vague connections to be in this section (maybe better in discussion) or at least should be toned down.

- In lines 125-127 the authors emphasise that commensals can have T3SS effectors similar to those in pathogens despite being in different orders, pointing towards HGT, but the examples given actually are two species that can be pathogenic (*E. albertii* and *Y. enterocolitica*), which does not help in illustrating this point. To confuse matters further, the authors then say that effectors in commensals are distinct from those in pathogens, which sounds a bit contradictory (line 132-134). The figures seem to largely support the latter conclusion, and so I would appreciate if the text was more clear.
- How many GFP+ cells are counted to obtain the MFI in the Icam expression experiments? A representative dot plot would be nice to show the populations.

Reviewer #3 (Remarks to the Author):

The paper is an exploration of T3SS effector-human protein interactions, specifically focussing on commensal Proteobacteria. With 1,263 interactions discovered between 289 effectors and 430 human proteins, the study is an opportunity to uncover some exciting new biology, however it falls short in this due to its lack of sensitive sequence/structure searching for protein function prediction, and lack of in depth molecular exploration of any new effector-human protein pair representatives. There is a problem with uninformative figures in the main text, and over-interpretation of data. My detailed comments follow below.

Major:

1. I am unsure about the initial premise that there is a clear distinction between pathogenic vs non-pathogenic bacteria. Some pathogens can be opportunistic, or condition-specific. Since the distinction is blurry, it seems a reasonable expectation that commensals can also have T3SSs for supporting growth in certain circumstances.
2. The authors acknowledge that the algorithm they use may miss systems in certain bacteria (lines 97-98). Biases of the algorithm may also be why they found many hits in the well studied *Escherichia* (line 99). So given that the results are unlikely to be exhaustive, I'm not sure Fig 1 is useful as a main figure. There is nothing wrong with such an approach for finding candidates for subsequent detailed experimental validation and exploration to find new biology, but since this computational part of the pipeline may be underestimating T3SS presence so much, the actual numbers of hits and in which taxa may not be meaningful.
3. Figure 2A is not interesting data worth reporting in a main figure. It can be hard to clone toxins, this does not tell us anything about what they do. However it should be noted that the ones that fail to clone are probably the most important effectors as they are they are the most toxic.
4. The section "Commensal effectors are unrelated to known pathogen effectors" is problematic since only identity calculated by simple pairwise sequence alignment is used. It would be appropriate here to use tools like Jackhmmer to sensitivity cluster effectors using iterative HHM searching, HHpred to find hits in the effectors to Pfam domains, Uniprot and PDB entries, and AlphaFold/RosettaFold/ESMFold coupled with Foldseek to find similarities that are only apparent at the structure level. Horizontally transferred toxins/effectors can often be small and divergent, meaning more sensitive methods are required for finding homology. The sentence "In fact, clustering effectors by their pairwise interaction similarity identified substantial overlap outside the homology clusters (Extended Data Fig. 3), indicating that dissimilar effectors may have similar functions in the host" (line 209) suggests that some effectors classified as dissimilar might actually be similar if more appropriate sensitive methods are used.
5. In showing which "effectors" and human proteins interact, Fig 2G is an interesting panel but it is not clear what kinds of proteins we're looking at. The names of the proteins (especially the human ones) should be given here (rotate by 90 degrees for ease of reading as some names will be long). In general this is a problem throughout the paper and SI that full names of proteins are not used, making the results not very human readable. It is usual to include Uniprot protein names and IDs in the SI for human proteins picked up in screens.
6. Why is sensitivity so low? Perhaps true effector-target interactions are often transient? Isn't it also possible that many of the interactions picked up are proteins that interact for other reasons? The authors seem to assume that interactions of a predicted effector with a human protein mean that the human protein is the target of the effect. While this might be true in the majority of cases, is not clear to me that this is always so. Is it not possible that some interactions represent an immune response to inactivate effector-like proteins? And maybe some protein domains (perhaps the PDZ domain) are just 'sticky' in that they can readily bind other proteins with low specificity?
7. The authors should take advantage of state of the art structural prediction methods to try to model some of the their interactions. AlphaFold now makes it easy to predict complexes and therefore interaction interfaces.
8. Fig 3c is not possible to extract any meaning from. How are the proteins organised on the vertical axes?
9. After identifying PDZ domains as potential interactors, the paper goes off on a tangent to investigate viral PDZ domains. It is not convincing that the 'competition' they observe is biologically meaningful. PDZ domains seem to be common domains that can bind to many different kinds of proteins.
10. What are the effectors that are more prevalent in CD than healthy guts? What kinds of proteins are they? Are these some that couldn't be cloned suggesting toxicity? Which are good targets for experimental exploration? The results are presented in a way that is removed from biological discovery and makes it hard for a reader to evaluate. The effector abbreviations used in the paper and SI are not helpful without a key (in each SI table) that allows the reader to easily find the gene/protein name and protein

sequence of the effector.

Minor:

It is not properly explained how interfaces are predicted. The principle should be briefly outlined in the results.

line 65: "Especially" seems out of place here and can be deleted.

line 200: "The function of unknown proteins can often be inferred from better-studied orthologues, but convergence could also result from high sequence similarity among effectors." What is meant by convergence here? Convergent evolution? It needs better explaining.

Reviewer #4 (Remarks to the Author):

Key results

The manuscript highlights the role of Type III Secretion Systems (T3SS) in the human gut microbiome focusing on the Pseudomonadota phylum. Analysing large datasets from human gut samples demonstrates that T3SS effectors show minimal sequence similarity to known pathogenic effectors, suggesting distinct functions. The study reveals the microbiome-host interactions highlighting significant convergence of effectors on specific host proteins. The manuscript also explores the structural features, such as PDZ domains, and proposes competition between bacterial and viral proteins for host binding partners. The manuscript further investigates the contribution of effectors to immunomodulatory functions and disease pathogenesis. The authors identified enrichment in immune signalling, particularly in the NF- κ B pathway, as well as metabolism, linking to conditions like cancer and immunological diseases. Functional assays demonstrate that certain effectors can both activate and inhibit the NF- κ B pathway, influencing cell adhesion and cytokine secretion. Clinical analysis of patients with inflammatory bowel disease (IBD) suggests altered prevalence of effectors, potentially distinguishing Crohn's disease and ulcerative colitis proposing that commensal effectors' role in disease pathogenesis by modulating host cellular functions.

Validity

The presented workflow is robust and reproducible, some databases and tools have their version number, but I would suggest describing it for all (eg. ELM, IUPred, InterProScan). All source code is provided as zip file, but to track changes and facilitate collaborations, I would suggest storing them in a version control system (e.g. Git).

Significance

The study provides a new insight into the immunomodulatory effect of commensal T3SS effectors. I would suggest to emphasise better the difference between pathogenic and commensal T3SS effectors. The authors compared them in terms of sequence similarity but it would be great to have a look at the differences from a network perspective (eg. convergence) and comparing the immunomodulatory effect of commensal effectors to pathogens according to the literature. The 'Discussion' section is heavily focused on the disease context and does not conclude the rest of the analysis carried out in the study (eg. the microbe-host interactome, convergence analysis or interface prediction).

Data and methodology

The manuscript describes a robust multi-layered analysis including experimental and in silico approaches in a well-organised, structured way. However, in the 'Meta-interactome mapping' method section, further clarification or contextualisation could improve the understanding, especially for readers less familiar with certain methodologies.

The utilisation of the Recon3D human metabolic model for metabolic subsystem analysis is appropriate. However, the assumptions and limitations of this model should be acknowledged for a more comprehensive interpretation of the metabolic findings.

Finally, limitations-related to this comprehensive analysis should be discussed in the 'Discussion' section.

Analytical approach

In general, the authors have employed a variety of statistical and computational methods to prove a robust and comprehensive workflow. However, the protein-protein interaction interface analysis highlights weaknesses in the computational pipeline. The domain-motif interaction based approach gives a directionality for the microbe-host interactions and the current study describes how domains on human proteins bind to motifs on bacterial effectors. But the authors aimed to demonstrate the impact of effectors on host proteins, hence the analysis does not answer the proposed goals. Besides, the workflow uses an earlier version of the IUPred tool that should be updated and the applied cut-off (0.2) is inappropriate to predict disordered regions on protein sequences.

As a computational biologist, assessing the experimental protocols and methods is outside the scope of my expertise.

Clarity and context

The manuscript provides a detailed description of methods and results referencing specific studies, databases, and tools. While most of the abbreviations are explained, others could be clarified to enhance understanding (eg. HU, HMM, BD, DMEM, HEK 293 - mention this is a cell line). One potential improvement could be the inclusion of a brief summary or overview at the beginning of the methods section to provide readers with a high-level understanding of the overall approach before delving into the details. Additionally, the integration of visual aids, such as flowcharts or diagrams, could enhance the clarity of the complex computational and experimental processes described in the methods.

References

The manuscript cites 110 articles covering the field of host-microbe interaction research, however there is a need for formal improvements: the citation for BastionHub is duplicated (22 and 79); in line 1152, there is a missing tab; doi IDs should be

formatted properly, either highlighting them as a link or using only the ID.

Suggested improvements

Major improvements

The authors explored the interface between microbe-host proteins connecting bacterial motifs to human domains. As the domain-motif interactions are directed (the domain is binding to the target SLiM), the current analysis describes how human proteins are able to bind to bacterial effectors. To explore the effect of commensal effectors on human proteins, I suggest redoing the interface analysis considering the bacterial domains connected to human SLiMs.

Besides, I suggest using one of the newer versions of IUPred (2A or 3) as they provide a more precise prediction of disordered regions through defining two scores.

The authors defined the cut-off for 0.2 in the manuscript while the IUPRED tool describes the following: 'Residues with a predicted score above 0.5 are considered disordered, while residues with lower scores considered to be ordered.' Hence, the currently used 0.2 cut-off is not appropriate to predict disordered regions, the score should be at least 0.5 for both IUPred and ANCHOR values.

Considering and describing the limitations of the study in the 'Discussion' section.

Minor improvements

The 'Main' section is currently very short and it provides a brief background about the topic

The 'Result' header is missing

'From 16,179 Pseudomonadota MAG bins with high or intermediate genome quality' - define what is high/intermediate quality (line 92-93)

Missing '(' in line 438

Provide more details on the previously developed Y2H screening pipeline

Insert a paragraph about the workflow summary/create a workflow figure

Extend the 'Discussion' section concluding all the significant results not only focusing on disease-related importance of the commensal effectors.

The authors mention 1638 effectors derived from BastionHub database (in line 117), later 1195 gathered effectors are mentioned (in line 474) - Please clarify this.

Description of 'odd ratio' is missing from the text

The authors used their mimicINT tool for interface prediction that has not been published yet. Why is this the best option for interface prediction? Is not there any similar tool that has been already published?

Provide version number for all tools and databases (eg. ELM, InterProScan)

The authors compared commensal and pathogen effectors in terms of sequence similarity but it would be great to have a look at the differences from a network perspective (eg. convergence) and comparing the immunomodulatory effect of commensal effectors to pathogens according to the literature.

The assumptions and limitations of the Recon3D metabolic model should be acknowledged for a more comprehensive interpretation of the metabolic findings.

While some of the abbreviations are explained, others could be clarified to enhance understanding (eg. HU, HMM, BD, DMEM, HEK 293 - mention this is a cell line).

BastionHub is cited twice (ref 22 and 79).

In line 1152, there is a missing tab by the citation.

doi IDs should be formatted properly, either highlighting them as a link or using only the ID.

Comments on figures

There is no need for Fig. 1c as the two numbers have been mentioned in the text

Describe the colour legend in Fig. 1d (blue-grey scale)

Reference for Fig. 2a is missing in the text

Fig. 2g is consisting of two panels that is confusing

What is 'HU' on Fig. 3e? Also, the yellow border is less visible

Fig. 3f is small, the 'FP' should be described in the figure caption

I would suggest changing the layout for Fig. 4d to a list/table instead of the current circle arrangement

Comments on Supplementary tables

As the tables consist of several worksheets related to different parts of the analysis, I would suggest to refer to the exact worksheet instead of the table number

Extended Data Table 2A-B describes the predicted effectors from collected reference strains and MAGs. Where is the HBC/UHGG-related data?

Points that should be clarified

Why did the authors select other phyla in the first part of the analysis from HBC and UHGG?

How did the authors select the commensal Pseudomonadota strains?

How did the authors select the pathogens from the BastionHub database?

What does 'mostly confidently predicted' mean in line 114 ("3,002 effector candidates from the 44 reference strains that were most confidently predicted by all tools")?

The authors set up a cut-off for 50% completeness and less than 5% contamination but how reliable are the computational tools of T3SS prediction when MAGs are incomplete or contaminated?

How representative are the collected 44 reference genomes and 770 MAGs of the overall gut microbiome?

How does the observed convergence in commensal effectors compare to the previously noted convergence in pathogenic effectors?

Version 1:

Reviewer comments:

Reviewer #1

(Remarks to the Author)

The authors have professionally and adequately addressed my previous comments. I have no further comments.

Reviewer #2

(Remarks to the Author)

I am content with the additional experiments performed by the authors to provide more biological relevance, and overall I think the manuscript is improved, with a thought-provoking discussion and providing a wealth of interesting information. However, I find that for these useful data to be accessible to the readers, some minor changes could be made:

- I found reading the manuscript difficult because it was often not intuitive to know what results/figures the text was referencing. Except for the main figures, when referring to extended data or the supplementary tables, only the ED Fig or Table is mentioned, without specifying the panel/sheet. I do understand there is a summary sheet in each of the supplementary tables, but even looking at that is sometimes not very informative as, for example, in supplementary table 6 not straight-forward to understand when you are referring to your candidate commensal effectors, and when you are referring to the known "pathogenic" effectors, thus I find it impossible to follow where this is shown. Another example is line 354, where you refer to table 5 but I do not know where to look for the results. Overall I think that specifying which ED fig panel or table sheet you are alluding to would allow readers to understand the study better and find the information they are interested in more easily.
- I think the data showing effector translocation really strengthens the study; however I would really appreciate if the "known pathogenic" effector names (eg Q99PZ6 is OspG from Shigella) were mentioned in the figure (fig 1e) or legend to facilitate reading
- For figure 2e, could the name of the control effector (espG) be in the figure or legend (preferably in the figure and then use the uniprot ID in the legend)? The uniprot identifier is not very useful. And maybe indicate it's your control as this is not in the legend I believe. Also, this is representative of how many repeats?
- I am confused by the text in lines 250-254: This refers to intriguing data for further experimentation, but it's difficult to see where the results are (if anywhere)? I know convergence is in Supp table 4, Sheet 4O but, as far as I can see, this doesn't show the pathogenic effectors?

Very minor errors to be corrected:

- Wrong figure 1 callouts in line 139 onwards, as I think the order has been reshuffled: Fig 1c should be 1b (line 139), 1d to 1c (lines 145, 150) and 1b to 1d (line 154)
- Line 342 onwards, callouts to supplementary table 7 I think are for supplementary table 6?. And supplementary table 8 (line 384 onwards) I believe is 7 (there is no 8).

Reviewer #3

(Remarks to the Author)

The first thing I want to mention is that, disappointingly, the response text to reviewer 3 (me) contains an assumption that the reviewer is male: "We thank the reviewer for this suggestion and have followed his advice". It is incredibly discouraging that in 2025 the default assumption remains that reviewers are male.

The authors have gone to great lengths in the last two years to improve the paper, and it is potentially much stronger in many areas and could become a useful resource. However, as with the first version I think there is a significant problem with interpretability of figures and explanation of data, which has really hindered review and made the whole package hard to judge. I give some examples below.

Fig 1b: there is not enough information on the graph or in the legend to interpret this. What is the histogram above vs the one below?

Fig 1c: what are those structures, mini networks and donut graphs? There are no labels and the legend does not explain at all.

Fig 3d: what are those structures? they are not mentioned in the text

It is also unclear on what basis a bacterium is classified as commensal or pathogenic. Is there a database somewhere that lists strains and pathogenicity?

My concern about biological insight remains. There is little mention of functional annotation of the effectors in the text. Are there over represented domains in the effectors? I also didn't manage to find effector sequences in the SI, but maybe they are there somewhere.

minor:

line 101: "that were isolated by the human microbiome project and other efforts." I feel this needs a reference

fig 1b is cited after 1c

Reviewer #4

(Remarks to the Author)

The revision and the point-by-point response letter properly and comprehensively resolved all but one previously raised comments. As for R4.7, to clarify the original comment: in some cases, host proteins have enzymatic domains, such as proteases that can cleave SLIMs on bacterial proteins. In these scenarios the direction is host-> microbe, and not the other way around - hence the comment on the importance on directionality in the proposed workflow. All other points are fine.

Decision Letter:

3rd September 2025

Dear Professor Falter-Braun,

Thank you for your patience while your manuscript "Meta-interactome network reveals human immune modulation by commensal T3SS- effectors" was under peer-review at Nature Microbiology. It has now been seen by 4 referees, whose comments you will find at the of this email. You will see from their comments below that while they find your work of interest, some important points are raised. We are very interested in the possibility of publishing your study in Nature Microbiology, but would like to consider your response to these concerns in the form of a revised manuscript before we make a final decision on publication.

In particular, we recommend that you modify the manuscript and the figures to make them more accessible and clear to the readers. Additionally, we recommend that you clarify definitions of 'mild' pathogen and commensal strains used in the manuscript. Some additional analysis to address R3's concern of biological relevance should also be added. The rest referees' reports are clear and the remaining issues should be straightforward to address.

If you have not done so already please begin to revise your manuscript so that it conforms to our Article format instructions at <http://www.nature.com/nmicrobiol/info/final-submission/>

The usual length limit for a Nature Microbiology Article is six display items (figures or tables) and 3,000 words. We have some flexibility, and can allow a revised manuscript at 3,500 words, but please consider this a firm upper limit. There is a trade-off of ~250 words per display item, so if you need more space, you could move a Figure or Table to Supplementary Information.

Some reduction could be achieved by focusing any introductory material and moving it to the start of your opening 'bold' paragraph, whose function is to outline the background to your work, describe in a sentence your new observations, and explain your main conclusions. The discussion should also be limited. Methods should be described in a separate section following the discussion, we do not place a word limit on Methods.

Nature Microbiology titles should give a sense of the main new findings of a manuscript, and should not contain punctuation. Please keep in mind that we strongly discourage active verbs in titles, and that they should ideally fit within 90 characters each (including spaces).

Please include a data availability statement as a separate section after Methods but before references, under the heading "Data Availability". This section should inform readers about the availability of the data used to support the conclusions of your study. This information includes accession codes to public repositories (data banks for protein, DNA or RNA sequences, microarray, proteomics data etc...), references to source data published alongside the paper, unique identifiers such as URLs to data repository entries, or data set DOIs, and any other statement about data availability. At a minimum, you should include the following statement: "The data that support the findings of this study are available from the corresponding author upon request", mentioning any restrictions on availability. If DOIs are provided, we also strongly encourage including these in the Reference list (authors, title, publisher (repository name), identifier, year). For more guidance on how to write this section please see: <http://www.nature.com/authors/policies/data/data-availability-statements-data-citations.pdf>

To improve the accessibility of your paper to readers from other research areas, please pay particular attention to the wording of the paper's opening paragraph, which serves both as an introduction and as a brief, non-technical summary in about 150 words. If, however, you require one or two extra sentences to explain your work clearly, please include them even if the paragraph is over-length as a result. The opening paragraph should not contain references. Because scientists from other sub-disciplines will be interested in your results and their implications, it is important to explain essential but specialised terms concisely. We suggest you show your summary paragraph to colleagues in other fields to uncover any problematic concepts.

If your paper is accepted for publication, we will edit your display items electronically so they conform to our house style and will reproduce clearly in print. If necessary, we will re-size figures to fit single or double column width. If your figures contain several parts, the parts should form a neat rectangle when assembled. Choosing the right electronic format at this stage will speed up the processing of your paper and give the best possible results in print. We would like the figures to be supplied as vector files - EPS, PDF, AI or postscript (PS) file formats (not raster or bitmap files), preferably generated with vector-graphics software (Adobe Illustrator for example). Please try to ensure that all figures are non-flattened and fully editable. All images should be at least 300 dpi resolution (when figures are scaled to approximately the size that they are to be printed at) and in RGB colour format. Please do not submit Jpeg or flattened TIFF files. Please see also 'Guidelines for Electronic Submission of Figures' at the end of this letter for further detail.

Figure legends must provide a brief description of the figure and the symbols used, within 350 words, including definitions of any error bars employed in the figures.

When submitting the revised version of your manuscript, please pay close attention to our [href="https://www.nature.com/nature-research/editorial-policies/image-integrity">Digital Image Integrity Guidelines. and to the following points below:](https://www.nature.com/nature-research/editorial-policies/image-integrity)

EXTENDED DATA FIGURES

Please include a statement before the acknowledgements naming the author to whom correspondence and requests for materials should be addressed.

Finally, we require authors to include a statement of their individual contributions to the paper -- such as experimental work, project planning, data analysis, etc. -- immediately after the acknowledgements. The statement should be short, and refer to authors by their initials. For details please see the Authorship section of our joint Editorial policies at http://www.nature.com/authors/editorial_policies/authorship.html

* include a point-by-point response to any editorial suggestions and to our referees. Please include your response to the editorial suggestions in your cover letter, and please upload your response to the referees as a separate document.

* ensure it complies with our format requirements for Letters as set out in our guide to authors at www.nature.com/nmicrobiol/info/gta/

* state in a cover note the length of the text, methods and legends; the number of references; number and estimated final size of figures and tables

* resubmit electronically if possible using the link below to access your home page:

Link Redacted

*This url links to your confidential homepage and associated information about manuscripts you may have submitted or be reviewing for us. If you wish to forward this e-mail to co-authors, please delete this link to your homepage first.

Please ensure that all correspondence is marked with your Nature Microbiology reference number in the subject line.

Nature Microbiology is committed to improving transparency in authorship. As part of our efforts in this direction, we are now requesting that all authors identified as 'corresponding author' on published papers create and link their Open Researcher and Contributor Identifier (ORCID) with their account on the Manuscript Tracking System (MTS), prior to acceptance. This applies to

primary research papers only. ORCID helps the scientific community achieve unambiguous attribution of all scholarly contributions. You can create and link your ORCID from the home page of the MTS by clicking on 'Modify my Springer Nature account'. For more information please visit www.springernature.com/orcid.

We hope to receive your revised paper within three weeks. If you cannot send it within this time, please let us know.

Yours sincerely,

Reviewers Comments:

Reviewer #1 (Remarks to the Author):

The authors have professionally and adequately addressed my previous comments. I have no further comments.

Reviewer #2 (Remarks to the Author):

I am content with the additional experiments performed by the authors to provide more biological relevance, and overall I think the manuscript is improved, with a thought-provoking discussion and providing a wealth of interesting information. However, I find that for these useful data to be accessible to the readers, some minor changes could be made:

- I found reading the manuscript difficult because it was often not intuitive to know what results/figures the text was referencing. Except for the main figures, when referring to extended data or the supplementary tables, only the ED Fig or Table is mentioned, without specifying the panel/sheet. I do understand there is a summary sheet in each of the supplementary tables, but even looking at that is sometimes not very informative as, for example, in supplementary table 6 not straight-forward to understand when you are referring to your candidate commensal effectors, and when you are referring to the known "pathogenic" effectors, thus I find it impossible to follow where this is shown. Another example is line 354, where you refer to table 5 but I do not know where to look for the results. Overall I think that specifying which ED fig panel or table sheet you are alluding to would allow readers to understand the study better and find the information they are interested in more easily.
- I think the data showing effector translocation really strengthens the study; however I would really appreciate if the "known pathogenic" effector names (eg Q99PZ6 is OspG from Shigella) were mentioned in the figure (fig 1e) or legend to facilitate reading
- For figure 2e, could the name of the control effector (espG) be in the figure or legend (preferably in the figure and then use the uniprot ID in the legend)? The uniprot identifier is not very useful. And maybe indicate it's your control as this is not in the legend I believe. Also, this is representative of how many repeats?
- I am confused by the text in lines 250-254: This refers to intriguing data for further experimentation, but it's difficult to see where the results are (if anywhere)? I know convergence is in Supp table 4, Sheet 4O but, as far as I can see, this doesn't show the pathogenic effectors?

Very minor errors to be corrected:

- Wrong figure 1 callouts in line 139 onwards, as I think the order has been reshuffled: Fig 1c should be 1b (line 139), 1d to 1c (lines 145, 150) and 1b to 1d (line 154)
- Line 342 onwards, callouts to supplementary table 7 I think are for supplementary table 6?. And supplementary table 8 (line 384 onwards) I believe is 7 (there is no 8).

Reviewer #3 (Remarks to the Author):

The first thing I want to mention is that, disappointingly, the response text to reviewer 3 (me) contains an assumption that the reviewer is male: "We thank the reviewer for this suggestion and have followed his advice". It is incredibly discouraging that in 2025 the default assumption remains that reviewers are male.

The authors have gone to great lengths in the last two years to improve the paper, and it is potentially much stronger in many areas and could become a useful resource. However, as with the first version I think there is a significant problem with interpretability of figures and explanation of data, which has really hindered review and made the whole package hard to judge. I give some examples below.

Fig 1b: there is not enough information on the graph or in the legend to interpret this. What is the histogram above vs the one below?

Fig 1c: what are those structures, mini networks and donut graphs? There are no labels and the legend does not explain at all.

Fig 3d: what are those structures? they are not mentioned in the text

It is also unclear on what basis a bacterium is classified as commensal or pathogenic. Is there a database somewhere that lists strains and pathogenicity?

My concern about biological insight remains. There is little mention of functional annotation of the effectors in the text. Are there over-represented domains in the effectors? I also didn't manage to find effector sequences in the SI, but maybe they are there somewhere.

minor:

line 101: "that were isolated by the human microbiome project and other efforts." I feel this needs a reference

fig 1b is cited after 1c

Reviewer #4 (Remarks to the Author):

The revision and the point-by-point response letter properly and comprehensively resolved all but one previously raised comment. As for R4.7, to clarify the original comment: in some cases, host proteins have enzymatic domains, such as proteases that can cleave SLiMs on bacterial proteins. In these scenarios the direction is host-> microbe, and not the other way around - hence the comment on the importance on directionality in the proposed workflow. All other points are fine.

Version 2:

Reviewer comments:

Reviewer #4

(Remarks to the Author)

In this revision round, Young et al further improved their manuscript. This time, I specifically looked at the responses and actions for the concerns made by Reviewer 3 and myself.

The responses to Reviewer 3's previous comments are fine, but not all made fully to the manuscript. Figure legends and description of the results shown on the figures have been added but the responses for Rev 3.6 and Rev 3.7 are more detailed in the response letter and less in the manuscript. This actually points to the problem of the Discussion section of the manuscript and less the Results section anymore. The Discussion section is currently too short and not covering key aspects, including gained biological insight, constraints and future opportunities. The limitations at the end of the current manuscript are very superficial and short. I recommend the authors to extend the Discussion section substantially with more details prompted by the comments of the reviewers, especially Reviewer 3, and provide a Discussion that both highlights better their nice work and its limitations.

As for the response for my previous comment on the directionality of the bacterial-host interactions, I cannot accept the answer. I agree with the authors that just using structural information, we often cannot determine the direction. However, in an interaction between a catalytic domain (kinase, protease, etc) and its target motif, it is highly common that the flow of signal is from the domain to the protein having the motif. Even for scaffold proteins with motifs and other proteins with binding domains, the direction of signal flow can be inferred, as such interactions influence the activity of the proteins bound by the scaffold. All this should be clarified and considered in the manuscript.

Most importantly, the authors' response that "The reason why we did not investigate human SLiMs binding to bacterial SLiM binding domains was merely that the bacterial effectors do not contain domains known to bind SLiMs." is unfortunately incorrect and scientifically not acceptable. For bacterial effectors with domains binding SLiMs, here are a few examples:

<https://www.science.org/doi/10.1126/science.1158160>, <https://pmc.ncbi.nlm.nih.gov/articles/PMC4715502>. Therefore, the authors should clarify the focus of their work, detail the selective nature of their workflow to direction, and further discuss the limitations in the Discussion section.

Additional, minor comments:

- Line 336-337. "Among immune diseases, CD (nominal $P = 8.5 \times 10^{-5}$, Fisher's exact test) and IBD (nominal $P = 0.0008$, Fisher's exact test) were enriched, but not ulcerative colitis (UC) (Fig. 4d, Supplementary Table 23)." As discussed later in the manuscript, CD and UC are both IBDs. Therefore, this sentence needs to be rephrased, perhaps first discuss CD vs UC, and then clarify and explain that for IBD still enrichment was found.
- Line 366. "...we analyzed a metagenome study with > 800 IBD patients and > 300 healthy controls (ref47)." Specify the number of IBD patients and healthy controls – "more than 800" / "more than 300" are not clear enough.
- Throughout the manuscript, many resources or communities are being written with all lower case, such as "gene ontology", "open targets", etc. Please have uppercase letter for the first words of these entities.

Decision Letter:

Our ref: NMICROBIOL-23092521B

22nd October 2025

Dear Dr. Falter-Braun,

Thank you for submitting your revised manuscript "Meta-interactome network reveals human immune modulation by commensal T3SS-effectors" (NMICROBIOL-23092521B). It has now been seen by the original referees and their comments are below. The reviewers find that the paper has improved in revision, and therefore we'll be happy in principle to publish it in Nature Microbiology, pending minor revisions to satisfy the referees' final requests and to comply with our editorial and formatting guidelines.

Thank you again for your interest in Nature Microbiology Please do not hesitate to contact me if you have any questions.

Sincerely,

Reviewer #4 (Remarks to the Author):

In this revision round, Young et al further improved their manuscript. This time, I specifically looked at the responses and actions for the concerns made by Reviewer 3 and myself.

The responses to Reviewer 3's previous comments are fine, but not all made fully to the manuscript. Figure legends and description of the results shown on the figures have been added but the responses for Rev 3.6 and Rev 3.7 are more detailed in the response letter and less in the manuscript. This actually points to the problem of the Discussion section of the manuscript and less the Results section anymore. The Discussion section is currently too short and not covering key aspects, including gained biological insight, constraints and future opportunities. The limitations at the end of the current manuscript are very superficial and short. I recommend the authors to extend the Discussion section substantially with more details prompted by the comments of the reviewers, especially Reviewer 3, and provide a Discussion that both highlights better their nice work and its limitations.

As for the response for my previous comment on the directionality of the bacterial-host interactions, I cannot accept the answer. I agree with the authors that just using structural information, we often cannot determine the direction. However, in an interaction between a catalytic domain (kinase, protease, etc) and its target motif, it is highly common that the flow of signal is from the domain to the protein having the motif. Even for scaffold proteins with motifs and other proteins with binding domains, the direction of signal flow can be inferred, as such interactions influence the activity of the proteins bound by the scaffold. All this should be clarified and considered in the manuscript.

Most importantly, the authors' response that "The reason why we did not investigate human SLiMs binding to bacterial SLiM binding domains was merely that the bacterial effectors do not contain domains known to bind SLiMs." is unfortunately incorrect and scientifically not acceptable. For bacterial effectors with domains binding SLiMs, here are a few examples: <https://www.science.org/doi/10.1126/science.1158160>, <https://pmc.ncbi.nlm.nih.gov/articles/PMC4715502>. Therefore, the authors should clarify the focus of their work, detail the selective nature of their workflow to direction, and further discuss the limitations in the Discussion section.

Additional, minor comments:

- Line 336-337. "Among immune diseases, CD (nominal $P = 8.5 \times 10^{-5}$, Fisher's exact test) and IBD (nominal $P = 0.0008$, Fisher's exact test) were enriched, but not ulcerative colitis (UC) (Fig. 4d, Supplementary Table 23)." As discussed later in the manuscript, CD and UC are both IBDs. Therefore, this sentence needs to be rephrased, perhaps first discuss CD vs UC, and then clarify and explain that for IBD still enrichment was found.
- Line 366. "...we analyzed a metagenome study with > 800 IBD patients and > 300 healthy controls (ref47)." Specify the number of IBD patients and healthy controls – "more than 800" / "more than 300" are not clear enough.
- Throughout the manuscript, many resources or communities are being written with all lower case, such as "gene ontology", "open targets", etc. Please have uppercase letter for the first words of these entities.

Version 3:

Decision Letter:

11th December 2025

Dear Pascal,

Although it took quite some back-and-forth, I am very happy that the long journey of this interesting manuscript has come to a successful end. I am pleased to accept your Article "Effector–host interactome map links type III secretion systems in healthy gut microbiomes to immune modulation" for publication in Nature Microbiology. Congratulations and thank you for choosing Nature Microbiology!

Authors may need to take specific actions to achieve compliance with funder and institutional open access mandates. If your research is supported by a funder that requires immediate open access (e.g. according to [a href="https://www.springernature.com/gp/open-science/plan-s-compliance"> Plan S principles](https://www.springernature.com/gp/open-science/plan-s-compliance) or the [a href="https://www.springernature.com/gp/open-science/us-federal-agency-compliance"> NIH public access policy](https://www.springernature.com/gp/open-science/us-federal-agency-compliance)) then you should select the gold OA route, and we will direct you to the compliant route where possible. Because authors warrant under our subscription licensing terms that they haven't committed to licensing any version of their article under a licence inconsistent with the terms of our agreement – including the applicable embargo period – publication under the subscription model isn't suitable for authors whose funders require no embargo.

An online order form for reprints of your paper is available at [a href="https://www.nature.com/reprints/author-reprints.html">https://www.nature.com/reprints/author-reprints.html](https://www.nature.com/reprints/author-reprints.html). All co-authors, authors' institutions and authors' funding agencies can order reprints using the form appropriate to their geographical region.

Best wishes and enjoy the festive season,

P.S. Click on the following link if you would like to recommend Nature Microbiology to your librarian
<http://www.nature.com/subscriptions/recommend.html#forms>

** Visit the Springer Nature Editorial and Publishing website at http://editorial-jobs.springernature.com?utm_source=ejP_NMicro_email&utm_medium=ejP_NMicro_email&utm_campaign=ejp_NMicro for more information about our career opportunities. If you have any questions please click [here](mailto:editorial.publishing.jobs@springernature.com).**

Review response for

"A gut meta-interactome map reveals modulation of human immunity by microbiome effectors"

We thank the reviewers and editorial staff for their thorough and constructive review. We have addressed all points with extensive new experimental data and analyses and responded to all conceptual questions and requests for clarification. The manuscript was modified substantially to encompass the new data and the deeper discussion. All new analyses support our initial conclusions.

We hope that the editor and reviewers will agree that these changes have clarified and substantially improved the manuscript. Below we respond to each of the points raised and describe how our revised manuscript was modified during the revision.

Editor (Remarks to the Author):

Thank you for your patience while your manuscript "A gut meta-interactome map reveals modulation of human immunity by microbiome effectors" was under peer-review at Nature Microbiology. It has now been seen by 4 referees, whose expertise and comments you will find at the end of this email. Although they find your work of some potential interest, they have raised a number of concerns that will need to be addressed before we can consider publication of the work in Nature Microbiology.

0.0.a. In particular, we ask you to address the technical concerns raised by Referees 3 and 4.

AUTHOR RESPONSE

We have addressed the technical concerns. All statistical analysis are now done by averaging the technical repeats and doing statistics only on the biological repeats.

0.0.b. Additionally, as suggested by all referees, please provide more in-depth comparison between commensal and pathogen effectors.

AUTHOR RESPONSE

In response to the reviewer feedback, we have now substantially expanded the comparison between effectors from pathogen and commensal strains on multiple levels. First, as suggested by reviewer 3, we have conducted a jackhmmer analysis as a more sensitive alternative than BLAST to detect sequence similarities. This analysis revealed additional similarities between commensal and pathogen effectors but still only for a minority of the effectors from commensal strains. In addition, as suggested by reviewers 1 and 3 (REV1.2; REV3.4), we have compared the three-dimensional structures of commensal and pathogen effectors using AlphaFold2 structural predictions followed by identification of structure similarity using Foldseek. We find a strong enrichment of homotypic effector groupings, i.e. effectors from pathogens have similar folds and effectors from commensal microbes also have similar folds, whereas only a small subset of folds is shared between effectors from beneficial and commensal strains. We believe this reinforces the differences between effector complements in bacterial strains isolated from healthy human guts compared to pathogen effectors.

On the network level, following suggestions by reviewers 1, 3 and 4, we compared the human proteins interacting with commensal and pathogen effectors respectively, which we find overlap significantly. Finally, we compare the processes targeted by effectors from both groups and, again, find commonalities, but also differences. These last two results for interaction patterns must be interpreted within the limitations of the existing literature, which we discuss.

Altogether, we believe these additional analyses strongly support our initial conclusion that the non-pathogenic strains of the human gut have functional effector complements (see next section), that are

profoundly different from the effectors of known pathogens.

Detailed descriptions of the analysis and text modifications are detailed below in the responses to reviewer questions.

0.0.b. We also ask you to provide experimental validation for at least one strain, to show that these effectors are expressed and translocated via a functional T3SS. This could be an in vitro validation.

AUTHOR RESPONSE

We agree that demonstration of secretion and injection of effectors by secretion systems is a critical set of experiments, and we have conducted these important controls.

We detail the experiments to demonstrate expression and injection of candidate effector proteins into human host cells in response to REV2.2. Briefly, we have picked about 100 diverse effectors from our cloned set and tested if these can be injected using a known and functional T3SS of the human pathogen *Salmonella* Typhimurium using a stringently controlled setup. In this assay 32 of the tested 97 effectors coming from many different strains were injected into human cells. Of the 6 control effectors, mostly from *Salmonella* Typhimurium, but also other pathogens, 4 were translocated into human cells. In a second set of experiments, we demonstrate for one of the commensal strains, *Edwardsiella tarda*, that this strain isolated from healthy human guts can inject 3 of the 4 tested effectors into human cells. Importantly, 2 of the 3 effectors were negative in the *Salmonella* system indicating that indeed host specific factors are important for successful translocation. For two of the other tested strains, we observed sporadic high signals, but these were inconsistent, possibly because of only stochastic activation of the T3SS in non-pathogens.

Together, we believe these additional experiments support our conclusion that many bacterial strains in healthy human guts have functional T3SS systems.

Reviewer #1 (Remarks to the Author):

This manuscript reports a comprehensive story of the identification of type III effectors from the human microbiome and their functional association with human disease. The authors detected T3SS and type III effectors in the genomes and metagenomes of human commensal bacteria belonging to Pseudomonadota (in particular Gammaproteobacteria). These commensal type III effectors are sequence-unrelated to pathogen effectors. A large-scale Y2H screen identified human proteins interacting with commensal effectors. Many of those effectors target proteins related to disease and immunity, to a certain degree convergently and with specificity. In addition, they validated the implication of some of commensal effectors in the modulation of immune responses. Interestingly, they also found that the prevalence of effectors is different in metagenomes of disease and healthy patients. Based on the presented data, they conclude that type III effectors from the human microbiome modulate immunity, thereby affecting human disease.

This manuscript is a novel and interesting work and is very well presented and written, with a huge amount of data and analyses. As the function of commensal type III effectors is unknown in both human and plant systems, it conceptually advances our understanding of the function of bacterial type III effectors in human microbiome, and in general microbiome. Nevertheless, I have two key suggestions and other comments that would further improve this manuscript.

(Major points)

REV 1.1 *While the authors showed that commensal effectors are sequence-unrelated to known pathogen effectors, it remains unclear whether their functions are similar or different compared to pathogen effectors as many of microbial effectors have been shown to be sequence-unrelated anyway. Thus, it is important to compare targets of commensal and pathogen type III effectors. This will be widely influential as the difference in properties of commensal and pathogenic bacteria receives huge attention in host-microbe interactions including plant-bacterial interactions. The authors mention that functions of commensal effectors are outside of the pathogen lifestyle (line 134). Yet, for CD, it is suggested that commensal effectors promote disease development (line 425). Thus, an important question is that functions of effectors from pathogenic and commensal bacteria are fundamentally different or similar as they did in Osborne et al 2023. With either result, this will be strongly influential to the community. The authors could perform Y2H with pathogen effectors. Alternatively, they may analyze already existing data with deep discussion.*

AUTHOR RESPONSE

We agree that a comparison of the processes and functions targeted by effectors from pathogens and commensals is an important question. Unfortunately, generating a systematic pathogen-host interactome map for one or a few pathogens that are directly comparable to our presented data requires additional funding. However, this is central to our interest, and we are seeking funding for this.

In the meantime, existing data are available. Unfortunately, these data are largely produced by hypothesis driven approaches or by few systematic studies that used a different experimental setup. As hypothesis-driven data are influenced by researcher focus, the small-scale literature data provide a very biased picture of the underlying biology^{1,2}. This contrasts with our analysis in Osborne *et al.*³, for which we had previously collected systematic pathogen-plant interaction data using an identical experimental setup. This enabled us to exclude inspection bias and experimental differences, and the data could be analyzed in a comparative and statistically meaningful way. This is not possible here.

Keeping these limitations in mind, we have gathered existing pathogen effector-host protein interaction data for human and looked for overlap in the targets, similarities in targeted pathways and conducted a function enrichment analysis. To obtain interaction data, we queried the IMEx consortium protein interaction databases via the PSICQUIC web service to identify interactions between effector proteins encoded in Type III Secretion System (T3SS)-positive pathogenic strains and human proteins. Only interactions verified by experimental assays indicating physical or direct interactions were considered. We identified 235 interactions involving 73 effector proteins from 16 pathogenic bacterial strains/species and 187 human proteins, as

documented in 55 publications.

We analyzed the data for overlap in direct targets and in targeted functions.

In addition, directly to the reviewer's question about the fundamental differences, we believe the new data on structural distinction of commensal and pathogen effectors are meaningful.

EXCERPTS FROM REVISED MANUSCRIPT

To explore if these are also targeted by pathogen effectors, we extracted 265 binary high-quality interactions between 217 human proteins and 80 effector proteins from 17 pathogenic bacterial strains from IntAct⁴. We found a numerically low albeit significant number of twelve human proteins that are targeted by both groups ($P = 0.014$, Fisher's exact test; OR = 2.26), of which three are subject of convergence by commensals ($P = 0.067$, Fisher's exact test, OR = 3.37). Given the relatively small numbers, experimental differences, and the non-systematic nature of the pathogen data it is difficult to draw strong conclusions. However, given our previous observations³, the data are consistent with overlap but also lifestyle dependent specificity in the host proteins targeted by effectors from commensal and pathogenic bacteria.

For the pathogen effector targets > 200 often closely related GO terms were identified (**Supplementary Table 6**). Comparison of functions targeted by pathogens and commensals revealed commonalities, such as "NF- κ B signaling", but also functions that are not significant or absent from the pathogen targets, such as "collagen biosynthesis" and "response to muramyl dipeptide". These data reinforce the notion that is life-style dependent specificity and functional overlap in microbe-human interactions between commensals and pathogens.

REV 1.2 *The authors showed that the effector function is largely independent of sequence similarity (line 213). It is known that effectors recognized by allelic receptors can be totally sequence-unrelated (i.e. doi.org/10.1073/pnas.2307604120) but fold a similar structure. Thus, it would be interesting to utilize AlphaFold to predict the structure of some sequence-unrelated effectors with predicted similar functions.*

AUTHOR RESPONSE

We thank the reviewers for this suggestion.

To assess structural similarity between commensal and pathogen effectors we used Foldseek⁵. When available, predicted effector structures were directly downloaded from the AlphaFold DB⁶. For all effectors without structure models, a sequence comparison using BLAST was performed against all bacterial proteins with an available model in the AlphaFold DB. The best matches among those with sequence identity >95% and sequence coverage >90% were selected as representative models for said effectors. Subsequently, all effectors were clustered into groups based on pairwise structural similarity to the group's representative structure. Technically, clustering was performed for different bidirectional query coverage (qcov) (0.5, 0.7, 0.9) and E-value thresholds (0.001, 0.01, 1, 10) using Foldseek cluster's greedy set cover algorithm. To focus on full length similarity, we used results obtained with qcov = 0.9 and e-value = 0.01 as suggested⁷ for presentation and discussion. The results did not change in any major way when the query threshold was changed.

We then addressed the reviewer's question whether sequence-unrelated effectors with similar interaction profiles may fold similar structures. Of the 153 effectors considered for this analysis, 140 could be assigned

to structure clusters, corresponding to 2,452 effector pairs. Across all interaction similarity values most effector pairs (2,406; 98%) belong to different sequence and structure clusters (qcov = 0.9; e-value = 0.01; fig1.a). Of the 2,452 pairs, 42 belonged to the same sequence clusters and 41 of them were also grouped into the same structural clusters – the one pair (Yen_19, Yen_10; JI = 0.3) with sequence similarity but structural dissimilarity constitutes a border case.

All remaining pairs across all interaction similarity levels belonged to different structural clusters. Most remarkable is the pair Yen_17 and Pst_9 with Jaccard similarity of 0.67. However, also the moderate but widespread interaction similarities, which we summarized in ED Fig. 3a, are among effectors that lack both sequence and structural similarity.

EXCERPT FROM REVISED MANUSCRIPT

Conversely, clustering effectors unrelated in sequence and structure by their pairwise interaction similarity in HuMMI_{MAIN} identified substantial overlap outside the homology clusters (**ED Fig. 3**), possibly indicating that dissimilar effectors can have similar functions in the host.

(Other points)

REV 1.3 *In lines 72-74 and 397, the authors did not clearly distinguish between fungal and bacterial type III effectors. As the function of commensal T3SS and type III effectors in plants is unknown, describing this more accurately will make readers even more appreciate this work.*

AUTHOR RESPONSE

We thank the reviewer for this comment and the underlying appreciation of our work. We have clarified the respective section in the manuscript. In both sentences, we referred only to beneficial bacteria with effectors. To introduce the topic more generally, however, we have now included a reference to fungal effectors in the introduction. However, we keep the discussion focused on bacteria as it was previously.

EXCERPT FROM REVISED MANUSCRIPT

While T3SS are classically linked to pathogen virulence⁸, in plants and other systems, mutualistic microbes also use effectors to establish beneficial host associations^{3,9}.

Type III secretion systems (T3SS) are classically viewed as virulence factors, yet in other hosts, such as plants and insects, they also mediate beneficial symbioses¹⁰.

REV 1.4 *The authors detected T3SS in 44 of 77 reference strain genomes (60%), in 478 of 2272 human gut gram-negative strains (21%), and in 770 of 16179 Pseudomonadota (5%). It is important to discuss this difference so that readers can get an idea what proportion of Pseudomonadota has T3SS in the human gut microbiome. I think that this would become a good basis of future comparisons in other systems.*

AUTHOR RESPONSE

We thank the reviewer for this suggestion and agree that these results will be a good basis for any future comparisons. We agree that these different numbers need to be discussed further, especially given the difference in the quality of data each dataset represents. Both the 77 reference strains, and the 2,272 human gut strains represent isolated bacteria, while the 16,179 *Pseudomonadota* are mostly MAGs. Given that MAGs represent a consensus of similar strains, rather than any specific isolate, we believe the isolate collection better captures the true value. To provide these values in a manner which is more understandable,

we have altered the study of Gram-negative strains to the study of *Pseudomonadota* strains as stated below.

EXCERPTS FROM REVISED MANUSCRIPT

Of the 568 *Pseudomonadota* genomes, 449 (79%) have complete T3SS (**ED Fig. 1**); similar proportions have T4SS (315) and T6SS (474), both of which can also deliver effectors into host cells but also have other functions (**ED Fig. 1, Supplementary Table 1**)¹¹. Together 527 of the 568 *Pseudomonadota* genomes, i.e., 92%, have at least one host-directed secretion system.

In a global collection of 16,179 *Pseudomonadota* metagenome-assembled genomes (MAGs) with high or intermediate genome quality¹²⁻¹⁴, 770, i.e., 5%, encoded complete T3SS (**ED Fig. 1, Supplementary Table 1**).

Reviewer #2 (Remarks to the Author):

The authors present a very interesting premise, asking if commensals express and translocate T3SS effectors in a similar manner to what is seen in plants, where T3SS effectors not only are involved in pathogenesis, but can also aid in establishing mutualistic relationships. The concept is novel, and the in silico analysis performed combined with the in vitro assays (Y2H and protein domain interactions via hold up and competition assays) provide a large wealth of new information that could open new routes of study, and would be of interest to many researchers in the fields of gut health, microbiome and immunity. However, I do have a major qualm with the study in regards to biological relevance which I will detail below.

The authors identify a series of commensal strains encoding a complete T3SS from genomic data from several sources using the EffectiveDB classifier and then proceed to use machine learning tools to identify putative T3SS effectors.

REV 2.1 *From then onwards, all of these putative effectors are called effectors and are assumed to: i) be expressed by the strains and ii) be translocated by the strains. This jump I am unable to agree with, without at least some examples where these hypothetical effectors are actually being expressed and translocated into host cells.*

AUTHOR RESPONSE

We agree with the reviewer that we should have discussed more cautiously the identified proteins as candidate effectors. We have now changed this throughout the manuscript especially prior to the demonstration that many are in fact translocated in a T3SS-dependent manner. Please also take note of the additional experiments to demonstrate expression and translocation of the candidate effectors, which are described in response to REV2.2.

EXCERPT FROM REVISED MANUSCRIPT (examples):

“... 3,002 effector candidates were confidently predicted in the 44 reference strains ...”

“A key question is whether these commensal effector candidates indeed get injected into human cells by a T3SS.”

The rest of the study defines the potential relevance of these effectors via the study of their interactions with host proteins (all in vitro) and the effects of their overexpression in two different cell types.

REV 2.2 *While this provides interesting insights, we still do not know if any of the commensal strains actually express and assemble a functional T3SS, and whether the effectors are being expressed and translocated, thus reducing the relevance of these networks. I do understand that commensals can be genetically intractable, but some of the effectors are encoded by E. coli strains which, if impossible to genetically manipulate, would allow these effectors and T3SS to at least be expressed in laboratory E. coli. The absence of any biological data supporting the expression and translocation of these effectors makes the rest of the study lack a solid foundation, as there is no data supporting that these commensal strains indeed “communicate” with host cells via T3SS effectors.*

AUTHOR RESPONSE

We agree that this was a gap, which we believe we have now closed with extensive additional experiments that required additional biosafety permits and profoundly extended the time required for this revision. We now address this question in two experiments each with a slightly different focus. First, we aimed to demonstrate that a diverse sample of our effectors can get specifically injected into human cells via T3SS by a commonly used model strain for T3 secretion. Second, we aimed to demonstrate that at least one of the strains from healthy guts can deliver *bona fide* T3SS-effectors into human cells.

To address the first question, we used the mild human pathogen *S. Typhimurium*. The T3SS of this bacterium

gets easily activated upon contact with human cells thus reducing the number of variables in setting up the system and screening many effector candidates. In addition, well-characterized control effectors are available for establishing the injection system. Lastly, and critical for robust conclusions, genetically engineered strains with a specific defect in the T3SS are available that enable demonstration of specificity of injection by the T3SS¹⁵. To detect injection into human cells, we used the Nano-luc complementation system, in which one small peptide, the HiBiT-tag, is genetically fused to candidate effectors, and constitutively expressed in *S. Typhimurium*. The cultures are added to HeLa cells that stably express the complementary Nano-luc LgBiT-fragment¹⁵. Upon injection of a tagged effector the HiBiT and LgBiT fragments bind to each other and thereby reconstitute functional luciferase enzyme inside the cells. Specificity is ensured by conducting the same experiment with the *S. Typhimurium* Δ sctV strain, in which the energy providing ATPase of T3SS 1 has been deleted, and for which no signal must be detected. In addition, addition of extracellular DarkBit peptide to the cell culture, which quenches background from free proteins in the media, was added. Using the *S. Typhimurium* effector *SipA* as positive control we could demonstrate specificity and sensitivity of this system.

Subsequently, we picked 100 of the cloned candidate effectors covering the phylogenetic diversity of our strains. Six pathogen effectors from our bhLit_BM-v1 set were included as additional controls. Of the 97 effectors heterogeneously expressed in *S. Typhimurium*, 32 were successfully injected into human HeLa cells. Of the 7 control proteins of *S. Typhimurium* tested in the same assay four were translocated into HeLa cells, whereas for the other three no signal could be detected. All experiments for positive scoring strains and the controls were conducted as five biological repeats with four technical repeats each. Significance of the difference between the signal from the wild-type (*wt*) and Δ sctV strains was assessed using Wilcoxon rank sum test. For all pairs, similar expression levels of the effectors in both strains were ensured by western-blot using a Flag-tag located between the end of effector coding sequence and the HiBiT tag. While not all proteins could be detected on the WB, all detectable proteins were expressed at similar levels in both strains, such that expression differences can be excluded as underlying the observed specific signal in the *wt* compared to the mutant strain. Thus, even though all candidate effectors were heterologously expressed in *Salmonella*, the overall measured success rate of injection was similar as for the *S. Typhimurium* derived control effectors. Thus, we conclude that our effector identification pipeline reliably identified *bona fide* substrate effectors for T3SS.

In a second set of experiments, we aimed to show that one of the commensal strains does have a functional secretion system that can translocate proteins directly into the human cell. We anticipated that these experiments would be considerably more challenging due to the general lack of understanding how expression of the T3SS is regulated, especially in commensal strains. To address this, we developed expression plasmids for the commensal strains from which the effectors were cloned and tested these in the same type of injection experiments as used with *Salmonella*. An unanticipated challenge was that many strains isolated from stool samples already were resistant to multiple commonly used antibiotics, rendering some experiments impossible and requiring the generation of additional expression plasmids and reapplication for biosafety permits. Ultimately, we managed to demonstrate injection of several effectors by *Edwardsiella tarda* into human HeLa cells, thus demonstrating that this bacterial strain isolated from healthy human guts, has a functional T3SS.

EXCERPT FROM REVISED MANUSCRIPT

Injection of commensal candidate effectors into human cells

A key question is whether these commensal effector candidates indeed get injected into human cells by a T3SS. To enable functional studies, we cloned open reading frames (ORFs) for effectors from 18 bacterial strains with diverse effector sets (**Fig. 1b**, **ED Fig. 1**). Our human microbiome effector ORFeome v1 (HuMEOme_v1) contains 910 sequence-verified full-length ORFs representing 746 strain-effectors and 164 meta-effectors (**Supplementary Table 2**). Cloning failure was generally due to lack of PCR amplification in the standard conditions established for each strain without indications of toxicity. Using *Salmonella enterica*

subsp. *enterica* sv. Typhimurium (hereafter: *S. Typhimurium*) as a model we established a Nano-Luciferase based injection assay¹⁵. Here, a 11 aa Nano-Luc HiBiT-tag is C-terminally fused to candidate effectors and expressed in bacteria; a complementary LgBiT-fragment is stably expressed in HeLa cells. Upon effector injection functional Nano-Luciferase is reconstituted by protein complementation. For all tests, specificity was ensured with T3SS-defective *S. Typhimurium* strain Δ sctV. Benchmarking using six pathogen effectors demonstrated effective translocation of four of these. Of 97 tested candidate effectors from 11 different strains 32 were specifically and significantly injected (**Fig. 1e, ED Fig 1, Supplementary Table 3**). The slightly higher success rate for the positive controls is unsurprising as our test set is phylogenetically diverse and specific chaperones and the absence cofactors likely prevents translocation. Thus, although we cannot rule out that individual candidate effectors are misidentified as such, the data demonstrate that our pipeline reliably identified *bona fide* T3SS substrate effector proteins from commensal strains in healthy human guts.

To assess the functionality of the T3SS in the commensal strains, we started with all bacterial strains, for which our initial set of experiments identified at least one T3SS-injectable effector. Of the 11 strains, six could not be tested due to antibiotic resistance or the inability to obtain transformants. While the two *E. coli* strains yielded no signals, *Citrobacter youngae* and *Phytobacter massiliensis* showed occasional signals clearly above background, which possibly reflect sporadic activation of the T3SS. In contrast, for *Edwardsiella tarda*, we observed reproducible and significant injection into HeLa cells for three of the four tested effectors (**Fig. 1f, Supplementary Table 3**). Notably, only one of those, Eta_3, was also positive in the *Salmonella* system, supporting the notion that missing factors affect the injection of effectors from other species thus causing false negatives in the first experiment. Overall, these data demonstrate that functional T3SS can be identified in strains from healthy human guts that can deliver the identified effector proteins into human cells.

In regards to the statistics, to the best of my knowledge, the statistical tests carried out are appropriate and defined in the figure legends, but I do have an issue with how some data sets are considered; even if the statistical test in itself is correct, I believe its not being applied on the adequate values.

REV 2.3 *This refers to Figure 5c and d, and extended figure 5f, which all refer to either Icam1 expression or cytokine release from Caco-2 cells. The graphs mix both technical/analytical and biological repeats (denoted by the symbols and clearly shown in the extended data tables 6D, 6H, 6I) and the authors carry statistical tests using both indistinguishably, which is incorrect if the aim is to extract biologically-relevant and robust conclusions. To take into account biological variability and obtain reproducible data, I believe one should take the average of technical replicates before carrying out the statistical tests.*

AUTHOR RESPONSE

For an unrelated reason we needed to remove the ICAM1 panel. For all other analyses including the new cytokine data, we are now averaging the technical repeats and do the statistical assessment with averages of the different technical repeats. More substantially, however, we have done additional experiments with more effectors that yielded additional effects. These were done with a slightly different experimental setup and are presented as a separate panel.

EXCERPT FROM REVISED MANUSCRIPT

METHODS: Stimulations were performed in three separate wells and the supernatant was pooled after 24 hours to measure cytokines using the Human Anti-virus Response Panel V02 (BioLegend). This experiment

was repeated a total of five times to test statistical significance using Kruskal-Wallis test with Dunn's post-hoc comparisons. During cell stimulation, cell proliferation and confluence was monitored using the Incucyte S3 Live cell analysis system (Essen BioScience).

Additional comments:

REV 2.4 *To avoid just relying on basic in vitro data, an immunoprecipitation experiment could be performed to see if the overexpressed effector indeed pulls-down the endogenously-expressed prey.*

AUTHOR RESPONSE

We have now done immunoprecipitation experiments from human cells with both newly identified interaction pairs and a few previously reported controls of pathogen effectors with human proteins. However, for large scale interaction datasets, it is not very informative to merely show one or two validations. Therefore, we tested more than thirty different interaction pairs along with the corresponding controls in which the effector protein was replaced by GFP. To stem this throughput, it was necessary to express tagged versions of the human proteins and standardize the immunoprecipitation experiments. Thus, in our set-up either a flag-tagged effector, or FLAG-tagged GFP as negative control, were co-expressed with a *bona fide* human interaction partner carrying a Myc-tag. Both samples, i.e. cells expressing effector-human pair or the GFP-human control pair, were always processed in parallel. The same amount of lysate (1mg) was used for immunoprecipitation using anti-FLAG (M2) antibodies covalently linked to Sepharose beads. After binding and washing, 10µl of lysate from each sample (not normalized) and 15% of the immunoprecipitate were loaded onto an SDS gel for subsequent western blot analysis using both anti-FLAG and anti-Myc antibodies. In these experiments, three novel pairs could not be assessed because the human protein showed unspecific binding in the FLAG-GFP samples, seven additional pairs could not be tested because expression of either the effector or the human protein could not be confirmed. Of the remaining 18 test pairs eight were clearly positive, and six look positive based on a stronger band in the effector IP compared to control. Four pairs could clearly not be immunoprecipitated, and one pair was undecidable. Thus, of the 18 testable pairs a large majority could be shown to interact in human cells. Of the positive controls from the literature, one of three pairs scored positive, demonstrating that not all *bona fide* interaction pairs score positive in this setup. Thus, while it is clear from this control and our extensive previous work¹⁶ that no single assay can reproduce all interaction pairs, our data clearly show that the identified effector-host protein-protein interactions can specifically occur in the context of human cells.

The results are shown in Fig. 2, and ED Fig 2. We hope that these experiments and results address the reviewer's concern.

EXCERPT FROM REVISED MANUSCRIPT

Finally, we aimed to demonstrate that the interactions can occur within the human cell environment. For 32 pairs, including four positive control pairs, we conducted immunoprecipitation (IP) experiments from HEK293 cells using FLAG-tagged effectors and negative control FLAG-GFP as bait and detecting the Myc-tag interaction partner by western-blot. While some pairs could not be evaluated due to unspecific binding of the human protein (3) or lack of expression of one of the partners (7), 18 pairs and three controls yielded meaningful data. Only one of the control pairs was positive, whereas 13 of the 18 candidate pairs yielded a detectable band specifically in the effector IP (**Fig. 2e; ED Fig. 2**). Altogether, the data demonstrate that we have identified biophysically reliable interactions that are robustly detected by different assays and can occur in the human cellular environment.

REV 2.5 *Because of the absence of any examples of effectors being expressed and translocated, I find some of the inferences made a bit far-fetched. In particular the relationships between these hypothetical effectors, their putative targets and their association to disease are very clearly mentioned in the results section (lines 262-277 for example), which seems a bit too hypothetical and based on vague connections to be in this section (maybe better in discussion) or at least should be toned down.*

AUTHOR RESPONSE

We hope the new experimental evidence demonstrating the translocation of effectors and functionality in at least some of the commensal bacteria alleviate some of these concerns. Beyond this, however, we have toned down some of the statements.

REV 2.6 *In lines 125-127 the authors emphasise that commensals can have T3SS effectors similar to those in pathogens despite being in different orders, pointing towards HGT, but the examples given actually are two species that can be pathogenic (*E. albertii* and *Y. enterocolitica*), which does not help in illustrating this point. To confuse matters further, the authors then say that effectors in commensals are distinct from those in pathogens, which sounds a bit contradictory (line 132-134). The figures seem to largely support the latter conclusion, and so I would appreciate if the text was more clear.*

AUTHOR RESPONSE

We agree and thank the reviewer for pointing this out. As we have added substantial new data we removed this section.

REV 2.7 *How many GFP+ cells are counted to obtain the MFI in the Icam expression experiments? A representative dot plot would be nice to show the populations.*

AUTHOR RESPONSE

We had to remove this figure panel as we noticed that met_7 was not a *bona fide* commensal effector, but identical to a pathogen protein.

Reviewer #3 (Remarks to the Author):

The paper is an exploration of T3SS effector-human protein interactions, specifically focussing on commensal Proteobacteria. With 1,263 interactions discovered between 289 effectors and 430 human proteins, the study is an opportunity to uncover some exciting new biology, however it falls short in this due to its lack of sensitive sequence/structure searching for protein function prediction, and lack of in depth molecular exploration of any new effector-human protein pair representatives. There is a problem with uninformative figures in the main text, and over-interpretation of data. My detailed comments follow below.

Major:

REV 3.1 *I am unsure about the initial premise that there is a clear distinction between pathogenic vs non-pathogenic bacteria. Some pathogens can be opportunistic, or condition specific. Since the distinction is blurry, it seems a reasonable expectation that commensals can also have T3SSs for supporting growth in certain circumstances.*

AUTHOR RESPONSE

We agree with the reviewer on this important point and have carefully rephrased the question and included a more nuanced discussion. At the same time, we think the additional evidence we provide do support the point that effectors in pathogens and those identified in bacteria from the healthy human gut are quite different. While this does not rule out that the T3SS we identified are exclusively there to support occasional pathogenic episodes, the observations are also compatible with the hypothesis that T3SS more generally support microbe-host “communication”.

EXCERPT FROM REVISED MANUSCRIPT

Comparative sequence, structure, and host-target analyses revealed that commensal and pathogenic effector repertoires are largely distinct, supporting a model in which many commensal T3SS are adapted for cooperative rather than pathogenic interactions. Notably, commensal effectors preferentially enhanced Pam3CSK4-induced TLR1/2 signaling, implicating a role in modulating responses to gram-positive *Bacteroidetes* and suggesting a potential mechanism for inter-phyla competition.

REV 3.2 *The authors acknowledge that the algorithm they use may miss systems in certain bacteria (lines 97-98). Biases of the algorithm may also be why they found many hits in the well-studied *Escherichia* (line 99). So given that the results are unlikely to be exhaustive, I'm not sure Fig 1 is useful as a main figure. There is nothing wrong with such an approach for finding candidates for subsequent detailed experimental validation and exploration to find new biology, but since this computational part of the pipeline may be underestimating T3SS presence so much, the actual numbers of hits and in which taxa may not be meaningful.*

AUTHOR RESPONSE

Indeed, these are not precise numbers and many T3SS may have been missed. The main point of this section is to provide a lower estimate and argue the point that T3SS are quite common in healthy human guts. We have kept this section but provided a bit more context and interpretation in the discussion.

EXCERPT FROM REVISED MANUSCRIPT

Although not detected in commensal beta- or delta-*Pseudomonadota*, divergent systems in these groups may have escaped current detection methods opening the possibility that T3SS may be even more common.

REV 3.3 *Figure 2A is not interesting data worth reporting in a main figure. It can be hard to clone toxins, this does not tell us anything about what they do. However it should be noted that the ones that fail to clone are probably the most important effectors as they are they are the most toxic.*

AUTHOR RESPONSE

We agree that the former Fig 2a is less critical. In the context of wider figure reorganization to accommodate new data we have moved the panel to the supplement.

Although toxins are secreted by bacterial pathogens, non-toxic effectors are more common. We believe the proteins we have identified as effector candidates are not actual toxins *sensu stricto* but the type of effector proteins that modulate the physiology of their target cells as functionally interconnected modifiers⁶. More importantly, independent of this working hypothesis, we do not see any evidence of effector toxicity in any of the transfections or transformations we have conducted in bacteria, yeast or human cells.

Specifically, to the reviewer's thought that failure to clone could be indicating toxicity: the failure to clone the missing ORFs was generally due to failure in the first or second PCR where no band was amplified. Only in rare cases did the cloning fail in the ligation/transformation. These numbers are comparable to previous cloning efforts for plant ORFs¹⁷. Therefore, we believe there is no reason to assume that ORFs which failed to amplify during genomic PCR amplification would be 'most important'.

EXCERPT FROM REVISED MANUSCRIPT

Cloning failure was generally due to lack of PCR amplification in the standard conditions established for each strain without indications of toxicity.

REV 3.4 *The section "Commensal effectors are unrelated to known pathogen effectors" is problematic since only identity calculated by simple pairwise sequence alignment is used. It would be appropriate here to use tools like Jackhmmer to sensitivity cluster effectors using iterative HHM searching, HHpred to find hits in the effectors to PFam domains, Uniprot and PDB entries, and AlpaFold/RosettaFold/ESMFold coupled with Foldseek to find similarities that are only apparent at the structure level. Horizontally transferred toxins/effectors can often be small and divergent, meaning more sensitive methods are required for finding homology. The sentence "In fact, clustering effectors by their pairwise interaction similarity identified substantial overlap outside the homology clusters (Extended Data Fig. 3), indicating that dissimilar effectors may have similar functions in the host" (line 209) suggests that some effectors classified as dissimilar might actually be similar if more appropriate sensitive methods are used.*

AUTHOR RESPONSE

We have now added several of the suggested analyses and removed the section on HGT.

The previous BLAST analyses were intended to identify obvious similarities with pathogen effectors to ensure that we are not working with close homologues. But we agree with all reviewers that also weak similarities and more distant relationships are very interesting. We have now conducted a more sensitive sequence analyses and included a new, structure-level analysis of our candidate effectors identified in strains from healthy human guts.

To detect weaker sequence similarities, we have performed iterative sequence searches with jackhmmer against a database of ~124M non-redundant bacterial sequences downloaded from UniRef90 by ensuring that the known T3 sequences from BastionHub were present. For each commensal candidate effector, we ran jackhmmer with 5 iterations using inclusion and comparison E-value thresholds of 10^{-5} . Among the 3,002 effectors from strains, we observed a significant match with known T3 effectors for 155 of them (~5%).

Regarding the 186 effectors identified from MAGs, 42 show a significant similarity (22.5%). Overall, these results support our previous conclusion that only a small fraction of our effectors may be considered as potential homologs to known T3 effectors. We would like to point out that, while generating the new plots for the jackhmmmer results, we realized that the data point selection of the best matches for our effectors was incorrect. We have now fixed the plotting issue in Fig 1 and all corresponding Extended Data Figure panels.

In addition to the jackhmmmer sequences search experiment, we have followed the reviewer's suggestion to explore the similarity between our effectors and known T3 effectors at the structural level by leveraging AlphaFold structure predictions and assess structural homology between commensal and pathogen effectors using Foldseek⁵. When available, predicted effector structures were directly downloaded from the AlphaFold DB. For all effectors without structure models, a sequence comparison using BLAST was performed against all bacterial proteins with an available model in the AlphaFold DB. The best matches among those with sequence identity > 95% and sequence coverage > 90%, were selected as representative models for said effectors. Subsequently, all effectors were clustered into groups based on pairwise structural similarity to the group's representative structure. Technically, clustering was performed for different bidirectional query coverage (qcov) (0.5, 0.7, 0.9) and E-value thresholds (0.001, 0.01, 1, 10) using Foldseek cluster's greedy set cover algorithm. To focus on full length similarity, we used results obtained with qcov = 0.9 and e-value = 0.01 as suggested⁷ for presentation and discussion. The results did not change in any major way when the query threshold was changed.

We observed a seemingly large number of homotypic effector groupings, i.e. effectors from pathogens have similar folds and effectors from commensal microbes also have similar folds, whereas only a small subset of folds is shared between effectors from beneficial and commensal strains. To assess significance of this observation, we performed label permutation tests ($n = 10^4$), while keeping the graph's topology intact. All obtained cluster distributions were significantly different from random expectation (new **Figure 1d**). The associated empirical and fitted P-values of the observed Foldseek cluster distributions are given in **Supplementary Table 2**.

The enrichment of homotypic structural clusters was stable when doing the similarity assessment with lower query coverage (i.e., 0.7 and 0.5) at e-value=0.01, even though, expectedly, the numbers and proportions of commensal effector candidates sharing similarity with pathogen effectors increases as shown in the table below. At the low query coverage value of qcov = 0.5, it seems likely that the structural similarity covers individual domains.

qcov	# of strain effectors	# of MAG effectors
0.9	129 (4.3%)	29 (15.6%)
0.7	369 (12.3%)	73 (39.2%)
0.5	669 (22.2%)	77 (41.4%)

Overall, the results confirm the intuition that more commensal effector candidates show hidden sequence similarity to known pathogen effectors when jackhmmmer is used instead of BLAST. However, the overall proportion remains quite low. Similarly, the structural analysis demonstrated that only a minority of candidate effectors identified in strains in normal human guts are structurally related to pathogen effectors. Thus, overall, we believe these results strengthen our initial conclusion. We have incorporated the new data and adjusted the text accordingly.

EXCERPT FROM REVISED MANUSCRIPT

Canonically, T3SS and substrate effectors support a pathogenic lifestyle. Therefore, we investigated if the candidate effectors share sequence similarity with 1,195 known T3SS pathogen effectors¹⁸. Only 17 of 3,002 (0.5%) strain-effectors and 6 of 186 (3%) meta-effectors showed extended high sequence similarity to those

of pathogens ($\geq 90\%$ similarity across $\geq 90\%$ length). To find weak similarities, we performed iterative sequence searches and multi-sequence alignments with JackHMMER¹⁹ with ~124M non-redundant bacterial sequences from UniRef90. Even with this sensitive approach significant similarity with pathogen effectors was found only for 155 commensal strain-effectors (~5%) and 42 meta-effectors (22.5%) (**Fig. 1c**).

As effectors can be structurally related despite lack of sequence similarity, we also conducted a tertiary structure comparison. Using AlphaFold⁶ we obtained structural models for all effectors and then compared and clustered them into structural groups using FoldSeek⁷. Surprisingly, homogenous structural clusters encompassing effectors only from commensal or pathogenic strains were highly enriched, whereas mixed structural clusters were markedly depleted (**Fig. 1d**, $P \ll 0,0001$, empirical P values). Moreover, the meta-effectors clustered exclusively with strain-effectors in from healthy guts, albeit at a frequency that is close to random expectation. All results were robust over varying FoldSeek parameters, and when considering only effectors of vertebrate or human pathogens (**Supplementary Table 2**). Thus, the candidate effectors in T3SS+ strains from healthy human guts markedly differ from pathogen effectors both in sequence and structure. (**Fig. 1d**, **Supplementary Table 2**).

REV 3.5 *In showing which "effectors" and human proteins interact, Fig 2G is an interesting panel but it is not clear what kinds of proteins we're looking at. The names of the proteins (especially the human ones) should be given here (rotate by 90 degrees for ease of reading as some names will be long). In general this is a problem throughout the paper and SI that full names of proteins are not used, making the results not very human readable. It is usual to include Uniprot protein names and IDs in the SI for human proteins picked up in screens.*

AUTHOR RESPONSE

We apologize for the frustration by incomplete supplementary tables. We have enhanced Supplementary Table 3 by incorporating two additional columns. These columns provide the UniProt Swiss-Prot identifiers for human proteins and their corresponding descriptions. However, in Fig. 2g, we continue to use human gene symbols, as these are the standard nomenclature commonly employed in interactome and genetic studies (see also Kim *et al.*, *Nat Biotechnol*, 2023).

REV 3.6 *(A) Why is sensitivity so low? (B) Perhaps true effector-target interactions are often transient? (C) Isn't it also possible that many of the interactions picked up are proteins that interact for other reasons? The authors seem to assume that interactions of a predicted effector with a human protein mean that the human protein is the target of the effect. While this might be true in the majority of cases, is not clear to me that this is always so. Is it not possible that some interactions represent an immune response to inactivate effector-like proteins? (D) And maybe some protein domains (perhaps the PDZ domain) are just 'sticky' in that they can readily bind other proteins with low specificity?*

AUTHOR RESPONSE

(A) We are not certain which sensitivity is being referred to here, but we assume the reviewer means the 'sampling sensitivity' and the 'assay sensitivity' which we determine as part of the interactome mapping quality control. If this is the case, then we would respectfully disagree with the reviewer's assessment that the sensitivity is particularly low, at least in comparison with other projects conducted with the same pipeline. In fact, both the sampling and the assay sensitivity are comparable to our other high-throughput experiments in this stringently controlled setup for the following reasons:

1. To generate reliable and trustworthy data, it is imperative for HT experiments to avoid false positives. To this end, the stringency of our screening assay is high, e.g., by using low copy plasmids and weak promoters both of which reduce the expression levels of the hybrid proteins in yeast. These features avoid 'driving' interactions artificially by high expression levels but conversely reduce the sensitivity of this assay version. In line with this, a similar assay sensitivity was obtained in a phytohormone signaling-centered interactome map for *A. thaliana*, which was acquired using two screening orientations (20,4%)¹⁷. In another recent publication on human virus-host interactions we did not explicitly determine assay sensitivity, but the data were consistent with similar detection rates²⁰.
2. In addition, this project aimed to explore the existence and potential functions of effectors identified in commensal strains. To increase the breadth of the mapping experiment we have reduced the depth by only screening all effector candidates in one orientation, meaning effector candidates were screened as DNA-binding fusion proteins, against proteins expressed as activation domain (AD-) fusion proteins. Depending on the biochemistry of the involved proteins, also screening the reverse orientation can add ~10% to 50% of interactions. Important for our decision was experience from a previous study in which we screened effectors of plant pathogens against an Arabidopsis ORFeome in both orientations. In that study, most interactions (> 90%) were found with effectors as DB fusions²¹. Thus, we prioritized the orientation from which we expected most interactions, based on experience from a previous related study.
Taken together, the perceptively 'low' sensitivity, is a consequence of a stringent assay sensitivity, which was optimized for high data quality, and efficient screening setup (reduced sampling sensitivity, i.e. saturation).
3. Despite all this, we agree and noted ourselves, that the assay sensitivity for reference interactions between bacterial effectors and human proteins (bhLit_BM-v1) were slightly lower than those of our human positive reference set (hsPRS-v2). We believe partly, this could be a consequence of the quality of the respective interaction pairs. Human protein-protein interactions are generally more intensely studied and better documented, so that the hsPRS-v2 could be composed of a much larger set and better documented protein pairs. A second aspect that we now discuss is that possibly, bacterial proteins are expressed less efficiently.
4. Finally, toxicity of effectors would have been detectable already at the transformation step as a lack of growth and general failure. We did not observe this and transformation of effector-coding expression plasmids into bacteria or yeast was as efficient as for proteins from other organisms without evidence of toxicity.

(B) This is indeed an attractive hypothesis. However, in light of the also a lower background rate of positive scoring bh-RRS pairs, we believe the lower scores are not due to features of the interactions, but of the bacterial proteins in general. As they also do not appear to be toxic, as evidenced by normal transformation rates and normal growth rates of the haploid strains expressing the effectors, we consider lower expression levels a likely explanation.

(C) Absolutely, it is a short-hand saying that the effectors 'target' the human proteins, when in fact some human proteins might intercept or detect the bacterial proteins as decoys or immune receptors. At the same time, as the reviewer agrees, it is likely true in the majority of cases. We therefore decided to overall adhere to the current phrasing but have brief clarifications in the main text about the functional nature of the interactions and the possibility of the interaction functioning to alter human processes (effector → human) or to sense the presence of the effector and bacterium and thus provide information to the host (human → effector).

(D) We consider interactions due to "stickiness" as false-positives and the extensive quality control is aimed to ensure that the generated dataset is as free of contaminations by much as possible. While

we likely did not eliminate all artifacts that lead to false positives, we think the quantitative interaction confirmation using the biochemical and highly orthogonal hold-up assay strongly suggests that the PDZ interactions are indeed *bona fide* interactions and not artifacts of biochemical promiscuity.

EXCERPTS FROM REVISED MANUSCRIPT

The saturation curve indicates that the single main screen has a sampling sensitivity of ~32% (**Fig. 2b**) consistent with previous studies^{17,22}

Benchmarking our Y2H with these and with the established human reference sets (hsPRS-v2, hsRRS-v2)²³ indicated an assay sensitivity of ~13% and 17.5%, respectively, which is consistent with previous observations^{23,24}

It is important to keep in mind that functional effects may go in both directions. While in most cases it is likely that the effectors cause a perturbation of the host cell, intracellular immune receptors may also bind to effectors to then initiate defense processes.

The degree distribution of HuMMI_{MAIN} shows that numerous human proteins are interacting with multiple effectors, often from different species (**Fig. 2f, Supplementary Table 4**), and random sampling demonstrates that the effector convergence is highly significant (**Fig. 2h**). In plant-pathogen networks, we previously demonstrated that effector-convergence on common host proteins reflects the importance of the targets for the microbe-host interaction^{25,26}.

REV 3.7 *The authors should take advantage of state-of-the-art structural prediction methods to try to model some of their interactions. AlphaFold now makes it easy to predict complexes and therefore interaction interfaces.*

AUTHOR RESPONSE

We thank the reviewer for this suggestion and have followed his advice. This structural analysis provided a more detailed picture of common and distinguishing structural features both of the host proteins and the effectors. Overall, this analysis has strengthened our conclusions.

EXCERPT FROM REVISED MANUSCRIPT

We then used structural modeling to understand how different effectors interact with the same human protein and vice versa, hypothesizing that this can help understand effector-host interactions and may provide functional leads. Using AlphaFold Multimer all effector-host interactions were subjected to structural modeling yielding predictions for 123 pairs (10%). Focusing on sets where one protein has multiple interactors, we classified interfaces as 'same' when they shared at least 60% contact residues, as 'different' if the overlap was less than 10%, and 'overlapping' otherwise (**Fig. 3d,e**). The larger proportion of identical interface binding observed on human proteins, compared to effectors, likely reflects the importance of targeting specific human functions. Mapping the binding interfaces to domain annotations further strengthens this hypothesis as even effectors that bind to different interfaces can target the same domain, e.g., the DNA-binding domain of LBX1.

However, more commonly, effectors with different interfaces bind to distinguishable parts of the host protein. Efe_11 and Kpn_9 bind the same interface in the E3 ubiquitin ligase domain of TRAF2, whereas Pem_8 targets the C-terminal MATH domain involved in TRAF2 trimerization and receptor binding. Similarly, Pma_4 binds the DNA binding domain of REL, whereas Yen_11 binds the Rel homology dimerization domain (**Supplementary Table 5**).

REV 3.8 *Fig 3c is not possible to extract any meaning from. How are the proteins organised on the vertical axes?*

AUTHOR RESPONSE

We agree that the bipartite graph in the former panel Fig 3c was not informative. We have removed that panel and rearranged the figures for more clarity.

REV 3.9 *After identifying PDZ domains as potential interactors, the paper goes off on a tangent to investigate viral PDZ domains. It is not convincing that the 'competition' they observe is biologically meaningful. PDZ domains seem to be common domains that can bind to many different kinds of proteins.*

AUTHOR RESPONSE

We now agree that the virus connection is a distracting detour. We have removed the respective section.

REV 3.10 *What are the effectors that are more prevalent in CD than healthy guts? What kinds of proteins are they? Are these some that couldn't be cloned suggesting toxicity? Which are good targets for experimental exploration? The results are presented in a way that is removed from biological discovery and makes it hard for a reader to evaluate. The effector abbreviations used in the paper and SI are not helpful without a key (in each SI table) that allows the reader to easily find the gene/protein name and protein sequence of the effector.*

AUTHOR RESPONSE

We apologize for the confusion and frustration caused by convoluted linkage of our uniform effector IDs to sequence IDs and only indirectly to other identifiers. This analysis was purposely limited to effectors that were not only cloned, but also positive in the protein interaction mapping experiment. This constraint was used to reduce the multiple testing correction burden and focus on effectors for which there is a chance to develop mechanistic hypotheses for potential follow-up. However, we agree that we did not take advantage of this focus and did not make it overly clear in the text. We have now corrected this by making additions to the supplementary tables, especially, Supplementary Table 2D. In addition, we have made the focus of this analysis clearer in the text and expanded the biological discussion, which is supported by a local network figure of an effector-targeted neighborhood that may be relevant to both CD and UC.

EXCERPT FROM REVISED MANUSCRIPT

As we identified genetic and functional links between commensal effectors and IBD, we wondered if we could also find support for a potential role of effectors in clinical data. Hypothesizing that a causal role of effectors in IBD etiology may be reflected in an altered prevalence in patients, we analyzed a large metagenome study

with >800 IBD patients and >300 healthy controls²⁷. To facilitate downstream interpretation, we focused on effectors for which we had physical interactions in HuMMI. Indeed, 64 effectors were significantly more prevalent in the metagenomes of CD patients compared to healthy controls, whereas effectors were less common in UC patients (**Fig. 5e,f, Supplementary Table 8**). These opposing trends were unexpected as an increased abundance of *Pseudomonadota* has been reported for both, CD and UC patients²⁸. We therefore explored if HuMMI could support the development of mechanistic hypothesis for the observed patterns. Indeed, we found effectors from *K. pneumonia*, *E. coli* and *E. fergusonii* that are highly prevalent in CD patients and interact with the CD susceptibility protein COG6. Intriguingly, COG6 is a direct interactor of RTP5, a susceptibility gene for UC (**Fig. 5g**). The same pattern was observed another *E. fergusonii* effector; Efe_13 directly interacts with the CD susceptibility protein TNIP1 (TNFAIP3 interacting protein 1), which functions in NF- κ B signaling and itself interacts with two genes that have been associated to UC. For other highly CD-enriched effectors, indirect links to CD and UC proteins through the same interaction partner are observed (**Fig 5g**). While the mechanistic relevance of these interactions must be clarified in future studies, it is intriguing to speculate that these direct and indirect connections to IBD disease proteins may cause a homeostatic shift that simultaneously increases the risk for CD and decreases the risk for UC.

Minor:

REV 3.11 *It is not properly explained how interfaces are predicted. The principle should be briefly outlined in the results.*

AUTHOR RESPONSE

We have now added a very brief explanation to the concept. The detailed approach including statistical evaluations is detailed in the methods section.

EXCERPT FROM REVISED MANUSCRIPT

Many interactions are mediated by short linear motifs (SLiM) in intrinsically disordered regions²⁹, which can evolve more rapidly and are recognized by specific protein domains. As such interactions are often missed by AlphaFold³⁰, we used the orthogonal *mimicINT* approach to identify SLiM-domain interactions, which matches interaction pairs to known SLiM-domain templates³¹ (**Fig. 3f**).

REV 3.12 *line 65: "Especially" seems out of place here and can be deleted.*

AUTHOR RESPONSE

We have followed the reviewer's advice.

EXCERPT FROM REVISED MANUSCRIPT

The host-associated microbiota influences human health in complex, genotype-dependent ways.

REV 3.13 *line 200: "The function of unknown proteins can often be inferred from better-studied orthologues,*

but convergence could also result from high sequence similarity among effectors. " What is meant by convergence here? Convergent evolution? It needs better explaining.

AUTHOR RESPONSE

“Convergence” refers to the analysis in the preceding paragraph, where we demonstrated that evolutionarily unrelated effectors interact with, i.e. converge on, the same host proteins more often than expected by chance. One aim of the structural analysis is to explore if this is due to common structural features, or if convergence is largely independent. We have modified the sentence to make the context more transparent how ‘convergence’ is used here.

EXCERPT FROM REVISED MANUSCRIPT

Many functional and interaction inference approaches rely on the assumption that sequence similarity implies interaction similarity; also, the observed convergence of effectors on common host proteins could simply be a consequence of effector similarity.

Reviewer #4 (Remarks to the Author):

Key results

The manuscript highlights the role of Type III Secretion Systems (T3SS) in the human gut microbiome focusing on the Pseudomonadota phylum. Analysing large datasets from human gut samples demonstrates that T3SS effectors show minimal sequence similarity to known pathogenic effectors, suggesting distinct functions. The study reveals the microbiome-host interactions highlighting significant convergence of effectors on specific host proteins. The manuscript also explores the structural features, such as PDZ domains, and proposes competition between bacterial and viral proteins for host binding partners. The manuscript further investigates the contribution of effectors to immunomodulatory functions and disease pathogenesis. The authors identified enrichment in immune signalling, particularly in the NF- κ B pathway, as well as metabolism, linking to conditions like cancer and immunological diseases. Functional assays demonstrate that certain effectors can both activate and inhibit the NF- κ B pathway, influencing cell adhesion and cytokine secretion. Clinical analysis of patients with inflammatory bowel disease (IBD) suggests altered prevalence of effectors, potentially distinguishing Crohn's disease and ulcerative colitis proposing that commensal effectors' role in disease pathogenesis by modulating host cellular functions.

Validity

The presented workflow is robust and reproducible, some databases and tools have their version number, but I would suggest describing it for all (eg. ELM, IUPred, InterProScan).

REV 4.1 *All source code is provided as zip file, but to track changes and facilitate collaborations, I would suggest storing them in a version control system (e.g. Git).*

AUTHOR RESPONSE

We have collected all code and data in zenodo archives provided in the code availability section.

REV 4.2 *The study provides a new insight into the immunomodulatory effect of commensal T3SS effectors. I would suggest to emphasize better the difference between pathogenic and commensal T3SS effectors. The authors compared them in terms of sequence similarity but it would be great to have a look at the differences from a network perspective (eg. convergence) and comparing the immunomodulatory effect of commensal effectors to pathogens according to the literature.*

AUTHOR RESPONSE

This is similar to REV1.1 and we have more extensively responded to this question further up. Briefly, this analysis is more complicated because differences in how the experiments are being done and which proteins are being investigated strongly influence both our systematic screen, as well as the results from hypothesis driven studies, that dominate the literature.

With this caveat in mind, we have downloaded all effector-host interactions from the literature and compared these in terms of shared protein targets and shared targeted functions. Briefly, the overlap among all targets is on the low side (12) but still statistically significant. Even the number of 'convergence proteins' that are also targeted by pathogens is significant. Regarding the functions, we see functions targeted by both groups of effectors and some that do not appear or are not significant among the pathogen targets, but which we found for the commensals.

As indicated, these findings must be interpreted with caution. Additional details can be found in response to REV1.2.

EXCERPT FROM REVISED MANUSCRIPT

To explore if these are also targeted by pathogen effectors, we extracted 265 binary high-quality interactions between 217 human proteins and 80 effector proteins from 17 pathogenic bacterial strains from IntAct⁴. We

found a numerically low albeit significant number of twelve human proteins that are targeted by both groups ($P = 0.014$, Fisher's exact test; OR = 2.26), of which three are subject of convergence by commensals ($P = 0.067$, Fisher's exact test, OR = 3.37). Given the relatively small numbers, experimental differences, and the non-systematic nature of the pathogen data it is difficult to draw strong conclusions. However, given our previous observations³, the data are consistent with overlap but also lifestyle dependent specificity in the host proteins targeted by effectors from commensal and pathogenic bacteria.

Finally, we wondered how the targeted functions compare to those of pathogens. For the pathogen effector targets more than 200 often closely related GO terms were identified (**Supplementary Table 6**). Comparison of functions targeted by pathogens and commensals revealed commonalities, such as "NF-kB signaling", but also functions that are not significant or absent from the pathogen targets, such as "collagen biosynthesis" and "response to muramyl dipeptide". These data reinforce the notion of both life-style dependent specificity and functional overlap in molecular protein-interactions of commensals and pathogens with the human host.

REV 4.3 *The 'Discussion' section is heavily focused on the disease context and does not conclude the rest of the analysis carried out in the study (eg. the microbe-host interactome, convergence analysis or interface prediction).*

AUTHOR RESPONSE

Yes, we agree and have expanded the discussion to more comprehensively cover the many different aspects our work touches on.

EXCERPT FROM REVISED MANUSCRIPT

Please see revised discussion.

Data and methodology

The manuscript describes a robust multi-layered analysis including experimental and in silico approaches in a well-organised, structured way.

REV 4.4 *However, in the 'Meta-interactome mapping' method section, further clarification or contextualisation could improve the understanding, especially for readers less familiar with certain methodologies.*

AUTHOR RESPONSE (identical to 4.17)

We have included a description of the pipeline in the main text.

EXCERPT FROM REVISED MANUSCRIPT

MAIN TEXT

The pipeline consists of four distinct interrogation steps and is optimized to ensure high data quality by using weak promoters and low-copy plasmids (**ED Fig. 2**). Initially, interaction identification was done using a yeast-2-hybrid-based pipeline consisting of primary screening, retesting, and a 4-fold verification.

METHODS

Briefly, in the initial 3-step yeast-2-hybrid (Y2H)-based interaction identification, haploid yeast expressing candidate effectors genetically fused to the Gal4 DNA binding domain (DB-X) and human proteins fused to the Gal4 activation domain (AD-Y) were combined by yeast mating. In the primary screen (1st step), individual DB-X constructs are screened against minipools of 188 AD-fusion proteins. Single positives are picked, and the isolated colonies are retested to ensure robustness and specificity of the phenotype (2nd step). Following sequence identification all identified interaction candidate pairs were tested in quadruplicate in the verification step (3rd step) using independently grown and freshly mated yeast cultures. The dataset of the resulting *bona fide* interactions is then subjected to experimental validation using a benchmarked orthogonal validation assay (4th step). Prior to screening, DB-X ORFs were tested for autoactivation by mating against AD-empty plasmids in Y8800 (MATa).

REV 4.5 *The utilisation of the Recon3D human metabolic model for metabolic subsystem analysis is appropriate. However, the assumptions and limitations of this model should be acknowledged for a more comprehensive interpretation of the metabolic findings. (identical to REV4.24)*

AUTHOR RESPONSE

Indeed, Recon3D is a comprehensive model of human metabolism that assumes a static network without dynamic changes over time, stoichiometric constraints that assume reactions occur at their theoretical maximum rates, and steady-state conditions for metabolite concentrations. It assumes specific gene-protein-reaction associations based on curated data and uses tissue-specific data to represent reactions and pathways. The limitations include incomplete data coverage, simplified kinetic enzymes without detailed regulatory mechanisms, and lack of context-specific variability capturing differences between cell types or physiological states. We now mention these assumptions and limitations in a condensed form in the main text.

EXCERPT FROM REVISED MANUSCRIPT

Several metabolism-related functions were significantly enriched in target proteins; therefore, we tested the abundance of targeted enzymes in metabolic subsystems using the human genome-scale metabolic model Recon3D³². Recon3D is a comprehensive tissue-resolved but static model of human metabolism that is based on curated but incomplete data and does not incorporate posttranslational or allosteric regulatory mechanisms.

REV 4.6 *Finally, limitations-related to this comprehensive analysis should be discussed in the 'Discussion' section.*

AUTHOR RESPONSE (this point is identical to REV 4.10)

We agree and we have now rewritten and expanded the discussion to discuss more aspects of our study, including limitations. Please note, that due to the space limitations, we could not go into a detailed discussion of all limitations. We therefore also touch on some minor limitations when presenting the results.

EXCERPT FROM REVISED MANUSCRIPT

Please see rewritten discussion.

Analytical approach

In general, the authors have employed a variety of statistical and computational methods to prove a robust and comprehensive workflow.

REV 4.7 *However, the protein-protein interaction interface analysis highlights weaknesses in the computational pipeline. The domain-motif interaction based approach gives a directionality for the microbe-host interactions and the current study describes how domains on human proteins bind to motifs on bacterial effectors. But the authors aimed to demonstrate the impact of effectors on host proteins, hence the analysis does not answer the proposed goals.*

And further below:

As the domain-motif interactions are directed (the domain is binding to the target SLiM), the current analysis describes how human proteins are able to bind to bacterial effectors. To explore the effect of commensal effectors on human proteins, I suggest redoing the interface analysis considering the bacterial domains connected to human SLiMs

AUTHOR RESPONSE

We are not entirely sure regarding the identified mismatch between the structural analysis and the functional 'impact' question. We agree that the structural analysis does not address how the effectors impact the host proteins. While this will be an exciting question for future studies, such structure-informed mechanistic studies are beyond the scope of this work.

In addition, we would respectfully disagree with the notion, more explicitly stated below in REV4.10, that the structural information inherently provides directionality from the human domain to the bacterial SLiM. As domains and domain specificities are more constrained by host biology, where these all have specific functions, and because SLiMs can evolve more rapidly, it would be more plausible to infer directionality from the SLiM containing proteins to the one harboring the domain.

Of note, the *mimic*INT workflow can also detect domain-domain interfaces, also involving also non-eukaryotic-like domains (ELDs), that can mediate protein-protein interactions according to available three-dimensional models listed in the 3did database. We indeed identified a possible domain-domain interface in four HuMMI interactions (Supplementary Table 4B).

(A) Directionality of SLiM-domain interaction inference

Eukaryotic short linear motifs are sequence elements mediating protein-protein interactions and mostly located in natively disordered or solvent-accessible regions, which are evolutionarily variable segments where motifs may appear or disappear³³. This process is facilitated by convergent evolution events, in which similar motifs may arise *de novo* in unrelated protein sequences. Thus, pathogenic -- and commensal -- organisms may take advantage of host systems if their secreted proteins also contain convergently evolved motifs, as reviewed by some of us³⁴ and also by Sámano-Sánchez & Gibson³⁵.

Over the years, many instances of ELDs have been detected in prokaryotic genomes. Previous studies showed that these ELDs were likely acquired via horizontal-gene transfer and are important for bacteria-host interactions. Many of them have kept their original function like the SET domain, present in *Legionella* effectors that participate in host histone methylation^{36,37}. Other domains have lost their "eukaryotic" functions as either their sequences have significantly diverged, like in the case of bacterial SH3 domains, which are not able to bind proline-rich motifs, or the bacterial proteins bearing them are not necessarily localized at the host interface (e.g., they are not injected in the host-cell cytosol) like in the case of bacterial PDZ domains³⁸, thus impeding the recognition of potential cognate motifs in host proteins.

These are the reasons why we sought to identify interaction interfaces in HuMMI that are potentially mediated by eukaryotic-like motifs present in effector sequences that can be recognized by human domains.

The more technical questions regarding the IUPred tools and cut-off are answered below in REV4.10.

EXCERPT FROM REVISED MANUSCRIPT

Many interactions are mediated by short linear motifs (SLiM) in intrinsically disordered regions²⁹, which can evolve more rapidly and are recognized by specific protein domains. As such interactions are often missed by AlphaFold³⁰, we used the orthogonal *mimicINT* approach to identify SLiM-domain interactions, which matches interaction pairs to known SLiM-domain templates³¹ (**Fig. 3f**).

It is important to keep in mind that functional effects may go in both directions. While in most cases likely the effectors cause a perturbation of the host cell, also intracellular immune receptors may also bind to effectors to then initiate defense processes.

As a computational biologist, assessing the experimental protocols and methods is outside the scope of my expertise.

Clarity and context

REV 4.8 *The manuscript provides a detailed description of methods and results referencing specific studies, databases, and tools. While most of the abbreviations are explained, others could be clarified to enhance understanding (eg. HU, HMM, BD, DMEM, HEK 293 - mention this is a cell line). One potential improvement could be the inclusion of a brief summary or overview at the beginning of the methods section to provide readers with a high-level understanding of the overall approach before delving into the details. Additionally, the integration of visual aids, such as flowcharts or diagrams, could enhance the clarity of the complex computational and experimental processes described in the methods.*

AUTHOR RESPONSE

We have now included the missing abbreviations.

We also considered adding a brief explanation of the flow of methods. In doing so, however, we noticed that this would largely recap the main narrative of the manuscript along which the methods are organized. We therefore decided to only preface the methods with a brief indication, what part of the manuscript the respective method was used for to increase clarity of the methods.

References

REV 4.9 *The manuscript cites 110 articles covering the field of host-microbe interaction research, however there is a need for formal improvements: the citation for BastionHub is duplicated (22 and 79); in line 1152, there is a missing tab; doi IDs should be formatted properly, either highlighting them as a link or using only the ID.*

AUTHOR RESPONSE

We apologize for this embarrassing oversight and have carefully checked the references to remove duplicates.

Suggested improvements

Major improvements

The authors explored the interface between microbe-host proteins connecting bacterial motifs to human domains.

REV 4.10.

(A) Besides, I suggest using one of the newer versions of IUPred (2A or 3) as they provide a more precise prediction of disordered regions through defining two scores.

(B) The authors defined the cut-off for 0.2 in the manuscript while the IUPRED tool describes the following: 'Residues with a predicted score above 0.5 are considered disordered, while residues with lower scores considered to be ordered.' Hence, the currently used 0.2 cut-off is not appropriate to predict disordered regions, the score should be at least 0.5 for both IUPred and ANCHOR values.

(The part of this question on directionality was cited and answered together with a related earlier comment further up in REV4.7)

AUTHOR RESPONSE

(A) Most recent version of IUPred

The three versions of the IUPred method (1.0, 2A and 3) use the same force field and the same architecture originally published in 2005³⁹. As stated by the IUPred developers, the original implementation (i.e., the one used in the *mimic*INT workflow) and the 2A version are virtually the same⁴⁰ (AUC = 0.855 and 0.856 for IUPred2A and IUPred, respectively, on the same test set). In the most recent version, IUPred3, the developers added two additional layers ("medium" and "strong") to further smooth the disorder propensity predictions⁴¹, achieving slightly better performances when comparing with version 2A on the same testing set (AUC=0.736 and 0.738 and 0.744 for 2A, 3 with medium smoothing, and 3 with strong smoothing, respectively). Overall, the three versions show comparable performances.

We have decided to use the first IUPred implementation in our workflow as it is written in the C programming language, which enables a 50x faster running time on very large sequence sets compared to the Python-coded versions 2A and 3.

Nevertheless, we sought to compare the IUPred 1.0 disorder propensity of motifs, which we inferred as potentially involved in interaction interfaces with cognate human domains, with the ones predicted by IUPred3. To do so, we calculated four average propensity scores for each motif involved in both disorder modalities (i.e., short and long, as we did for IUPred 1.0 predictions) and using both smoothing layers (i.e., medium and strong) for a total of four propensity scores per motif. None of the motifs has all four propensity scores below the 0.2 threshold, meaning that even in case we had used IUPred3 for disorder prediction, we would have been able to identify the same number of putative interfaces with *mimic*INT.

(B) Choice of the disorder threshold

We set the disorder propensity threshold to 0.2 as this is the common practice in SLiM detection to limit the false negative rate⁴². Indeed, it has been shown that ~30% of experimentally characterized functional motifs in the ELM database have an average disorder propensity between 0.2 and 0.5, i.e., the standard IUPred threshold; for 10% of the validated motifs the average disorder is below 0.2²⁹. However, virtually all false positive / true negative motif instances in the ELM database have a low average disorder propensity (<0.4).

To control for false positive motif detection, we have implemented a two-step strategy in the *mimic*INT workflow. First, we consider only SLiMs with a pattern probability < 0.01 as reported in the ELM database. Second, as described in this manuscript and in the *mimic*INT publication³¹, we use Monte-Carlo simulations to assess the probability of a given SLiM to occur by chance in our effectors by randomly shuffling the disordered regions of the effector sequences to generate a large set of N randomized proteins (e.g., 100 000 randomized sequences per effector). The occurrences of each detected motif in an effector sequence were compared to the occurrences observed in the corresponding set of shuffled sequences. We considered as

significant all the motif occurrences having an empirical P value lower than 0.1. In other words, the rarer the motif, the higher is the probability that the given motif can be functional⁴³.

REV 4.11 *Considering and describing the limitations of the study in the 'Discussion' section.*

AUTHOR RESPONSE

We agree and we have now rewritten and expanded the discussion to discuss more aspects of our study, including limitations.

EXCERPT FROM REVISED MANUSCRIPT

Please see rewritten discussion.

Minor improvements

REV 4.12 *The 'Main' section is currently very short and it provides a brief background about the topic*

AUTHOR RESPONSE

Length of the introductory paragraph, as the discussion, are always challenging given the word limitation of 3,500 words. Considering this constrain and extensive new data that also require description and discussion, we have added some thoughts to the introduction, but not substantially expanded it. We hope the reviewer understands and agrees with this prioritization.

EXCERPT FROM REVISED MANUSCRIPT

Please see revised introductory paragraph.

REV 4.13 *The 'Result' header is missing*

AUTHOR RESPONSE

Thank you - we have now included the results header.

REV 4.14 *'From 16,179 Pseudomonadota MAG bins with high or intermediate genome quality' - define what is high/intermediate quality (line 92-93)*

AUTHOR RESPONSE

We have added a reference to the MIMAG standards, according to which the MAG bins have been classified.

REV 4.15 *Missing '(' in line 438*

AUTHOR RESPONSE

Thank you

REV 4.16 *Provide more details on the previously developed Y2H screening pipeline (identical to Rev 4.4)*

AUTHOR RESPONSE

We have now amended the main section and especially the methods sections with additional details and references.

EXCERPT FROM REVISED MANUSCRIPT

The pipeline consists of four distinct interrogation steps and is optimized to ensure high data quality by using weak promoters and low-copy plasmids (**ED Fig. 2**). Initially interaction identification was done using a yeast-2-hybrid-based pipeline consisting of primary screening, retesting, and a 4-fold verification.

Briefly, in the initial 3-step yeast-2-hybrid (Y2H)-based interaction identification, haploid yeast expressing candidate effectors genetically fused to the Gal4 DNA binding domain (DB-X) and human proteins fused to the Gal4 activation domain (AD-Y) were combined by yeast mating. In the primary screen (1st step), individual DB-X constructs are screened against minipools of 188 AD-fusion proteins. Single positives are picked, and the isolated colonies are retested to ensure robustness and specificity of the phenotype (2nd step). Following sequence identification all identified interaction candidate pairs were tested in quadruplicate in the verification step (3rd step) using independently grown and freshly mated yeast cultures. The dataset of the resulting *bona fide* interactions is then subjected to experimental validation using a benchmarked orthogonal validation assay (4th step). Prior to screening, DB-X ORFs were tested for autoactivation by mating against AD-empty plasmids in Y8800 (MATa).

REV 4.17 *Insert a paragraph about the workflow summary/create a workflow figure*

AUTHOR RESPONSE (same answer as to REV 4.4)

We have included a description of the pipeline in the main text.

EXCERPT FROM REVISED MANUSCRIPT

The pipeline consists of four distinct interrogation steps and is optimized to ensure high data quality by using weak promoters and low-copy plasmids (**ED Fig. 2**). Initially interaction identification was done using a yeast-2-hybrid-based pipeline consisting of primary screening, retesting, and a 4-fold verification.

REV 4.18 *Extend the 'Discussion' section concluding all the significant results not only focusing on disease-related importance of the commensal effectors.*

AUTHOR RESPONSE

We have rewritten and expanded the discussion to discuss more aspects of our study, including limitations

however within the constraints of the available space.

EXCERPT FROM REVISED MANUSCRIPT

Please see rewritten discussion.

REV 4.19 *The authors mention 1638 effectors derived from BastionHub database (in line 117), later 1195 gathered effectors are mentioned (in line 474) - Please clarify this.*

AUTHOR RESPONSE

We thank the reviewer for spotting this inconsistency. The correct number of T3 effector sequences collected from BastionHub is 1195. We have now fixed it.

EXCERPT FROM REVISED MANUSCRIPT

Therefore we investigated if the candidate effectors share sequence similarity with 1,195 known T3SS pathogen effectors¹⁸.

REV 4.20 *Description of 'odds ratio' is missing from the text*

AUTHOR RESPONSE

We added the description of the odds ratio in the legend of panels a and b of **Fig. 4**.

EXCERPT FROM REVISED MANUSCRIPT

In panels a and b, the odds ratio, calculated as the odds of function-annotated/trait-associated human genes encoding effector targets to function-unannotated/trait-unassociated human genes encoding effector targets in the target set, divided by the same ratio in the HuRI set (see methods), estimates the effect size of each tested function/trait.

We calculated the odds ratio and the fold enrichment to estimate the effect size of each tested function. The odds ratio was calculated for each function as the odds in the target set divided by the odds in the HuRI set. The odds in the target set are the number of function-annotated target proteins divided by that of the function-unannotated target proteins.

REV 4.21 *The authors used their mimicINT tool for interface prediction that has not been published yet. Why is this the best option for interface prediction? Is not there any similar tool that has been already published?*

AUTHOR RESPONSE

We first proposed the template-based inference approach in an article published in 2017⁴⁴, in which we inferred the interactions between the secretome of *F. nucleatum* and human proteins. We further improved the approach, by implementing the mimicINT open-source workflow, which was first described in a 2022 preprint (<https://www.biorxiv.org/content/10.1101/2022.11.04.515250v1>), and has now been published in

F1000Research³¹. Indeed, over the years, several computational methods have been developed to predict pathogen-host protein interactions, some of which are based on the detection of sequence or structural mimicry elements^{45,46}. However, the source code of many of these tools is not freely available to the community (e.g., Becerra et al., 2017⁴⁷; Guven-Maiorov et al., 2017⁴⁸; Lasso et al., 2019⁴⁹) or can be only used through a web interface (e.g. Guven-Maiorov et al., 2020⁵⁰), thus limiting reproducibility and tool usability.

Among the most recently developed ones, the MicrobioLink pipeline's source code is publicly available⁵¹. Like *mimicINT*, it infers microbe-host protein interactions using known domain-domain and motif-domain templates. However, compared to *mimicINT*, MicrobioLink only provides a disorder propensity filter to limit false positive predictions when inferring motif-mediated interactions. More recently, the SLiMAN tool has been proposed to infer motif-mediated interactions^{50,52}.

Besides the disorder propensity filter, SLiMAN, as *mimicINT*, includes a domain specificity score that can be used to rank and select inferred interaction of higher confidence. It is based on the HSM approach proposed by Cunningham and colleagues⁵³. However, the HSM method is limited to few motif-binding domains (i.e., PDZ, SH2, SH3, WW, WH1, and PTB), whereas the *mimicINT* domain score provides a higher coverage of the known motif-binding domain space, covering 212 ELM motif classes and their cognate domains (73% of the 290 motif classes present in ELM, August 2020). In addition, SLiMAN is a web-only tool accepting only UniProt protein accession numbers as input, thus limiting its usability and large-scale applicability.

All in all, compared to recently developed similar tools, *mimicINT* allows large-scale inferences and, by computing motif probability and domain specificity scores, identifies potential high-confidence interaction interfaces.

We have now included citations for *mimicINT*.

REV 4.22 *Provide version number for all tools and databases (eg. ELM, InterProScan)*

AUTHOR RESPONSE

We have included the version or release number for each tool or database whenever applicable. For those tools or databases without release or version, we provided the relevant reference. For instance, InterProScan is referenced as InterProScan 5, and the ELM database is cited as the 2022 release.

REV 4.23 *The authors compared commensal and pathogen effectors in terms of sequence similarity but it would be great to have a look at the differences from a network perspective (eg. convergence) and comparing the immunomodulatory effect of commensal effectors to pathogens according to the literature.*

AUTHOR RESPONSE

This is similar to questions REV1.1 and REV4.44 and we would like to point to the respective sections for a more detailed discussion. Briefly, such a comparison is technically challenging as no systematic data for pathogens from a technically similar setup are available. Thereby, technical differences and biases influence the results and limit the conclusions. Nonetheless, we have downloaded the corresponding data from a curated PPI repository (IntAct) and conducted the node and network level comparison.

EXCERPT FROM REVISED MANUSCRIPT

To explore if these are also targeted by pathogen effectors, we extracted 265 binary high-quality interactions between 217 human proteins and 80 effector proteins from 17 pathogenic bacterial strains from IntAct⁴. We found a numerically low albeit significant number of twelve human proteins that are targeted by both groups

($P = 0.014$, Fisher's exact test; OR = 2.26), of which three are subject of convergence by commensals ($P = 0.067$, Fisher's exact test, OR = 3.37). Given the relatively small numbers, experimental differences, and the non-systematic nature of the pathogen data it is difficult to draw strong conclusions. However, given our previous observations³, the data are consistent with overlap but also lifestyle dependent specificity in the host proteins targeted by effectors from commensal and pathogenic bacteria.

For the pathogen effector targets more than 200 often closely related GO terms were identified (**Supplementary Table 6**). Comparison of functions targeted by pathogens and commensals revealed commonalities, such as "NF- κ B signaling", but also functions that are not significant or absent from the pathogen targets, such as "collagen biosynthesis" and "response to muramyl dipeptide". These data reinforce the notion of both life-style dependent specificity and functional overlap in molecular protein-interactions of commensals and pathogens with the human host.

REV 4.24 *The assumptions and limitations of the Recon3D metabolic model should be acknowledged for a more comprehensive interpretation of the metabolic findings.*

AUTHOR RESPONSE

Please see response to comment REV 4.5.

REV 4.25 *While some of the abbreviations are explained, others could be clarified to enhance understanding (eg. HU, HMM, BD, DMEM, HEK 293 - mention this is a cell line).*

AUTHOR RESPONSE

We have now included the requested explanations. Specifically, they refer to the Hold-up assay (HU), Hidden markov model (HMM), Beckton Dickinson – a company, Dulbecco's modified eagle medium (DMEM) and Human embryonic kidney 293 cells.

REV 4.26 *BastionHub is cited twice (ref 22 and 79).*

AUTHOR RESPONSE

Thank you for catching this. We have removed the excess reference.

REV 4.27 *In line 1152, there is a missing tab by the citation.*

AUTHOR RESPONSE

Thank you, we have corrected this.

REV 4.28 doi IDs should be formatted properly, either highlighting them as a link or using only the ID.

AUTHOR RESPONSE

We thank the reviewer for catching this inconsistency and have now unified doi IDs in the reference list.

REV 4.29 Comments on figures

There is no need for Fig. 1c as the two numbers have been mentioned in the text

AUTHOR RESPONSE

We agree and we have removed the panel.

REV 4.30 Describe the colour legend in Fig. 1d (blue-grey scale)

AUTHOR RESPONSE

The blue-gray density area was computed using a Kernel Density Estimation (KDE) in R to estimate the density in the similarity-coverage data generated by the pairwise sequence comparisons between our effectors and known T3 effectors. The density contours cover ~80% of the data points in Fig. 1b.

However, as we found the density coloring difficult to interpret, we have now added bar graph projections of the number of effectors in each 5% bin to the plot, to indicate the cumulative effector count. At the same time, we have reserved the use of color, to indicate commensal effector candidates that have significant similarity to pathogen effectors by the jackhmmmer analysis. We hope that the figure panel is now clearer.

REV 4.31 Reference for Fig. 2a is missing in the text

AUTHOR RESPONSE

Following the recommendation of reviewer 3, we have removed this panel altogether from the main text and moved it to the **ED Fig. 2**.

REV 4.32 Fig. 2g is consisting of two panels that is confusing

AUTHOR RESPONSE

We agree and have changed the numbering so that each panel has a separate letter.

REV 4.33 What is 'HU' on Fig. 3e? Also, the yellow border is less visible

AUTHOR RESPONSE

This refers to the Hold-up assay as referred to in the text and legend. We have remade the figure and increased the contrast of the corresponding box. We also redefine the abbreviation in the figure legend now

for clarity.

REV 4.34 *Fig. 3f is small, the 'FP' should be described in the figure caption*

AUTHOR RESPONSE

We agree. However, as another reviewer was not convinced of the relevance of this data, and we have extensive new data, we have removed this figure panel from the main figures and increased the size in the supplement.

REV 4.35 *I would suggest changing the layout for Fig. 4d to a list/table instead of the current circle arrangement*

AUTHOR RESPONSE

We have considered this suggestion. However, a list or table format would require more space and thus prevent the display of all traits. We have therefore decided to keep the current display.

REV 4.36 *Comments on Supplementary tables*

As the tables consist of several worksheets related to different parts of the analysis, I would suggest to refer to the exact worksheet instead of the table number

AUTHOR RESPONSE

We agree that this would be more helpful. We were under the impression that it is Nature policy to only list the supplementary material by major number and not specifically call individual subsheets (that applies especially to Extended Data Figure, where we did not find examples of published papers in which specific ED Fig panels are called out). We will clarify this with the editor and will gladly implement this suggestion if allowed, should the manuscript be accepted.

REV 4.37 *Extended Data Table 2A-B describes the predicted effectors from collected reference strains and MAGs. Where is the HBC/UHGG-related data?*

AUTHOR RESPONSE

We apologize for this oversight.

In addition, we noticed that the UHGG was erroneously given as a strain collection, which it isn't. We have now corrected both and the information regarding the detection of secretory systems in the HBC, the Broad Institute-OpenBiome Microbiome Library (BIO-ML), and in the Global Microbiome Conservancy (GMC) collections are included in **Supplementary Table 2**.

REV 4.38 *Points that should be clarified*

Why did the authors select other phyla in the first part of the analysis from HBC and UHGG?

AUTHOR RESPONSE

We agree that this was a confusing inconsistency that we have now removed. In addition, we noticed that the UHGG was erroneously named as a strain collection, which it isn't.

EXCERPT FROM REVISED MANUSCRIPT

To expand the scope, we analyzed genomes of 4,752 phylogenetically diverse strains from the human intestinal bacteria collection (HiBC)^{54,55}, and Broad Institute-OpenBiome Microbiome Library (BIO-ML), and Global Microbiome Conservancy (GMC)^{56,57}. Of the 568 *Pseudomonadota* genomes, 449 (79%) had complete T3SS (**ED Fig. 1a**); similar proportions have T4SS (315) and T6SS (474), both of which can also deliver effectors into host cells but also have other functions (**ED Fig. 1, Supplementary Table 1**)¹¹. Together 527 of the 568 *Pseudomonadota* genomes, *i.e.*, 92%, have at least one host-directed secretion system. Because culturing can bias the relative proportions of taxa, we sought to confirm the presence of T3SS in commensal microbiota using metagenome datasets. In a global collection of 16,179 *Pseudomonadota* metagenome-assembled genomes (MAGs) with high or intermediate genome quality¹²⁻¹⁴ 770, *i.e.*, 5%, encoded complete T3SS (**ED Fig. 1a, Supplementary Table 1**).

REV 4.39 *How did the authors select the commensal Pseudomonadota strains?*

AUTHOR RESPONSE

From the Deutsche Sammlung von Mikroorganismen und Zellkulturen (DSMZ) and the Human Microbiome Project (HMP) 44 *Pseudomonadota* strains were identified as being isolated from the human gut, encoding a T3SS, and with accessible DNA for cloning. Of those, we aimed to select diverse, diet-dependent strains. The former selection criteria gives a broader overview of the effectors' capabilities and functions, while the second selection criteria arose from the link between diet, *Pseudomonadota* abundance, and disease: *Pseudomonadota* abundance fluctuates in response to dietary intake and bacterial abundance is associated with certain diseases. Based on a literature search in the third quarter of 2019, six strains (*Escherichia coli* MS 200-1, *Escherichia coli* MS 69-1, *Edwardsiella tarda*, *Klebsiella pneumoniae*, *Providencia rettgeri_D*, *Providencia stuartii*) were selected due to nutritional impacts on bacterial abundance or bacterial metabolism (e.g., a decrease in abundance after pre-/pro-/symbiotic administration or the ability to metabolize choline to trimethylamine). To select phylogenetically diverse strains, effector genes were clustered according to 90% sequence identity over 90% sequence length. This analysis mostly coincided with the clustering of the strains according to genera revealing similarities between effectors of the same genus. To reduce redundancy between the cloned effectors, only one strain per genus was chosen, besides the strains that had already been selected based on their association with nutrition. Per genus, the strain was selected for which genomic DNA or living cultures were accessible or which encoded the highest numbers of predicted effectors.

REV 4.40 *How did the authors select the pathogens from the BastionHub database?*

AUTHOR RESPONSE

To be most inclusive in the comparison we included all effectors from BastionHub. We did not remove any effectors in the respective comparisons. The structural analysis was also done by limiting this analysis to only vertebrate (incl. human) pathogens and to include only human pathogens with the same results.

REV 4.41 *What does ‘mostly confidently predicted’ mean in line 114 (“3,002 effector candidates from the 44 reference strains that were most confidently predicted by all tools”)?*

AUTHOR RESPONSE

This means that all prediction tools gave a positive prediction for these candidates. Specifically, each of the employed tools provides confidence scores to the identification. We gave 1 ‘point’ for a confident identification and 2 ‘points’ for the highest score assigned by the respective tool. We then used a combined 4 points from three tools as a cutoff. The rationale was that two 90% confident predictions with one 99% confident prediction reflect good agreement. Similarly, as the three tools identify candidate effectors using different conceptual and computational approaches, we also wanted to include those predictions where two tools agree strongly with 99% confidence, while the third tool assigns a lower than 90% confidence. We have clarified this scoring in the methods section.

EXCERPT FROM REVISED MANUSCRIPT

Machine-learning based tools were used to predict T3SS signals (EffectiveT3 v.2.0.1 and DeepT3 2.0⁵⁸) or effector homology using pEffect⁵⁹ to extract potential effector proteins. The results of all three tools were combined using a 0 - 2 scoring scheme: 2 for perfect score (pEffect > 90, EffectiveT3 > 0.9999, DeepT3: both classifiers positive prediction), 1 for positive prediction as defined by default settings (pEffect > 50, EffectiveT3 > 0.95, DeepT3: one classifier) and 0 for negative prediction. Sequences with a sum score above 4 were regarded as potential effectors.

REV 4.42 *The authors set up a cut-off for 50% completeness and less than 5% contamination but how reliable are the computational tools of T3SS prediction when MAGs are incomplete or contaminated?*

AUTHOR RESPONSE

Indeed, the likelihood of identifying T3SS in more fragmented MAGs decreases. The ‘intermediate quality’ is an unofficial grouping that takes the 50% completeness of the official ‘medium quality’ MAG category but requires the stricter threshold of 5% contamination that is also applied to high-quality bins, instead of the 10% that is required for the official ‘medium quality’ assignment. This category is also sometimes referred to as ‘upper medium’ quality. Consequently, the sensitivity for T3SS prediction is expected to be decreased in the intermediate quality bin due to the lower completeness, but the risk of false positive detections is the same as for the high-quality bins due to the same strict upper limit for contamination. It should also be noted that only a portion (<15%) of intermediate quality MAGs have a completeness of less than 60% whereas most are 60-90%, but do not reach the 90% completeness required to be considered high quality. Indeed, for the intermediate quality bins, the detection rate of T3SS decreases to 0,4% compared to 1,1% for high quality bins.

REV 4.43 *How representative are the collected 44 reference genomes and 770 MAGs of the overall gut microbiome?*

AUTHOR RESPONSE

This question is difficult to answer, because ‘the microbiome’ does not exist and the microbiota in individuals differ enormously between age, environmental living conditions, diet and many other factors.

Second, because only *Pseudomonadota* have T3SS, these strains are not representative of the microbiome as a whole. In general, *Pseudomonadota* is the third most abundant phylum in the healthy human gut microbiome, which is otherwise dominated by *Bacteroidetes* and *Firmicutes*. Both are gram-positive eubacteria that have a cell wall and for which injection of effector proteins into the host is not a widespread mechanism of microbe-host interaction. As only a portion of the *Pseudomonadota* in all analyzed datasets are T3SS positive, these are not representative of the gut microbiome. Nonetheless, we believe that these could exert a profound effect on the host, as the impact of effectors can be amplified by the modulated immune signaling that we document.

Importantly, the analyzed strain collections are state-of-the art strain collections that represent the diversity of the gut microbiota as best as currently possible. Nonetheless, there will still be a cultivation bias and at this stage, the estimate of the proportion of T3SS+ strains is difficult to estimate and likely varies considerably between individuals.

REV 4.44 *How does the observed convergence in commensal effectors compare to the previously noted convergence in pathogenic effectors?*

AUTHOR RESPONSE

We are not sure how to compare the convergence to address the reviewer's question, especially given that, despite a similar overall question, there are substantial differences between the studies. Previously we studied candidate effectors from three evolutionary very distant pathogens (eubacteria, oomycete, fungus), whereas here we investigate the interactions of candidate effectors from 18 different bacterial strains with the human host. In addition to these different biological setups, there are also important technical differences that make a quantitative comparison more misleading than insightful. Beyond the already mentioned different numbers of microbes there are differences in the number of effectors studied for each of the microbes. Especially for the previously studied eukaryotic oomycete and fungus more effectors could be identified than for most of the bacterial strains. Beyond this, the number of host ORFs that could be tested differed substantially between the experiments. Finally, there are differences in how the experiment was conducted in that we previously used two repeat screens, not least because the search space was much smaller. Thus, we would respectfully point out that in our honest opinion, the 'convergence' observed in the different studies cannot be compared in a meaningful way.

References response letter

- 1 Yu, H. *et al.* High-quality binary protein interaction map of the yeast interactome network. *Science* **322**, 104-110 (2008). <https://doi.org/10.1126/science.1158684>
- 2 Cusick, M. E. *et al.* Literature-curated protein interaction datasets. *Nat Methods* **6**, 39-46 (2009).
- 3 Osborne, R. *et al.* Symbiont-host interactome mapping reveals effector-targeted modulation of hormone networks and activation of growth promotion. *Nat Commun* **14**, 4065 (2023). <https://doi.org/10.1038/s41467-023-39885-5>
- 4 Del Toro, N. *et al.* The IntAct database: efficient access to fine-grained molecular interaction data. *Nucleic Acids Res* **50**, D648-D653 (2022). <https://doi.org/10.1093/nar/gkab1006>
- 5 van Kempen, M. *et al.* Fast and accurate protein structure search with Foldseek. *Nat Biotechnol* **42**, 243-246 (2024). <https://doi.org/10.1038/s41587-023-01773-0>
- 6 Varadi, M. *et al.* AlphaFold Protein Structure Database in 2024: providing structure coverage for over 214 million protein sequences. *Nucleic Acids Res* **52**, D368-D375 (2024). <https://doi.org/10.1093/nar/gkad1011>
- 7 Barrio-Hernandez, I. *et al.* Clustering predicted structures at the scale of the known protein universe. *Nature* **622**, 637-645 (2023). <https://doi.org/10.1038/s41586-023-06510-w>
- 8 Deng, W. Y. *et al.* Assembly, structure, function and regulation of type III secretion systems. *Nature Reviews Microbiology* **15**, 323-337 (2017). <https://doi.org/10.1038/nrmicro.2017.20>
- 9 Miwa, H. & Okazaki, S. How effectors promote beneficial interactions. *Current Opinion in Plant Biology* **38**, 148-154 (2017). <https://doi.org/10.1016/j.pbi.2017.05.011>
- 10 Egan, F., Barret, M. & O'Gara, F. The SPI-1-like Type III secretion system: more roles than you think. *Front Plant Sci* **5**, 34 (2014). <https://doi.org/10.3389/fpls.2014.00034>
- 11 Yang, X. B., Pan, J. F., Wang, Y. & Shen, X. H. Type VI Secretion Systems Present New Insights on Pathogenic Yersinia. *Frontiers in Cellular and Infection Microbiology* **8** (2018). <https://doi.org/10.3389/fcimb.2018.00260>
- 12 Almeida, A. *et al.* A new genomic blueprint of the human gut microbiota. *Nature* **568**, 499-504 (2019). <https://doi.org/10.1038/s41586-019-0965-1>
- 13 Pasolli, E. *et al.* Extensive Unexplored Human Microbiome Diversity Revealed by Over 150,000 Genomes from Metagenomes Spanning Age, Geography, and Lifestyle. *Cell* **176**, 649-662 e620 (2019). <https://doi.org/10.1016/j.cell.2019.01.001>
- 14 Bowers, R. M. *et al.* Minimum information about a single amplified genome (MISAG) and a metagenome-assembled genome (MIMAG) of bacteria and archaea. *Nat. Biotechnol.* **35**, 725-731 (2017). <https://doi.org/10.1038/nbt.3893>
- 15 Westerhausen, S. *et al.* A NanoLuc luciferase-based assay enabling the real-time analysis of protein secretion and injection by bacterial type III secretion systems. *Mol Microbiol* **113**, 1240-1254 (2020). <https://doi.org/10.1111/mmi.14490>
- 16 Braun, P. *et al.* An experimentally derived confidence score for binary protein-protein interactions. *Nat. Methods* **6**, 91-97 (2009). <https://doi.org/10.1038/nmeth.1281>
- 17 Altmann, M. *et al.* Extensive signal integration by the phytohormone protein network. *Nature* **583**, 271-276 (2020). <https://doi.org/10.1038/s41586-020-2460-0>
- 18 Wang, J. W. *et al.* BastionHub: a universal platform for integrating and analyzing substrates secreted by Gram-negative bacteria. *Nucleic Acids Research* **49**, D651-D659 (2021). <https://doi.org/10.1093/nar/gkaa899>
- 19 Eddy, S. R. Accelerated Profile HMM Searches. *PLoS Comput Biol* **7**, e1002195 (2011). <https://doi.org/10.1371/journal.pcbi.1002195>
- 20 Kim, D. K. *et al.* A proteome-scale map of the SARS-CoV-2-human contactome. *Nat Biotechnol* (2022). <https://doi.org/10.1038/s41587-022-01475-z>
- 21 Mukhtar, M. S. *et al.* Independently evolved virulence effectors converge onto hubs in a plant immune system network. *Science* **333**, 596-601 (2011). <https://doi.org/10.1126/science.1203659>
- 22 Consortium, A. I. M. Evidence for network evolution in an Arabidopsis interactome map. *Science* **333**, 601-607 (2011). <https://doi.org/10.1126/science.1203877>
- 23 Choi, S. G. *et al.* Maximizing binary interactome mapping with a minimal number of assays. *Nat. Commun.* **10**, 3907 (2019). <https://doi.org/10.1038/s41467-019-11809-2>
- 24 Braun, P. *et al.* An experimentally derived confidence score for binary protein-protein interactions. *Nat Methods* **6**, 91-97 (2009).
- 25 Wessling, R. *et al.* Convergent targeting of a common host protein-network by pathogen effectors from three kingdoms of life. *Cell host & microbe* **16**, 364-375 (2014).

- <https://doi.org/10.1016/j.chom.2014.08.004>
- 26 Mukhtar, M. S. *et al.* Independently evolved virulence effectors converge onto hubs in a plant immune system network. *Science* **333**, 596-601 (2011). <https://doi.org/10.1126/science.1203659>
- 27 Lloyd-Price, J. *et al.* Multi-omics of the gut microbial ecosystem in inflammatory bowel diseases. *Nature* **569**, 655-+ (2019). <https://doi.org/10.1038/s41586-019-1237-9>
- 28 Franzosa, E. A. *et al.* Gut microbiome structure and metabolic activity in inflammatory bowel disease. *Nature Microbiology* **4**, 293-305 (2019). <https://doi.org/10.1038/s41564-018-0306-4>
- 29 Davey, N. E. *et al.* Attributes of short linear motifs. *Mol Biosyst* **8**, 268-281 (2012). <https://doi.org/10.1039/c1mb05231d>
- 30 Burke, D. F. *et al.* Towards a structurally resolved human protein interaction network. *Nat Struct Mol Biol* **30**, 216-225 (2023). <https://doi.org/10.1038/s41594-022-00910-8>
- 31 Choteau, S. *et al.* mimicINT: A workflow for microbe-host protein interaction inference [version 2; peer review: 2 approved, 1 approved with reservations]. *F1000Research* **14** (2025). <https://doi.org/10.12688/f1000research.160063.2>
- 32 Brunk, E. *et al.* Recon3D enables a three-dimensional view of gene variation in human metabolism. *Nat Biotechnol* **36**, 272-281 (2018). <https://doi.org/10.1038/nbt.4072>
- 33 Neduva, V. & Russell, R. B. Linear motifs: evolutionary interaction switches. *FEBS Lett* **579**, 3342-3345 (2005). <https://doi.org/10.1016/j.febslet.2005.04.005>
- 34 Samano-Sanchez, H. & Gibson, T. J. Mimicry of Short Linear Motifs by Bacterial Pathogens: A Drugging Opportunity. *Trends Biochem Sci* **45**, 526-544 (2020). <https://doi.org/10.1016/j.tibs.2020.03.003>
- 35 Via, A., Uyar, B., Brun, C. & Zanzoni, A. How pathogens use linear motifs to perturb host cell networks. *Trends Biochem Sci* **40**, 36-48 (2015). <https://doi.org/10.1016/j.tibs.2014.11.001>
- 36 Li, T. *et al.* SET-domain bacterial effectors target heterochromatin protein 1 to activate host rDNA transcription. *EMBO Rep* **14**, 733-740 (2013). <https://doi.org/10.1038/embor.2013.86>
- 37 Rolando, M. *et al.* The SET and ankyrin domains of the secreted *Legionella pneumophila* histone methyltransferase work together to modify host chromatin. *mBio* **14**, e0165523 (2023). <https://doi.org/10.1128/mbio.01655-23>
- 38 Muley, V. Y., Akhter, Y. & Galande, S. PDZ Domains Across the Microbial World: Molecular Link to the Proteases, Stress Response, and Protein Synthesis. *Genome Biol Evol* **11**, 644-659 (2019). <https://doi.org/10.1093/gbe/evz023>
- 39 Dosztanyi, Z., Csizmek, V., Tompa, P. & Simon, I. The pairwise energy content estimated from amino acid composition discriminates between folded and intrinsically unstructured proteins. *J Mol Biol* **347**, 827-839 (2005). <https://doi.org/10.1016/j.jmb.2005.01.071>
- 40 Meszaros, B., Erdos, G. & Dosztanyi, Z. IUPred2A: context-dependent prediction of protein disorder as a function of redox state and protein binding. *Nucleic Acids Res* **46**, W329-W337 (2018). <https://doi.org/10.1093/nar/gky384>
- 41 Erdos, G., Pajkos, M. & Dosztanyi, Z. IUPred3: prediction of protein disorder enhanced with unambiguous experimental annotation and visualization of evolutionary conservation. *Nucleic Acids Res* **49**, W297-W303 (2021). <https://doi.org/10.1093/nar/gkab408>
- 42 Edwards, R. J. & Palopoli, N. Computational prediction of short linear motifs from protein sequences. *Methods Mol Biol* **1268**, 89-141 (2015). https://doi.org/10.1007/978-1-4939-2285-7_6
- 43 Hagai, T., Azia, A., Babu, M. M. & Andino, R. Use of host-like peptide motifs in viral proteins is a prevalent strategy in host-virus interactions. *Cell reports* **7**, 1729-1739 (2014). <https://doi.org/10.1016/j.celrep.2014.04.052>
- 44 Zanzoni, A., Spinelli, L., Braham, S. & Brun, C. Perturbed human sub-networks by *Fusobacterium nucleatum* candidate virulence proteins. *Microbiome* **5**, 89 (2017). <https://doi.org/10.1186/s40168-017-0307-1>
- 45 Nourani, E., Khunjush, F. & Durmus, S. Computational approaches for prediction of pathogen-host protein-protein interactions. *Front Microbiol* **6**, 94 (2015). <https://doi.org/10.3389/fmicb.2015.00094>
- 46 Zhang, Y., Thomas, J. P., Korcsmaros, T. & Gul, L. Integrating multi-omics to unravel host-microbiome interactions in inflammatory bowel disease. *Cell Rep Med* **5**, 101738 (2024). <https://doi.org/10.1016/j.xcrm.2024.101738>
- 47 Becerra, A., Bucheli, V. A. & Moreno, P. A. Prediction of virus-host protein-protein interactions mediated by short linear motifs. *BMC Bioinformatics* **18**, 163 (2017). <https://doi.org/10.1186/s12859-017-1570-7>
- 48 Guven-Maiorov, E. *et al.* HMI-PRED: A Web Server for Structural Prediction of Host-Microbe Interactions Based on Interface Mimicry. *J Mol Biol* **432**, 3395-3403 (2020).

- <https://doi.org/10.1016/j.jmb.2020.01.025>
- 49 Lasso, G. *et al.* A Structure-Informed Atlas of Human-Virus Interactions. *Cell* **178**, 1526-1541 e1516 (2019). <https://doi.org/10.1016/j.cell.2019.08.005>
- 50 Reys, V., Pons, J. L. & Labesse, G. SLiMAN 2.0: meaningful navigation through peptide-protein interaction networks. *Nucleic Acids Res* **52**, W313-W317 (2024). <https://doi.org/10.1093/nar/gkae398>
- 51 Andrighetti, T., Bohar, B., Lemke, N., Sudhakar, P. & Korcsmaros, T. MicrobioLink: An Integrated Computational Pipeline to Infer Functional Effects of Microbiome-Host Interactions. *Cells* **9** (2020). <https://doi.org/10.3390/cells9051278>
- 52 Reys, V. & Labesse, G. SLiMAN: An Integrative Web Server for Exploring Short Linear Motif-Mediated Interactions in Interactomes. *J Proteome Res* **21**, 1654-1663 (2022). <https://doi.org/10.1021/acs.jproteome.1c00964>
- 53 Cunningham, J. M., Koytiger, G., Sorger, P. K. & AlQuraishi, M. Biophysical prediction of protein-peptide interactions and signaling networks using machine learning. *Nat Methods* **17**, 175-183 (2020). <https://doi.org/10.1038/s41592-019-0687-1>
- 54 Forster, S. C. *et al.* A human gut bacterial genome and culture collection for improved metagenomic analyses. *Nat Biotechnol* **37**, 186-192 (2019). <https://doi.org/10.1038/s41587-018-0009-7>
- 55 Hitch, T. C. A. *et al.* HiBC: a publicly available collection of bacterial strains isolated from the human gut. *Nat. Commun.* **16**, 4203 (2025). <https://doi.org/10.1038/s41467-025-59229-9>
- 56 Poyet, M. *et al.* A library of human gut bacterial isolates paired with longitudinal multiomics data enables mechanistic microbiome research. *Nat Med* **25**, 1442-1452 (2019). <https://doi.org/10.1038/s41591-019-0559-3>
- 57 Groussin, M. *et al.* Elevated rates of horizontal gene transfer in the industrialized human microbiome. *Cell* **184**, 2053-2067 e2018 (2021). <https://doi.org/10.1016/j.cell.2021.02.052>
- 58 Jing, R. *et al.* DeepT3 2.0: improving type III secreted effector predictions by an integrative deep learning framework. *NAR Genom Bioinform* **3**, lqab086 (2021). <https://doi.org/10.1093/nargab/lqab086>
- 59 Goldberg, T., Rost, B. & Bromberg, Y. Computational prediction shines light on type III secretion origins. *Scientific reports* **6**, 34516 (2016). <https://doi.org/10.1038/srep34516>

**Review response for
“Meta-interactome network reveals human immune modulation by commensal T3SS-
effectors”**

We extend our gratitude to the reviewers and the editorial team for their thorough and constructive evaluation of our manuscript. All comments and recommendations have been carefully considered and addressed in the revised version. The manuscript has been amended to incorporate the reviewers' remarks and to enhance its overall clarity.

We hope that the revisions undertaken resolve the concerns raised and contribute to the improvement of the manuscript. A detailed, point-by-point response is provided below, outlining the modifications implemented during the revision process.

Editor (Remarks to the Author):

Thank you for your patience while your manuscript "Meta-interactome network reveals human immune modulation by commensal T3SS- effectors" was under peer-review at Nature Microbiology. It has now been seen by 4 referees, whose comments you will find at the of this email. You will see from their comments below that while they find your work of interest, some important points are raised. We are very interested in the possibility of publishing your study in Nature Microbiology, but would like to consider your response to these concerns in the form of a revised manuscript before we make a final decision on publication.

0.0.a. In particular, we recommend that you modify the manuscript and the figures to make them more accessible and clear to the readers.

AUTHOR RESPONSE

We thank the editors and reviewers for their helpful recommendation. We have revised the manuscript, figures and supplementary material to improve accessibility and clarity for readers. Specifically, we shortened the manuscript significantly and carefully re-worked the figure legends to provide clearer explanations of the data and modified the supplementary tables and callouts in the manuscript. We believe these changes improve the overall clarity and accessibility of the manuscript and the figures.

0.0.b. Additionally, we recommend that you clarify definitions of 'mild' pathogen and commensal strains used in the manuscript.

AUTHOR RESPONSE

We agree that 'mild' pathogen is indeed a very imprecise if not misleading phrase, which we used in the response letter referring to the *Salmonella* Thyphimurium strain for the injection assays. As this is not used in the manuscript, we hope it is acceptable if we omit a further discussion of this unfortunate term. We do now clarify our terminology relating to pathogens and commensals and discuss used databases and approaches in response to **REV3.5**.

Briefly, a strict dichotomy between pathogenic and commensal strains is difficult to establish, particularly as most of the strains included in our study have not been characterized in depth. Because pathogenic and commensal roles vary at the strain level, species-level annotations (e.g., *Escherichia coli*) do not always reflect the properties of all strains, some of which are commensal or even beneficial, such as the probiotic strain *E. coli* Nissle 1917. Our cross-reference with the gcPathogen database showed that only 7 of the 44 strains in our study fall within species annotated as pathogenic, confirming that most strains analysed are not classified as pathogens.

0.0.c. Some additional analysis to address R3's concern of biological relevance should also be added. The rest referees' reports are clear and the remaining issues should be straightforward to address.

AUTHOR RESPONSE

We now did an extensive domain analysis for the commensal effectors and describe this in the response to **REV3.6**.

Reviewer #1 (Remarks to the Author):

The authors have professionally and adequately addressed my previous comments. I have no further comments.

AUTHOR RESPONSE

We sincerely appreciate the reviewer's positive assessment of our revisions.

Reviewer #2 (Remarks to the Author):

I am content with the additional experiments performed by the authors to provide more biological relevance, and overall, I think the manuscript is improved, with a thought-provoking discussion and providing a wealth of interesting information. However, I find that for these useful data to be accessible to the readers, some minor changes could be made:

Rev 2.1 *I found reading the manuscript difficult because it was often not intuitive to know what results/figures the text was referencing. Except for the main figures, when referring to extended data or the supplementary tables, only the ED Fig or Table is mentioned, without specifying the panel/sheet. I do understand there is a summary sheet in each of the supplementary tables, but even looking at that is sometimes not very informative as, for example, in supplementary table 6 not straight-forward to understand when you are referring to your candidate commensal effectors, and when you are referring to the known “pathogenic” effectors, thus I find it impossible to follow where this is shown. Another example is line 354, where you refer to table 5 but I do not know where to look for the results. Overall I think that specifying which ED fig panel or table sheet you are alluding to would allow readers to understand the study better and find the information they are interested in more easily.*

AUTHOR RESPONSE

We agree with the reviewer that locating the relevant information within the supplementary tables was not straightforward given the large number of sheets contained in each table. However, the journal guidelines do not permit callouts of individual sheets or panels in the supplementary figures or tables. To address this issue and improve readability, we have now reorganized the supplementary material by more finely dividing the tables according to experiments and topics. This has increased the number of tables from 7 to 26 and reduced the number of sheets in each Supplementary Table. Thereby, each table has less sheets and more specific information relating to the respective callouts, which should facilitate finding the relevant information of interest. We have also included a *Guide to Supplementary Material* now to facilitate finding relevant information. We hope that these changes will substantially enhance readability and make it considerably easier for readers to navigate and extract the relevant information from the supplementary material.

EXCERPT FROM REVISED MANUSCRIPT

Please see the revised Supplementary Tables.

Rev 2.2 I think the data showing effector translocation really strengthens the study; however I would really appreciate if the “known pathogenic” effector names (eg Q99PZ6 is OspG from *Shigella*) were mentioned in the figure (fig 1e) or legend to facilitate reading

AUTHOR RESPONSE

We thank the reviewer for the appreciation of the additional experiments. We followed the suggestion to alter the effector names in the figure and to refer to the respective bacterial organism in the legend of Figure 1.

EXCERPT FROM REVISED MANUSCRIPT

e, Injection of indicated effectors by wt and Δ sctV (T3SS defective) *Salmonella* Typhimurium into HeLa cells detected by luminescence of reconstituted nano-luciferase (y-axis). Control pathogen effectors (left): sseJ (A0A0F6B1Q8), sopA (Q8ZNR3), pipB2 (A0A0F6B5H5) from *Salmonella* Typhimurium; yopJ (A0A0N9NCU6) from *Yersinia pseudotuberculosis*, and ipaH9.8 (Q8VSC3) and ospG (Q99PZ6) from *Shigella flexneri*. SipA is assay control as reference. Asterisks: statistically significant difference between the wt and Δ sctV-negative strains (Wilcoxon test, n = 5 biological repeats with four technical repeats each).

Rev 2.3 For figure2e, could the name of the control effector (espG) be in the figure or legend (preferably in the figure and then use the uniprot ID in the legend)? The uniprot identifier is not very useful. And maybe indicate it's your control as this is not in the legend I believe. Also, this is representative of how many repeats?

AUTHOR RESPONSE

We agree with the reviewer's suggestion and changed the effector name in the figure and the legend. Additionally, we indicated the pathogenic effector as control and added the number of repeats.

EXCERPT FROM REVISED MANUSCRIPT

Co-immunoprecipitation of Myc-tagged human proteins by FLAG-tagged effectors or FLAG-GFP as negative control. Input: cell lysates. Green dots: successful co-immunoprecipitation. Red dot: no co-immunoprecipitation. Effector espG of *Escherichia coli* (Q7DB50_ECO57) as positive control (one biological replicate).

Rev 2.4 I am confused by the text in lines 250-254: This refers to intriguing data for further experimentation, but it's difficult to see where the results are (if anywhere)? I know convergence is in Supp table 4, Sheet 40 but, as far as I can see, this doesn't show the pathogenic effectors?

AUTHOR RESPONSE

We thank the reviewer for raising this concern, which allowed us to refine the supplementary materials.

1. To improve clarity and comply with the journal's guidelines, we reorganized the supplementary tables, expanding them from 7 to 26 according to experiments and topics. For the example mentioned by the reviewer, the relevant information can now be found as follows: Supplementary Table 15 summarizes interactions between 80 pathogen effectors and 217 human proteins curated from the IMEx database, while Supplementary Table 20 presents the functional enrichment analysis of these 217 human proteins.
2. We have included a *Guide to Supplementary Material* for easier navigation and finding of relevant data.

Furthermore, we need to clarify a misunderstanding.

3. The convergence analysis reported in Supplementary Table 11, however, only pertains to effectors from commensals and not pathogens. We have now tried to make this more explicit. In contrast to the previously published plant work, which we refer to and cite in the sentences before, in the human system pathogen convergence could not be analyzed, as the available datasets are curated collections from hypothesis-driven approaches rather than from systematic experimental screens; the inherent bias precludes meaningful network analyses. However, we did check which of the 60 human proteins that effectors from the commensals converge on, have known interactions with pathogen effectors. We have added a column in Supplementary Table 11 indicating which human interaction partner of commensal effectors is also targeted by a pathogen effector.

Very minor errors to be corrected:

Rev 2.5 Wrong figure 1 callouts in line 139 onwards, as I think the order has been reshuffled: Fig 1c should be 1b (line 139), 1d to 1c (lines 145, 150) and 1b to 1d (line 154)

AUTHOR RESPONSE

We thank the reviewer for pointing out the incorrect figure callouts. We have carefully reviewed and corrected the figure references in the manuscript so that they now accurately correspond to the intended panels. We have verified all other figure callouts to ensure consistency throughout the text.

Rev 2.6 Line 342 onwards, callouts to supplementary table 7 I think are for supplementary table 6?. And supplementary table 8 (line 384 onwards) I believe is 7 (there is no 8).

AUTHOR RESPONSE

We thank the reviewer for bringing these inconsistencies to our attention. We have carefully reviewed the manuscript and updated the supplementary table callouts to reflect the reorganized table structure, which we implemented to improve clarity and comply with the journal's guidelines.

Reviewer #3 (Remarks to the Author):

Rev 3.1 *The first thing I want to mention is that, disappointingly, the response text to reviewer 3 (me) contains an assumption that the reviewer is male: "We thank the reviewer for this suggestion and have followed his advice". It is incredibly discouraging that in 2025 the default assumption remains that reviewers are male.*

AUTHOR RESPONSE

We agree. And we are genuinely embarrassed. And upset at our own thoughtlessness. As difficult as it likely is to believe now, we do consider inclusive language and reflection on our own biases and assumptions incredibly important for ourselves and our immediate working environment, for science, and for society as a whole. Because we do, our unreflective use of the male form is genuinely embarrassing. We share the reviewer's disappointment in our phrasing and take this as a reminder that even though we attach immense importance to the use of unbiased language and thinking, we need to stay attentive to this matter to not 'accidentally' fall into, and thereby promote, such anachronistic, yet still omnipresent, stereotypes and thinking.

The authors have gone to great lengths in the last two years to improve the paper, and it is potentially much stronger in many areas and could become a useful resource. However, as with the first version I think there is a significant problem with interpretability of figures and explanation of data, which has really hindered review and made the whole package hard to judge. I give some examples below.

Rev 3.2 *Fig 1b: there is not enough information on the graph or in the legend to interpret this. What is the histogram above vs the one below?*

AUTHOR RESPONSE

Thank you for pointing this out. We have clarified in the figure legend that the upper histogram shows marginal histograms displaying the distribution of alignment coverage, while the histogram to the right shows marginal histograms displaying the aggregated distribution of sequence similarity. Practically, these are binned summaries of the number of datapoints at the respective levels.

EXCERPT FROM REVISED MANUSCRIPT

b, Sequence similarity of 3,002 candidate commensal T3SS-effectors with 1,195 effectors from pathogenic bacteria across alignment coverages (left, bottom). Each dot represents a pairwise sequence comparison. Dot color indicates effectors with significant and insignificant Jackhammer results (inset legend) indicating homology to pathogen effectors. Marginal histograms display the aggregated distribution of alignment coverage (top) and aggregated sequence similarity (right) with color indicating Jackhammer outcome.

Rev 3.3 Fig 1c: what are those structures, mini networks and donut graphs? There are no labels and the legend does not explain at all.

AUTHOR RESPONSE

Our apologies for having a legend that insufficiently describes the panel. We added more details to the legend for a more precise explanation.

EXCERPT FROM REVISED MANUSCRIPT

c, Left: Number of the structure-clusters observed in Foldseek analysis (red arrow) compared to random expectation for that group (homogeneous or mixed) in grey (n = 10,000); middle: Example structures for one cluster in the group; small networks: a representative structure-cluster for the group with an anchor structure in the center and similar structures connected by links. Donut plots: proportion of proteins with origins indicated by color in all clusters of (homogeneous or mixed) structure-cluster group.

Rev 3.4 Fig 3d: what are those structures? they are not mentioned in the text

AUTHOR RESPONSE

The structures of interacting protein pairs, labelled with the gene symbol (human protein) and the effector names, respectively, illustrate the different interaction modes we describe in the text. In the revised manuscript, we clarified the purpose of the structures shown in Fig. 3d by referring to them as examples of the different interaction interface similarities in the results section. Specifically, we now highlight examples of effectors binding either the same or distinct interfaces on human proteins. This addition ensures that the structures depicted in Fig. 3d are directly explained in the text and linked to specific biological interpretations

EXCERPT FROM REVISED MANUSCRIPT

For instance, Mmo_5 binds to TPD52L and BROCS6 via the same interface, whereas Pfa_4 interacts with NOTO and LBX1 with different interfaces, possibly enabling simultaneous interactions with both (**Fig. 3d, Supplementary Table 17**). Analogously, Pse_2 and Pfa_9 bind to the same interface of TCF4, whereas Yen_2 targets a different part of the protein (**Fig. 3e**). Identical interface binding was more frequent on human proteins than on effectors, suggesting an importance of targeting functions linked to specific domains. Mapping the binding interfaces to domain annotations strengthened this hypothesis, as even effectors binding via different interfaces may target the same domain, e.g., the DNA-binding domain of LBX1.

Rev 3.5 *It is also unclear on what basis a bacterium is classified as commensal or pathogenic. Is there a database somewhere that lists strains and pathogenicity?*

AUTHOR RESPONSE

We appreciate the reviewer's comment for the opportunity to clarify the criteria for classifying strains as commensal or pathogenic. In practice, such a distinction is not straightforward, particularly because many of the strains in our study have not been comprehensively characterized. Species-level annotations may label an organism as pathogenic, but this does not necessarily reflect the behaviour of every individual strain. For example, some strains of *Escherichia coli* are highly pathogenic, such as EPEC or EHEC, other strains are harmless or even beneficial, such as *E. coli* Nissle 1917, which is used therapeutically as a probiotic¹. Practically, we considered strains identified in 'apparently' healthy individuals, i.e. not suffering from an obvious infection, as commensals. Conversely, we considered strains encoding effectors in the BastionHub database as pathogens - these are classic pathogens such as *Yersinia pestis*, *Pseudomonas aeruginosa*, *Salmonella enterica* and *Salmonella typhi*, and others.

Naturally, there is an inherent lack of clarity due to the niche and context dependent behaviour of many microbes, which we recognize. Indeed, a few of the commensal species may also become opportunistically pathogenic in other niches.

To provide a more systematic perspective, we picked up on the reviewer's suggestion to investigate pathogenicity of our strains based on a curated database for pathogenic microbes: gcPathogen². Thus, focusing on the reference strains we compared these against the 452 human bacterial pathogens listed in gcPathogen. 7 of the 44 strains in our study (including 3 of the 18 strains from which we cloned effectors) belong to species with pathogenic annotations (*Escherichia coli*, *Pseudomonas aeruginosa*, *Vibrio fluvialis*, *Aeromonas dhakensis*, *Aeromonas jandaei*, *Vibrio furnissii*, and *Enterobacter roggenkampii*). As for *E. coli*, there seem to be considerable differences between strains such that pathogenic and non-pathogenic strains may belong to the same species. The remaining strains are not classified as pathogens, and even within species that are annotated as pathogenic, strain-specific differences allow for commensal or beneficial roles, as illustrated by the example of *E. coli* Nissle 1917.

EXCERPT FROM REVISED MANUSCRIPT

We consulted a curated database for pathogenic microbes: gcPathogen² covering 452 human bacterial pathogens. 7 of the 44 strains in our study (including 3 of the 18 strains from which we cloned effectors) belong to species with pathogenic annotations (*E. coli*, *P. aeruginosa*, *V. fluvialis*, *V. furnissii*, *Aeromonas dhakensis*, *A. jandaei*, and *E. roggenkampii*). Species-level annotations may label an organism as pathogenic, but this does not necessarily reflect the behavior of every individual strain. For example, while some strains of *E. coli* are highly pathogenic and justify its inclusion in the gcPathogen, such as EPEC or EHEC, other strains are harmless or even beneficial, such as *E. coli* Nissle 1917, which is used therapeutically as

a probiotic¹. Practically, we considered strains identified in ‘apparently’ healthy individuals, i.e. not suffering from an obvious infection, as commensals.

Rev 3.6 *My concern about biological insight remains. There is little mention of functional annotation of the effectors in the text. Are there over represented domains in the effectors? I also didn't manage to find effector sequences in the SI, but maybe they are there somewhere.*

AUTHOR RESPONSE

Admittedly we had some difficulty with this comment by the reviewer in the first round, as we considered the bulk of the analyses in the manuscript to address “biological relevance” and we provide multiple levels of “biological insight”: hundreds of novel high-quality physical interactions show biochemical connectivity, the bioinformatic analyses reveal effector-targeted pathways and functions in the human host, the follow-up of specific hypotheses in functional assays validate these and demonstrate immune perturbation by the effectors and ultimately link to clinical observations suggesting a relevance to human health. In our view, all these analyses directly address the biological relevance of effectors in the human system and go a long way from discovery of a new phenomenon to concrete insights. In this mindset, we addressed the critique that our analysis was too shallow and insufficient, e.g., by exploring potential interactions of the CD enriched effectors in the human network, linking this to genetics and thereby providing further hypotheses that may act as leads for detailed follow-up studies.

Additional confusion emerged from the fact that we describe for the first time, that commensal strains can inject candidate proteins into human cells – a finding we consider a substantial biological insight. Because of the novelty, however, there were very few prior studies that would allow us to investigate potential other functions in the human host – which is precisely the focus of our work.

With the slightly rephrased question and specific suggestions, we believe we were now able to address the reviewer’s concern. As suggested, we conducted a systematic analysis of domains found in the 3,002 candidate effectors identified from strains. For all, we identified domains using InterPro scan now listed in **Supplementary Table 5**. For a comparison, we conducted the same analysis for effectors of human and vertebrate pathogens in BastionHub. We also conducted an enrichment analysis of domains identified in the strains against all domains annotated for proteins encoded in the respective genome, reasoning that a pooled analysis of all effectors against all proteins in all genomes would mix the underlying biology of the organism and thus be biologically meaningless if not misleading. However, the enrichment analysis for each genome identified only few enriched domains in effectors of organisms that we conducted no further analyses with. As these were also not particularly meaningful, we adhered to the absolute counts of domain occurrences in the commensal effectors and pathogen effectors as an aggregated analysis, and provide all domains identified in all effectors in an easy to find format in the **Supplementary Table 5** along with the identifiers used in the manuscript and Genbank accession numbers.

Notably, 860 of the candidate effectors identified in the reference strains did not contain any domain, thus constituting the by far largest group. The most abundant domains that were

identified were the Diguanylate cyclase, GGDEF domain (PF00990) (58 effectors), ABC transporter (PF00005) (57), EAL domain (PF00563) (50), none of which were found in pathogen effectors. The Diguanylate cyclase domain has been described to be involved in bacterial signal transduction. In recent years, dinucleotides have emerged as important immune regulators in all kingdoms of life, making a role in interkingdom communication plausible³. Intriguingly, the EAL domain is described as a candidate for a diguanylate phosphodiesterase function, thus opposing the effect of the cyclase⁴. The fact that these two domains acting on the same second messenger occur at high frequency among the commensal effector candidates further adds to the plausibility of the biological significance of this. The ABC transporter domain has the ATP cassette that provides energy to many transport processes. The domain is common among human proteins, and thus likely able to physically and functionally insert into the host interactome. Notable is also the PAS-fold (PF08447) domain (in 32 effectors), which also occurs in human proteins and is a ligand binding domain hypothesized to act as a sensor and in some effectors occurs together with the guanylate cyclase⁴.

Not unexpectedly given the FoldSeek analysis, few commensal effector domains are shared with pathogens, and the annotations of these are informed by the knowledge from pathogens, such as virulence or secretion system effector (e.g. PF04888, PF06416).

We have inserted a brief description of this analysis after the fold-seek analysis along with a callout to the relevant **Supplementary Table 5** with all domain related data. In conjunction with the substantially reduced complexity of the individual Supplementary Tables, a new *Guide to Supplementary Material* and the more specific callouts, we genuinely hope that the reviewer can locate the respective data better now.

EXCERPT FROM REVISED MANUSCRIPT

We analysed all candidate effectors from the strains for annotated domains. Besides 860 proteins without any identifiable domain, among the most common finds were the diguanylate cyclase, GGDEF domain (PF00990) (58 effectors), and EAL domain (PF00563) (50), none of which was found in pathogen effectors (**Supplementary Table 5**). Cyclic-Diguanylate is a known second messenger in bacterial signal transduction and the EAL domain is thought to be a diguanylate phosphodiesterase thus opposing the effect of the cyclase⁴. As cyclic dinucleotides recently emerged as important immune regulators in all kingdoms of life³, the observation that two domains acting on the same second messenger occur at high frequency among the commensal effector candidates make a role for this signaling molecule in interkingdom communication plausible.

[...]

Consistently, Cpa_12, an ABC domain containing effector, reduced secretion of several cytokines with and without stimulation (**Fig. 5c, Supplementary Table 25**).

minor:

Rev 3.7 line 101: "that were isolated by the human microbiome project and other efforts." I feel this needs a reference

AUTHOR RESPONSE

We thank the reviewer for highlighting this point. We have added a reference to support the statement regarding the Human Microbiome Project and clarified additional sources from which the strains were isolated.

EXCERPT FROM REVISED MANUSCRIPT

We first analyzed reference genomes of *Pseudomonadota* strains from healthy gut and stool samples isolated, e.g., by the human microbiome project⁵.

Rev 3.8 fig 1b is cited after 1c

AUTHOR RESPONSE

We thank the reviewer for noticing this. We have corrected the manuscript so that Fig. 1b is cited in the correct order before Fig. 1c and have verified all other figure callouts for consistency.

Reviewer #4 (Remarks to the Author):

The revision and the point-by-point response letter properly and comprehensively resolved all but one previously raised comments.

Rev 4.1 *As for R4.7, to clarify the original comment: in some cases, host proteins have enzymatic domains, such as proteases that can cleave SLiMs on bacterial proteins. In these scenarios the direction is host -> microbe, and not the other way around - hence the comment on the importance on directionality in the proposed workflow. All other points are fine.*

AUTHOR RESPONSE

We agree that this is an important point and thank the reviewer for the opportunity to clarify our thinking. We do not dispute the importance of directionality or the possibility of information flow in both directions – just the contrary.

What we meant to say is that the architecture of an interaction, i.e. which protein has the SLiM with motifs and which protein carries the domain that the SLiM can bind to, is independent of the directionality of functional ‘effect’. We believe we fundamentally agree with the reviewer but express the same thought in slightly different ways.

Indeed, human proteins may “use” a domain to recognize or neutralize a bacterial protein, e.g. as part of a defence response. At the same time, the bacterial SLiM may block a signaling node to manipulate host responses. This is exactly what we meant to express: the architecture of a given interaction (bacterial SLiM, human domain), does not imply a functional directionality from SLiM to domain; information and functional effects may also go from domain to SLiM. Thus, even though we looked for bacterial SLiMs binding to human domains, we do not think that this interaction architecture relates to any single direction of information flow while excluding the other. In addition to the protease domain the reviewer cites, kinases are common enzymes that phosphorylate their substrates in SLiMs thus providing a non-destructive mode of domain-to-SLiM information flow.

That said, as we previously wrote, SLiMs rapidly evolve and from an evolutionary perspective it is more likely that a SLiM-mediated interaction involving a eukaryotic like interaction domain favours the microbe. Otherwise, the interacting sequence may be rapidly counter-selected and disappear from the pool. However, whether the effect occurs via competitive inhibition of an intracellular host pathway, or potentially by ‘receiving’ an activating enzymatic modification can only be solved by detailed mechanistic studies and not inferred from the architecture.

The reason why we did not investigate human SLiMs binding to bacterial SLiM binding domains was merely that the bacterial effectors do not contain domains known to bind SLiMs.

We do hope that this clarifies our response.

REFERENCES

- 1 Sassone-Corsi, M. & Raffatellu, M. No vacancy: how beneficial microbes cooperate with immunity to provide colonization resistance to pathogens. *J Immunol* **194**, 4081-4087 (2015). <https://doi.org/10.4049/jimmunol.1403169>
- 2 Guo, C. *et al.* gcPathogen: a comprehensive genomic resource of human pathogens for public health. *Nucleic Acids Res* **52**, D714-D723 (2024). <https://doi.org/10.1093/nar/gkad875>
- 3 Zaver, S. A. & Woodward, J. J. Cyclic dinucleotides at the forefront of innate immunity. *Curr Opin Cell Biol* **63**, 49-56 (2020). <https://doi.org/10.1016/j.ceb.2019.12.004>
- 4 Galperin, M. Y., Nikolskaya, A. N. & Koonin, E. V. Novel domains of the prokaryotic two-component signal transduction systems. *FEMS Microbiol Lett* **203**, 11-21 (2001). <https://doi.org/10.1111/j.1574-6968.2001.tb10814.x>
- 5 Nelson, K. E. *et al.* A catalog of reference genomes from the human microbiome. *Science* **328**, 994-999 (2010). <https://doi.org/10.1126/science.1183605>

**Review response for
“Meta-interactome network reveals human immune modulation by commensal T3SS-
effectors”**

We thank the reviewer for the careful assessment of our revised manuscript. We have incorporated all suggestions. The manuscript has been updated to reflect the reviewer's input and substantially expand the discussion section.

Editor (Remarks to the Author):

Thank you for submitting your revised manuscript "Meta-interactome network reveals human immune modulation by commensal T3SS-effectors" (NMICROBIOL-23092521B). It has now been seen by the original referees and their comments are below. The reviewers find that the paper has improved in revision, and therefore we'll be happy in principle to publish it in Nature Microbiology, pending minor revisions to satisfy the referees' final requests and to comply with our editorial and formatting guidelines.

Reviewer #4 (Remarks to the Author):

In this revision round, Young et al further improved their manuscript. This time, I specifically looked at the responses and actions for the concerns made by Reviewer 3 and myself.

Rev 4.1 *The responses to Reviewer 3's previous comments are fine, but not all made fully to the manuscript. Figure legends and description of the results shown on the figures have been added but the responses for Rev 3.6 and Rev 3.7 are more detailed in the response letter and less in the manuscript. This actually points to the problem of the Discussion section of the manuscript and less the Results section anymore.*

AUTHOR RESPONSE

We have now expanded the discussion of the distinction between commensals and pathogens, relating to Rev 3.7, and the domain analysis of the effector proteins that relate to question Rev 3.6 in the previous revision.

EXCERPT FROM REVISED MANUSCRIPT

Interpreting T3SS functionality in the human gut requires moving beyond species-level labels such as “commensal” or “pathogenic,” which often obscure substantial within-species diversity. As observed in other host kingdoms⁵³, these categories are fluid: *E. coli* includes both highly pathogenic lineages, e.g., EPEC, EHEC, and harmless or beneficial ones, like the probiotic *E. coli* Nissle 1917⁵⁴. In our analyses, strains isolated from apparently healthy individuals were considered commensals, whereas strains encoding known virulence effectors²², including *P. aeruginosa* and *Salmonella* spp, were designated pathogens. Importantly, between these poles lie opportunistic pathogens, whose infectious potential

emerges only in specific environmental, or host related conditions. A key question is therefore whether commensal T3SS primarily support opportunistic pathogenicity, or whether they have adaptive functions in the non-pathogenic lifestyle. Multiple lines of evidence from our study support the latter.

Comparative sequence and structure analyses revealed that commensal and pathogen effector repertoires are largely distinct, supporting a model in which commensal T3SS are adapted for cooperative rather than pathogenic interactions. Homotypic clustering of effector structures and depletion of mixed commensal–pathogen clusters indicate that commensal effectors follow separate selective trajectories. The domain analysis supports this, revealing many domains found only in commensal effectors likely for a non-pathogenic lifestyle. Notably, numerous effectors involved in cyclic diguanylate synthesis or degradation, often paired with PAS sensor domains, were identified suggesting environmentally responsive functions. Intriguingly, several effectors from gram-negative commensal *Pseudomonadota* potentiated Pam3CSK4-induced TLR1/2 signaling, suggesting that T3SS may modulate host responses to gram-positive Bacteroidetes and thereby influence inter-phyla competition within the gut ecosystem.

Rev 4.2 *The Discussion section is currently too short and not covering key aspects, including gained biological insight, constraints and future opportunities. The limitations at the end of the current manuscript are very superficial and short. I recommend the authors to extend the Discussion section substantially with more details prompted by the comments of the reviewers, especially Reviewer 3, and provide a Discussion that both highlights better their nice work and its limitations.*

AUTHOR RESPONSE

We have substantially expanded the entire section, to now more deeply discuss our findings along with alternative interpretations of the data and limitations of our study.

Rev 4.3a *As for the response for my previous comment on the directionality of the bacterial-host interactions, I cannot accept the answer. I agree with the authors that just using structural information, we often cannot determine the direction. However, in an interaction between a catalytic domain (kinase, protease, etc) and its target motif, it is highly common that the flow of signal is from the domain to the protein having the motif. Even for scaffold proteins with motifs and other proteins with binding domains, the direction of signal flow can be inferred, as such interactions influence the activity of the proteins bound by the scaffold. All this should be clarified and considered in the manuscript.*

AUTHOR RESPONSE

We agree with the reviewer that for interactions between a catalytic domain and second protein that carries the recognition sequence it is exceedingly likely that a catalytic domain modifies the SLiM encoded substrate. When both partners are host proteins this usually corresponds to information flow from, say, the kinase to the substrate. Previously we were considering the situation that the SLiMs could be pseudo substrates and thus inhibit the catalytic domain – in our data this is not the case, though, and so we drop the argument, even though in principle that situation may arise.

Another part of our argument is that we considered the injected effector the ‘new information’ in the host cell, which may tap into the host signaling network to properly execute its function. Thus, it is possible that the bacterium, via the effector being phosphorylated by the kinase, to obtain information about the state of the host and thereby ‘listens in’ on the information flow in the host or to mimic phosphorylation dependent binding interfaces. This is exemplified by the *H. pylori* effector CagA, which in all strains has 2 conserved Tyr-phosphorylation motifs (EPIYA-A and -B), but a third motif (EPIYA-C) is prevalent in strains in Europe and North America and a fourth is prevalent in East Asian strains. Moreover, the kinases phosphorylating CagA at these sites, c-SRC and ABL, appear to be activated sequentially by the bacterium. Thus, even though there is an immediate effect of the kinase phosphorylating the substrate, this phosphorylation supports effector function for the bacterium’s benefit. The reciprocal possibility is that a PTM is a defence mechanism, e.g. to degrade the foreign protein, however, in such a case the substantial selective pressure would presumably lead to rapid counterselection and degeneration of the motif. Thus, while there is indeed a very high likelihood of a biochemical directionality from the domain to the SLiM, we maintain that most of the resulting modifications support the lifestyle of the injecting bacterium. We now express this complexity in the discussion as advised (see below).

Rev 4.3b *Most importantly, the authors’ response that “The reason why we did not investigate human SLiMs binding to bacterial SLiM binding domains was merely that the bacterial effectors do not contain domains known to bind SLiMs.” is unfortunately incorrect and scientifically not acceptable. For bacterial effectors with domains binding SLiMs, here are a few examples: <https://www.science.org/doi/10.1126/science.1158160>, <https://pmc.ncbi.nlm.nih.gov/articles/PMC4715502>. Therefore, the authors should clarify the focus of their work, detail the selective nature of their workflow to direction, and further discuss the limitations in the Discussion section.*

AUTHOR RESPONSE

Indeed, this was an internal miscommunication, for which we apologize. We also thank the reviewer for bringing to our attention the cases of the Ankyrin repeats and HD domains in *Legionella* effectors. Ankyrin repeats are known to bind cognate SLiMs and can potentially recognise such in the host proteins, although to the best of our knowledge, the motifs bound by bacterial ankyrin repeats are not known. We have now checked our dataset for interactions potentially mediated by this binding mode. We searched for motif-binding domains in sequences of effectors with at least one main screen interaction. Of the identified 5 effectors with such domains, only in one pair, Efe1 - VAC14, the domain of the effector matches the cognate SLiM of the human interactor. Efe1 has a Calcineurin-like phosphoesterase domain (PF00149) that, according to ELM, recognises DOC_PP2B_LxvP_1 motifs. LxVP docking

motifs are present on substrates and regulators of active calcineurin and VAC14 has such motif. We now mention this in the results section. While this observation proves the existence of such pairings if the low number is due to their paucity or technical reasons is unknown and discussed in the revised discussion. As we previously mentioned, host-like non-enzymatic motif-binding domains have been detected in prokaryotic proteomes. Although some of them have kept their original function, others have lost their "eukaryotic" functions as either their sequences have significantly diverged and are not able to bind the cognate motif (e.g., bacterial SH3 domains), or the bacterial proteins bearing them are not necessarily localized at the host interface thus impeding the recognition of potential cognate motifs in host proteins (e.g., bacterial PDZ domains).

We have amended the manuscript to underline that our approach is focused on the detection of host-like SLiMs in effectors and included its limitations in the discussion.

EXCERPTS FROM REVISED MANUSCRIPT

RESULTS

Some of the matched motifs encompass phosphorylation sites that interact with kinases or phosphorylation-dependent binding domains such as SH2 domains. Although, conversely, several commensal effectors encode predicted enzymatic domains (**Supplementary Data 5**), using an analogous approach we found no case in which these engage cognate substrate motifs on host proteins, and only a single effector-domain-SLiM match consistent with known docking specificity: the calcineurin-like phosphoesterase domain (PF00149) of Efe1 and the canonical LxVP docking motif in VAC14.

Discussion

The interaction-structure models provide leads for dissecting effector mechanisms. Targeted host protein domains can indicate which processes an effector may perturb and, when mediated by a corresponding motif, whether the effector may get post-translationally modified, as seen for *H. pylori* CagA⁵³. Conversely posttranslational modification of host proteins is a common mechanism of pathogenicity⁵⁴. A manual analysis matching the mimicINT workflow, however, revealed no clear cases in which these engage cognate host substrate motifs and only a single example consistent with known docking specificity. If this reflects differences between commensal and pathogen effectors, the prevalence of functional mimicry without sequence similarity⁵⁴, or merely limitations of our approach remain to be clarified. Post-translational modification of effectors by host enzymes may either enhance effector function or act in host defence, such as by targeting foreign proteins for degradation. The latter would, however, be expected to select against motif retention. Thus, while biochemical directionality from host-domain to effector-SLiM is plausible, the available evidence suggests that such modifications predominantly support the lifestyle of the injecting bacterium. When

commensals act as pathobionts and contribute to non-communicable diseases, such interactions may become intervention targets.

Additional, minor comments:

Rev 4.4 Line 336-337. *“Among immune diseases, CD (nominal $P = 8.5 * 10^{-5}$, Fisher’s exact test) and IBD (nominal $P = 0.0008$, Fisher’s exact test) were enriched, but not ulcerative colitis (UC) (Fig. 4d, Supplementary Table 23).” As discussed later in the manuscript, CD and UC are both IBDs. Therefore, this sentence needs to be rephrased, perhaps first discuss CD vs UC, and then clarify and explain that for IBD still enrichment was found.*

AUTHOR RESPONSE

We have rearranged that sentence.

EXCERPTS FROM REVISED MANUSCRIPT

Among immune diseases, IBD were enriched (nominal $P = 0.0008$, Fisher’s exact test), particularly CD (nominal $P = 8.5 * 10^{-5}$, Fisher’s exact test), but not ulcerative colitis (UC) (**Fig. 4d, Supplementary Data 23**).

Rev 4.5 Line 366. *“...we analyzed a metagenome study with > 800 IBD patients and > 300 healthy controls (ref47).” Specify the number of IBD patients and healthy controls – “more than 800” / “more than 300” are not clear enough.*

AUTHOR RESPONSE

We have added this information.

EXCERPTS FROM REVISED MANUSCRIPT

Hypothesizing that causal involvement in IBD etiology may be reflected in altered effector prevalence, we analyzed a metagenome study with > 800 IBD patients (504 CD, 302 UC) and 334 healthy controls⁴⁸.

Rev 4.6 *Throughout the manuscript, many resources or communities are being written with all lower case, such as “gene ontology”, “open targets”, etc. Please have uppercase letter for the first words of these entities.*

AUTHOR RESPONSE

We have checked the most common spelling and accordingly adjusted all mentions of Open Targets and Gene Ontology to the widely used capitalization of both words.